# Practical do-Shapley Explanations with Estimand-Agnostic Causal Inference

**Álvaro Parafita**[1]    **Tomas Garriga**[1,2]    **Axel Brando**[1]    **Francisco J. Cazorla**[1]

[1]Barcelona Supercomputing Center    [2]Novartis

{alvaro.parafita,axel.brando,francisco.cazorla}@bsc.es
tomas.garriga_dicuzzo@novartis.com

## Abstract

Among explainability techniques, SHAP stands out as one of the most popular, but often overlooks the causal structure of the problem. In response, do-SHAP employs interventional queries, but its reliance on estimands hinders its practical application. To address this problem, we propose the use of estimand-agnostic approaches, which allow for the estimation of any identifiable query from a single model, making do-SHAP feasible on complex graphs. We also develop a novel algorithm to significantly accelerate its computation at a negligible cost, as well as a method to explain inaccessible Data Generating Processes. We demonstrate the estimation and computational performance of our approach, and validate it on two real-world datasets, highlighting its potential in obtaining reliable explanations.

## 1 Introduction

The widespread adoption of Machine Learning (ML) systems has raised concerns about their limitations: models can replicate human biases [1], base their outcomes on spurious correlations [2], or be vulnerable to malicious adversarial attacks [3]. Since most of these systems are black-boxes, there is an ever-increasing need for explainability techniques to make sense of the model. This is especially relevant *w.r.t.* fairness, the right to explanation [4], debugging, auditing, and fostering user trust in the system.

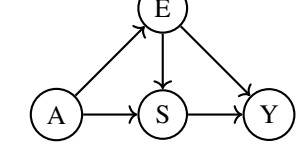

Figure 1: *Salary* causal graph: Age (A), Education (E), Seniority (S) and Salary (Y).

In response to this pressing need, the field of explainability has steadily gained traction, resulting in several approaches [5] to explain model predictions. Among them, the Shapley value (SV, or SHAP) [6] is one of the most popular, being the unique attribution strategy fulfilling a set of axioms aligned with human intuition (see appendix A). SVs are derived from a *value function* $\nu$ measuring the effect of a subset (*coalition*) **S** of features **X** on the model's prediction; each $\nu$ results in a different kind of SV, the most common being *marginal* and *conditional* SHAP [7].

However, both of these options ignore the *causal structure* underlying the data; for instance, fig. 1 represents the salary $Y$ of an employee of age $A$ with a certain education level $E$ and seniority level $S$. Let $f$ be a ML model $f(\mathbf{X}) \approx \mathbb{E}[Y \mid \mathbf{X}]$, learning $Y$ given inputs $\mathbf{X} = \{A, E, S\}$. Consider $\nu(\{E\})$. In marginal SHAP, $\nu$ assigns values $\{E = e\}$ and marginalizes the complementary set $(a, s) \sim \mathcal{P}(A, S)$ regardless of how the coalition's values causally affect them ($E \rightarrow S$). Conditional SHAP does consider these effects, but conditionally, $(a, s) \sim \mathcal{P}(A, S \mid E = e)$, producing anti-causal effects to $A$ (i.e., we cannot change *age* by granting them a degree). Thus, we cannot ignore the problem's causal structure. Please refer to appendix G for an extended discussion on this example.

Several works [8, 9, 10, 11] discussed the limitations of non-causal SHAP and proposed limited approaches with a causal interpretation. Jung *et al*. [12] proposed do-SHAP, defining $\nu$ as a causal,

interventional query estimated with Causal Inference: first transforming the query into a probabilistic formula (the *estimand*) only containing terms from the observational distribution $P(\mathbf{X})$, training ML models on these terms to estimate them and bringing it back into the formula for the final estimation. The main drawback of *estimand-based* (EB) estimation is that do-SVs require computing up to $2^{|\mathbf{X}|}$ causal queries, one for each coalition $\mathbf{S} \subseteq \mathbf{X}$, with different estimands and ML models to estimate their terms. As a result, the process becomes too manual and impractical for complex graphs. In fact, do-SHAP's authors stated they "are not aware of any general causal effect estimators suitable for estimating the expression".

Here lies our first contribution: by employing the **estimand-agnostic (EA)** approach [13], based on Structural Causal Models (SCMs), any causal effect required by do-SHAP can be estimated from a single model following a general procedure instead of query-specific estimands, thereby enabling the computation of do-SVs in a general, practical way (sections 3.2 and 4.1). Secondly, we propose the **Frontier-Reducibility Algorithm** (FRA), which substantially reduces the number of coalitions that need to be evaluated. Although FRA retains the exponential complexity of exact do-SHAP, it delivers significant speed-ups at virtually no additional cost (section 4.2). Thirdly, we devise a do-SHAP explainability strategy, not only for accessible ML systems, but also for natural, ***inaccessible* Data Generating Processes** (section 4.3). We validate the estimation capabilities of the EA approach on do-SVs, demonstrate the speedup of FRA, and showcase these techniques on two real-world datasets to illustrate the power of do-SHAP explanations (section 5 and appendix F). Finally, we address the limitations of our approach in section 6 and finish with our conclusions in section 7.

## 2   Related work

Among many explainability techniques (see the survey in [5]), we are particularly interested in feature attributions, particularly Shapley values [14]. There is a vast literature on SHAP, discussing the choice of the value function $\nu$ and estimation strategies for the SV (e.g., permutation sampling, adaptive sampling or model-specific strategies); refer to [7] for an extensive survey on the topic. Our main focus is in SHAP approaches that leverage the underlying causal structure of the data, operating from Pearl's Structural Causal Models perspective [15]. Asymmetric Shapley values [8] employ a topological order of the graph to restrict which permutations are considered in the computation of conditional-SHAP, thereby granting more attribution to ancestors of other features. Causal Shapley values [9] properly considers the impact of causal interventions on Shapley attributions, but assumes a partial causal ordering of the graph in order to avoid dealing with latent confounders. do-SHAP [12] does require a full causal graph, but provides a full method to compute attribution on all variables, as long as an estimand can be found for every causal query. Finally, in a different direction, Shapley flow [16] studies causal attributions on the causal graph's edges instead of its nodes/variables.

In order to avoid do-SHAP's limitations, we propose EA methods [13], which train a SCM modelling the data distribution to estimate causal queries from it. This approach is explored in the Neural Causal Models framework [17]. In this line, recent contributions employ Deep Learning for SCM modeling: CausalGAN [18] uses Generative Adversarial Networks [19] to model images in an SCM containing descriptive factors of the image; Parafita & Vitrià [20] propose the Distributional Causal Node as a way to model mixed-type distributions (i.e., with discrete and continuous random variables) and expand their framework with Deep Causal Graphs [13]; Pawlowski *et al*. [21] propose Normalizing Flows (NFs) [22] and Variational AutoEncoders [23] for SCMs with medical image nodes; and Diffusion-based Causal Models [24] use Diffusion Models [25] to train their SCMs on high-dimensional data. A promising alternative models SCMs not node by node as all previous works, but the graph as a whole with a single function of its noise signals, thereby avoiding error propagation when sampling. Two of these approaches are VACA [26], which uses Graph Neural Networks [27], and Causal Normalizing Flows [28], with a single NF for the whole graph.

## 3   Preliminaries

This section establishes the concepts and notations needed throughout this work. We start with Structural Causal Models (SCMs), a transparent and concise framework to define the causal assumptions of a data distribution, followed by a discussion on how (identifiable) causal queries can be estimated through SCMs, even in the presence of latent confounders. We then define the general Shapley value, from which we can derive do-SHAP.

**Notation** Sets are represented by boldface letters ($\mathbf{X}$) and their elements by simple letters ($X \in \mathbf{X}$), unless clearly distinguishable. Let the power set of $\mathbf{X}$ be denoted by $\mathbb{P}(\mathbf{X}) := \{\varnothing \subseteq \mathbf{S} \subseteq \mathbf{X}\}$, $[K] := \{1, \ldots, K\}$ an index set, $\Pi(\mathbf{S})$ the set of permutations of elements of $\mathbf{S}$ and $<_\pi$ the order entailed by $\pi$ (e.g., $3 <_\pi 2$ in $\pi = (3, 1, 2)$). Given a graph $\mathcal{G} = (\mathbf{V}, \mathbf{E})$ and a subset of vertices $\mathbf{X} \subseteq \mathbf{V}$, $An(\mathbf{X})$ denotes the set of ancestors of $\mathbf{X}$ (including $\mathbf{X}$) and $De(\mathbf{X})$ the set of descendants (including $\mathbf{X}$). For a certain node $X \in \mathbf{V}$, let $Pa_X$ denote the set of parents of $X$ (not including $X$). Random variables (r.v.s) are denoted in uppercase ($X$) with realizations in lowercase ($x$). Let $x \sim \mathcal{P}(X)$ denote the generation of a new sample $x$ from the distribution $\mathcal{P}(X)$.

## 3.1 Structural Causal Models

Let $\mathcal{M} = (\mathcal{V}, \mathcal{W}, \mathcal{P}, \mathcal{F})$ be a Structural Causal Model (SCM), consisting of a set of *measured* r.v.s $\mathcal{V} = (V_1, \ldots, V_K)$, a set of *latent* r.v.s $\mathcal{W}$, their *priors* $\mathcal{P}(\mathcal{W}) = \prod_{W \in \mathcal{W}} \mathcal{P}(W)$ (all mutually independent), and a set of *functions* $\mathcal{F} := \{f_k\}_{k \in [K]}$ for each measured variable. The set of latent variables consists of $\mathcal{W} := \mathcal{E} \cup \mathcal{U}$, with $\mathcal{E} := (E_1, \ldots, E_K)$ the *exogenous noise signals*, $E_k$ corresponding to $V_k$, and $\mathcal{U}$ a set of *confounders* $U_{\{k,l\}}$ affecting distinct $\mathcal{V}$-nodes $V_k$ and $V_l$.[1] Finally, each $f_k \in \mathcal{F}$ is a deterministic function $V_k = f_k(Pa_k, \mathcal{U}_{\{k,\cdot\}}, E_k)$ that returns $V_k$'s realizations given its measured parents $Pa_k \subseteq \mathcal{V} \setminus \{V_k\}$, confounders $\mathcal{U}_{\{k,\cdot\}} := \{U_{\{k,l\}} \in \mathcal{U} \mid l \in [K]\}$ and the corresponding $E_k$. Let $Pa'_k := Pa_k \cup \mathcal{U}_{\{k,\cdot\}}$.

Let $\mathcal{G}_{\mathcal{M}} = (\mathbf{V}, \mathbf{E})$ be the *directed graph* associated to $\mathcal{M}$, with vertices, nodes or variables $\mathbf{V} := \mathcal{V} \cup \mathcal{W}$ and edges $\mathbf{E} := \{X \to V_k \mid \forall V_k \in \mathcal{V}, X \in Pa'_k \cup \{E_k\}\}$.[2] If $\mathcal{G}_{\mathcal{M}}$ is a *Directed Acyclic Graph* (DAG), there is a *topological order*[3] for $\mathcal{V}$. In that case, we can define $\mathcal{M}$'s probability distribution $\mathcal{P}_{\mathcal{M}}(\mathcal{V})$ from the $SCM$'s *sampling procedure*: it starts by sampling any latent variable $E_X \in \mathcal{E}, U \in \mathcal{U}$ from their priors $\varepsilon_X \sim \mathcal{P}(E_X), u \sim \mathcal{P}(U)$; then, following the topological order $k = 1..K$, it samples every $V_k \in \mathcal{V}$ by applying $v_k = f_k(pa_k, u_{\{k,\cdot\}}, \varepsilon_k)$.

We define an intervention $do(X = x)$, also denoted $\widehat{x}$, on a variable $X \in \mathcal{V}$ as the replacement of $f_x$ with the assignment $X \leftarrow x$. $X$ takes this value independently of its parents, but any descendants may be affected by this change. Note that this transforms the SCM $\mathcal{M}$ into an *intervened model* $\mathcal{M}_x$, *graph* $\mathcal{G}_x := \mathcal{G}_{\mathcal{M}_x}$ (without any edges pointing to $X$), and *distribution* $\mathcal{P}_x(\mathcal{V}) := \mathcal{P}_{\mathcal{M}_x}(\mathcal{V})$. We can also define interventions on sets of variables $do(\mathbf{X} = \mathbf{x})$ by the replacement of each of the corresponding functions $\{f_X \mid X \in \mathbf{X}\}$. The terms $P_{\mathbf{x}}(Y) = P(Y \mid do(\mathbf{X} = \mathbf{x})) = P(Y \mid \widehat{\mathbf{x}})$ are used interchangeably, for clarity or economy of notation depending on the case.

## 3.2 Identifiability and the estimand-agnostic approach

Given r.v.s $\mathcal{V}$ and an i.i.d. dataset $\mathcal{D} = (v^{(i)})_{i \in [N]} \sim \mathcal{P}(\mathcal{V})$ sampled from an unknown *Data Generating Process* (DGP) with a strictly positive probability measure $\mathcal{P}(\mathcal{V})$, assume that $\mathcal{P}(\mathcal{V})$ follows an unknown SCM $\mathcal{M}$, but whose graph $\mathcal{G}_{\mathcal{M}}$ is known. For example, in fig. 1, $\mathcal{V} = (A, E, S, Y)$ and $\mathcal{U} = \varnothing$. Let us estimate the *causal query* $\mathcal{Q} := \mathbb{E}_Y[Y \mid \widehat{e}]$ by transforming it into an observational formula through the rules of do-calculus [15] (see appendix C). At the end of this process, we obtain the final formula, the *estimand*: in the example, $\mathcal{Q} = \mathbb{E}_Y[Y \mid \widehat{e}] = \mathbb{E}_A[\mathbb{E}_Y[Y \mid e, A]]$. If such a formula exists, the query is said to be (non-parametrically) *identifiable* in $\mathcal{G}_{\mathcal{M}}$. Fortunately, there are algorithms to automatically determine identifiability and obtain the corresponding estimand [29, 30], implemented in R [31] and Python [32].

The EB approach employs ML models to approximate each of the probabilistic terms in an estimand; in the example, we can train a model for $f(E, A) \approx \mathbb{E}_Y[Y \mid E, A]$ and then estimate the formula with Monte Carlo for the final estimation. However, this approach does not scale, since, for each and every query, we need to 1) derive the corresponding estimand for that query; 2) train ML models to estimate each term in the formula; and 3) put it all together to arrive at an answer for the query. Even with algorithms to automatically extract the estimand, it is not trivial to compute these formulas, especially if we need to answer exponentially many queries, as will be the case for do-SHAP.

---

[1]The case when $\mathcal{U}$ is empty, i.e., no latent confounders, is known as the *causal sufficiency assumption*.

[2]When drawing $\mathcal{G}_{\mathcal{M}}$, we usually omit $\mathcal{W}$: $\mathcal{E}$ is implicit, and any confounders $U_{\{k,l\}}$ are denoted by $V_k \leftrightarrow V_l$.

[3]We say $\mathcal{V} = (V_1, \ldots, V_K)$ is in a *topological order* of the DAG $\mathcal{G}$ if $\forall k, l \in [K], V_k \in An(V_l) \Rightarrow k \leq l$. Let $<_{\mathcal{G}}$ represent the particular order defined by $\mathcal{G}$: $X <_{\mathcal{G}} Y$.

However, if we had access to the original SCM $\mathcal{M}$, we could simply take $N$ i.i.d. samples from the intervened distribution, $(y^{(i)})_{i \in [N]} \sim \mathcal{P}_e(Y)$, using $\mathcal{M}_e$'s sampling procedure. Regrettably, we rarely have access to the underlying DGP. Instead, let us consider a family of proxy SCMs $\mathcal{M}_\Theta = (\mathcal{V}, \mathcal{W}, \mathcal{P}', \mathcal{F}_\Theta)$ following the same graph $\mathcal{G}_{\mathcal{M}_\Theta} = \mathcal{G}_\mathcal{M}$ and whose $\mathcal{F}_\Theta$ depends on a set of parameters $\Theta$ (e.g., an untrained neural network with parameter space $\Theta$). Irrespective of our choice of prior $\mathcal{P}'$ and functions $\mathcal{F}_\Theta$, if both are expressive enough, we can train $\mathcal{M}_\Theta$ to find a value $\theta$ so that it models the data distribution exactly, $\mathcal{P}_{\mathcal{M}_\theta}(\mathcal{V}) = \mathcal{P}(\mathcal{V})$ (in an infinite data setting). Then, by the application of the sampling procedure on the intervened learned model, we could estimate the query through the proxy SCM procedure, without ever using the estimand. If the query was identifiable in $\mathcal{G}$, the result is guaranteed to be the same as if we had used the original SCM $\mathcal{M}$.

It is trivial to see why[4]: since our identifiable query $Q$'s value is derived from the observational formula of the estimand, it depends exclusively on observational terms resulting from the joint distribution $\mathcal{P}_M(\mathcal{V})$, which we assume is correctly represented by our trained proxy $\mathcal{M}_\theta$. Therefore, as long as we derive its value from the distribution entailed by the proxy, we will necessarily arrive at the same result as with $\mathcal{M}$; otherwise, $\mathcal{P}_{\mathcal{M}_\theta}(\mathcal{V}) \neq \mathcal{P}_\mathcal{M}(\mathcal{V})$. In other words, even though its latent priors and functional assignments may be different, we can still compute the causal query through the proxy SCM *because* there is an estimand for $\mathcal{Q}$ in $\mathcal{G}_\mathcal{M}$. Hence, this results in an alternative approach for causal query estimation, the EA approach [13]: define a trainable SCM $\mathcal{M}'$ with the underlying SCM's graph $\mathcal{G}$, train it to learn the observational distribution $\mathcal{P}_\mathcal{M}(\mathcal{V})$, and compute any identifiable queries from that single model using the SCM's procedures, not the specific estimand for each query. This will become essential for the computation of do-Shapley values.

### 3.3 The Shapley value

Consider a set of $K$ players $\mathbf{X}$ and a *value function* $\nu : \mathbb{P}(\mathbf{X}) \to \mathbb{R}$. Let $\Delta_\nu(\mathbf{S}) := \nu(\mathbf{S}) - \nu(\varnothing)$ be the corresponding *coalitional* (cooperative) *game*. We define the Shapley value [33] $\phi_{\Delta_\nu}(X)$ for a player $X \in \mathbf{X}$, denoted by $\phi_\nu(X)$ or simply $\phi_X$ unless when leading to ambiguity, as:

$$\phi_X := \sum_{\mathbf{S} \subseteq \mathbf{X} \setminus \{X\}} \frac{1}{K} \binom{K-1}{|\mathbf{S}|}^{-1} (\nu(\mathbf{S} \cup \{X\}) - \nu(\mathbf{S})) \tag{1}$$

$$= \frac{1}{K!} \sum_{\pi \in \Pi(\mathbf{X})} (\nu(\mathbf{X}_{\leq_\pi X}) - \nu(\mathbf{X}_{<_\pi X})), \tag{2}$$

where $\mathbf{X}_{<_\pi X} := \{X' \in \mathbf{X} \mid X' <_\pi X\}$, and equivalently for $\mathbf{X}_{\leq_\pi X}$ and $\leq_\pi$. Both equations are equivalent given that the sum over weighted subsets $\mathbf{S}$ results from the average over all permutations of the set of players $\mathbf{X}$. Note that SVs fulfill *efficiency*: $\sum_{X \in \mathbf{X}} \phi_X = \nu(\mathbf{X}) - \nu(\varnothing) = \Delta_\nu(\mathbf{X})$ (i.e., SHAP *attributions* add up to the contributions of the whole set $\mathbf{X}$).

### 3.4 Shapley value estimation

Even though eq. (2) requires $2 \cdot K!$ computations of $\nu$, we can consider each permutation $\pi \in \Pi([K])$ as a sample from the uniform distribution over the set of permutations, $\pi \sim \mathcal{U}(\Pi([K]))$, resulting in $\phi_X = \mathbb{E}_{\pi \sim \mathcal{U}(\Pi([K]))} [\nu(\mathbf{X}_{\leq_\pi X}) - \nu(\mathbf{X}_{<_\pi X})]$, which can be approximated by Monte Carlo, sampling $N$ i.i.d. permutations and averaging their results [34]. Quasi-random and adaptive sampling strategies can also be employed for faster convergence; please refer to [6] for more details.

On the other hand, both methods result in a significant number of subset collisions, making it worthwhile to cache the $\nu(\mathbf{S})$ values to avoid unnecessary computations. We derive the expected number of coalitions sampled after $N$ permutations in appendix B, which let us define a computational budget (i.e., how many permutations to sample) based on desired coalition coverage.

## 4 Method

In the following, we present our contributions: we propose EA techniques to make do-SHAP practical on complex graphs; we present the Frontier-Reducibility Algorithm (FRA), an efficient procedure to

---

[4]Please refer to [17], Corollary 2, for a formal proof. In their terms, our SCMs are $\mathcal{G}$-constrained, $L_1$-consistent Neural Causal Models that can effectively estimate interventional queries (such as do-SHAP's $\nu(\mathbf{S})$ coalition values) as long as they are identifiable by using the mutilation process (see appendix H.4).

significantly reduce the number of coalitions to evaluate during do-SHAP (regardless of using EB or EA methods); finally, we present a theorem allowing for do-Shapley explanations on inaccessible DGPs. Please refer to appendix H for a detailed example illustrating the application of our approach from start to finish.

Please note that throughout this work, we do not claim polynomial-time tractability of do-SHAP. Any discussion of computational *efficiency* refers solely to FRA in isolation, and should not be interpreted as implying polynomial-time performance for do-SHAP.

## 4.1 The do-Shapley value

Consider an SCM $\mathcal{M} = (\mathcal{V}, \mathcal{W}, \mathcal{P}, \mathcal{F})$, a target r.v. $Y \in \mathcal{V}$, a subset of $K$ variables $\mathbf{X} \subseteq \mathcal{V} \setminus \{Y\}$ and a certain sample $\mathbf{x} \sim \mathcal{P}(\mathbf{X})$ we wish to explain. Given a coalition $\mathbf{S} \in \mathbb{P}(\mathbf{X})$ with realizations $\mathbf{s}$ (a subset of $\mathbf{x}$), let us define the *value function* $\nu_{\mathbf{x}}(\mathbf{S})$:

$$\nu_{\mathbf{x}}(\mathbf{S}) := \mathbb{E}\left[Y \mid do(\mathbf{S} = \mathbf{s})\right] \tag{3}$$

Then, the do-Shapley value (do-SV) [12] over variables $\mathbf{X}$ with realizations $\mathbf{x} \sim \mathcal{P}(\mathbf{X})$ on a variable $X \in \mathbf{X}$ is $\phi_X := \phi_{\nu_{\mathbf{x}}}(X)$. For economy of notation, we will simply write $\nu := \nu_{\mathbf{x}}$.

**Assumption 4.1.** Let us assume an unknown SCM $\mathcal{M}$ but with known DAG $\mathcal{G}_{\mathcal{M}}$[5]. Further assume its do-SVs are *identifiable* in $\mathcal{G}_{\mathcal{M}}$ (i.e., $\nu(\mathbf{S})$ is identifiable[6] $\forall \mathbf{S} \subseteq \mathbf{X}$) and that the resulting $\mathcal{P}(\mathcal{V})$ is strictly positive. Note that $\mathcal{G}_{\mathcal{M}}$ may include latent confounders as long as its do-SVs are identifiable.

Jung *et al.* [12] employed the EB approach, using an estimand for each term $\nu(\mathbf{S})$. This makes do-SHAP impractical, requiring different ML models for the estimand's terms and an ad-hoc computation following the formula. In response, we propose to use the **EA approach**: 1) train a single SCM to learn $\mathcal{P}(\mathcal{V})$; 2) for each query $\nu(\mathbf{S})$, determine if it is identifiable (as we do in the EB approach); and 3) use general SCM-based procedures, not the estimand, to estimate the query. An illustrative example of this SCM strategy is provided in appendix H; for further details, please refer to [18, 21, 13, 28]).

## 4.2 Efficient estimation of the do-Shapley value

In this section, we derive a novel algorithm to accelerate do-SHAP. We leave all proofs and the more efficient version of the algorithm to appendix D.

**Proposition 4.2.** *For any non-ancestor $X$ of $Y$, $\phi_X = 0$.*

Consequently, we can restrict $\mathcal{G}$ to $Y$'s ancestors, since every other do-SV will necessarily be 0.

**Assumption 4.3.** Given an SCM $\mathcal{M} = (\mathcal{V}, \mathcal{W}, \mathcal{P}, \mathcal{F})$ and a target r.v. $Y \in \mathcal{V}$, we assume $\mathcal{M}$ to be the projected SCM $\mathcal{M}[An(Y)]$ (see theorem C.8) and simply denote it $\mathcal{M}$. From now on, $\mathcal{V} = \mathbf{X} \cup \{Y\}$ with $\mathbf{X} := An(Y) \setminus \{Y\} = (V_0, \ldots, V_{K-1})$ in a topological order. Let $Y := V_K$.

Let us introduce the concept of *frontiers*, with which we will reduce coalitions.

**Definition 4.4.** Given any node $X \in \mathbf{X}$, a subset $\mathbf{S} \subseteq \mathbf{X}$ is a *frontier* between $X$ and $Y$ if $X \notin \mathbf{S}$ and all directed paths $p = (X, \ldots, Y)$ from $X$ to $Y$ are blocked by $\mathbf{S}$, i.e., $\exists Z \in \mathbf{S}$ s.t. $Z \in p$. We denote the set of frontiers between $X$ and $Y$ in $\mathcal{G}$ as $Fr_{\mathcal{G}}(X, Y)$.

**Proposition 4.5.** *Given $X \in \mathbf{X}$ and $Y$, and a subset $\mathbf{S} \in Fr_{\mathcal{G}}(X, Y)$, then $\nu(\mathbf{S} \cup \{X\}) = \nu(\mathbf{S})$.*

*Remark* 4.6. For any parent $X \in Pa_Y$, no subset $\mathbf{S} \subseteq \mathbf{X} \setminus \{X\}$ is a frontier between $X$ and $Y$.

Finally, using these properties, let us reduce any given coalition $\mathbf{S}$ into its *irreducible subset* $\mathbf{S}' \subseteq \mathbf{S}$, i.e., $\nu(\mathbf{S}) = \nu(\mathbf{S}')$ and $\forall X \in \mathbf{S}', \mathbf{S}' \setminus \{X\} \notin Fr_{\mathcal{G}}(X, Y)$. While an alternative definition exists (see theorem D.10), the following form motivates an algorithm to efficiently reduce coalitions.

---

[5]This is a standard assumption in the SCM community. If the graph is not known, Causal Discovery algorithms [35] can derive a potential graph, later refined with domain expertise and/or randomized experiments.

[6]Running the identifiability algorithms (section 3.2) on all $2^{|\mathbf{X}|}$ terms a priori is unnecessary. Instead, when using the approximation method discussed in section 3.4, we can test identifiability for each new sampled query, caching results for repeated coalitions. If any coalition is found to be non-identifiable during this process, an error state should halt do-SHAP immediately; otherwise, if no non-identifiable coalition is found, our do-SV estimation will be valid. Moreover, certain graph structures (e.g., no latent confounders) make do-SVs trivially identifiable; a general graphical criterion for do-SV identifiability is left for future work.

**Algorithm 1** Frontier-Reducibility Algorithm (FRA) – set version

**Require:** $\mathbf{S} \subseteq \mathbf{X}$, coalition.
**Require:** Fr, a map: tuple[int] $\rightarrow$ bool.
**Require:** $\mathcal{G}$, causal graph.
1: **procedure** FRA($\mathbf{S}$, Fr; $\mathcal{G}$)
2:   SORT($\mathbf{S}, <_{\mathcal{G}}$)
3:   $\mathbf{P} \leftarrow \varnothing$
4:   $\mathbf{Z} \leftarrow \varnothing$
5:   $k \leftarrow |\mathbf{S}|$
6:   **while** $k > 0$ **do**
7:     $X \leftarrow \mathbf{S}[k]$
8:     **if** $X \notin Pa_{\mathcal{G}}(Y)$ **then**
9:       $\mathbf{P'} \leftarrow \mathbf{P} \cap De_{\mathcal{G}}(X)$
10:      $\mathbf{T} \leftarrow (\mathbf{P'} \setminus \mathbf{Z}) \cup \{X\}$
11:      **if** $\mathbf{T} \notin$ Fr **then**
12:        $\mathbf{C} \leftarrow \{X\}$
13:        **while** $\mathbf{C} \neq \varnothing$ and $Y \notin \mathbf{C}$ **do**
14:          $\mathbf{P'} \leftarrow \mathbf{P'} \cup \mathbf{C}$
15:          $\mathbf{C} \leftarrow \bigcup_{C \in \mathbf{C}} Ch_{\mathcal{G}}(C) \setminus \mathbf{P'}$
16:        **end while**
17:        Fr[$\mathbf{T}$] $\leftarrow (\mathbf{C} = \varnothing)$
18:      **end if**
19:      **if** Fr[$\mathbf{T}$] **then**
20:        $\mathbf{Z} \leftarrow \mathbf{Z} \cup \{X\}$
21:      **end if**
22:    **end if**
23:    $\mathbf{P} \leftarrow \mathbf{P} \cup \{X\}$
24:    $k \leftarrow k - 1$
25:  **end while**
26:  **return** $\mathbf{S} \setminus \mathbf{Z}$
27: **end procedure**

**Theorem 4.7.** *Given a topological order $<_{\mathcal{G}}$ in $\mathcal{G}$ and $\mathbf{S} \subseteq \mathbf{X}$, let $\mathbf{Z} := \{X \in \mathbf{S} \mid \mathbf{S}_{>_{\mathcal{G}} X} \in Fr_{\mathcal{G}}(X,Y)\}$, with $\mathbf{S}_{>_{\mathcal{G}} X} := \{Z \in \mathbf{S} \mid Z >_{\mathcal{G}} X\}$. Then $\nu(\mathbf{S}) = \nu(\mathbf{S} \setminus \mathbf{Z})$, and $\mathbf{S} \setminus \mathbf{Z}$ is irreducible.*

Thanks to this theorem, we can derive the irreducible subset of any coalition. Then, we evaluate its $\nu$-value with our SCM and store it in a cache. If, during the do-SHAP calculation, we encounter another subset $\mathbf{S'}$ with the same irreducible subset as the previous $\mathbf{S}$, we can employ the cached $\nu$-value of that irreducible subset, thereby reducing the number of coalitions to evaluate.

From theorem 4.7, we derive the **Frontier-Reducibility Algorithm (FRA)** in algorithm 1, which determines the irreducible subset of any coalition $\mathbf{S}$ efficiently. We start by sorting the coalition in the order defined by $\mathcal{G}$ (line 2) and then iterate through the nodes descendingly (lines 5–6, 24). Given a step $k$ and the corresponding node $X$ (line 7), $\mathbf{P}$ contains the nodes following $X$ in $\mathbf{S}$ (line 23). If $X$ is a parent of $Y$, it trivially has no frontiers, so it cannot be removed (line 8). Otherwise, if $\mathbf{P}$ blocks all paths from $X$ to $Y$ (lines 12–17), it is a frontier for $X$, which means $X$ must be included in $\mathbf{Z}$ (line 20), the set of superfluous nodes.

Let us exemplify with fig. 2, where we try to reduce $\mathbf{S} := \{A, C, E, F\}$. For instance, on $k = 1$, $X = A$, and $\mathbf{P} = \{C, E, F\}$. $A$ is not a parent of $Y$, so we need to determine if $\mathbf{P} \in Fr_{\mathcal{G}}(A, Y)$. For that, we move through every path from $X$ to $Y$, and if no path reaches $Y$ without being interrupted by a node in $\mathbf{P}$, then $\mathbf{P}$ is a frontier between $X$ and $Y$, and $X$ can be removed by adding it to $\mathbf{Z}$. Note that we use the $Fr$ map to cache these computations (lines 9–11, 17) using a reduced version of $\mathbf{P}$, since $\mathbf{P} \in Fr_{\mathcal{G}}(A, Y)$ iif $(\mathbf{P} \cap De_{\mathcal{G}}(X)) \setminus \mathbf{Z} \in Fr_{\mathcal{G}}(A, Y)$. $Fr$ stores whether the key $\mathbf{T}$ can be reduced by its first (sorted) element $X$. Please refer to appendix D.2.2 for an in-depth explanation of FRA and for algorithm 3, an alternative, more efficient version using integer-encoded coalitions.

To finish this subsection, let us draw parallels to Luther *et al.*'s work [36], in which, for conditional-SHAP, pairs $\nu(\mathbf{S} \cup \{X\}) = \nu(\mathbf{S})$ were identified to be skipped, since they resulted in a null differential in SHAP's formula. We move further by: 1) extending the method to do-SHAP, which requires the use of do-calculus to derive these properties; and 2) by describing and efficiently computing the irreducible subset $\mathbf{S'}$ such that $\nu(\mathbf{S}) = \nu(\mathbf{S'})$, instead of only removing a single node.

This results in a greater speed-up. As an example, consider the chain graph $A \rightarrow B \rightarrow Y$. For $\phi_A$, the pairs are $\nu(A) \neq \nu(\varnothing)$ and $\nu(A, B) = \nu(B)$, while for $\phi_B$, $\nu(B) \neq \nu(\varnothing)$ and $\nu(A, B) \neq \nu(A)$. FRA would only need to compute the irreducible sets $\varnothing, \{A\}$, and $\{B\}$, since $\nu(A, B) = \nu(B)$, while Luther's method needs to evaluate every single coalition eventually, even if $\nu(A, B) = \nu(B)$ could be skipped while computing $\phi_A$. Now, let us demonstrate this behavior in general:

**Proposition 4.8.** *Under assumption 4.3, $\forall \mathbf{S} \subseteq \mathcal{X}, \exists Z \in \mathcal{X}$ s.t. at least one of the following is true:*

- $Z \notin \mathbf{S}$ and $\mathbf{S} \notin Fr_{\mathcal{G}}(Z, Y)$.

- $Z \in \mathbf{S}$ and $\mathbf{S} \setminus \{Z\} \notin Fr_{\mathcal{G}}(Z, Y)$.

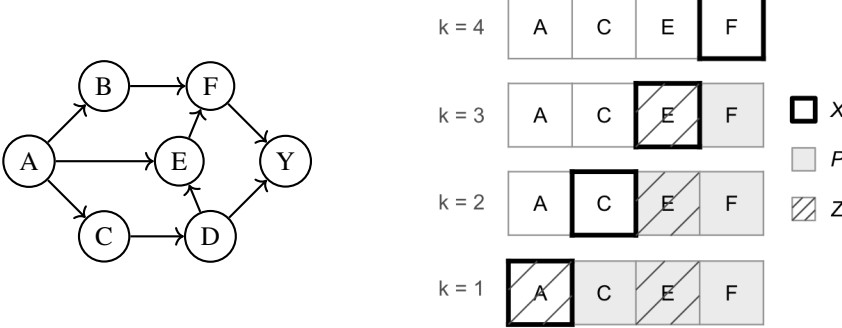

Figure 2: FRA execution example. a) Causal Graph with nodes in alphabetical order representing the selected topological order. b) FRA execution steps, with $k$ representing the loop step (lines 5–6, 24), $X$ the current node (line 7), $\mathbf{P}$ the potential frontier for $X$, and $\mathbf{Z}$ storing the nodes to be removed from $\mathbf{S}$. The result of this execution is the coalition reduction $\{A, C, E, F\} \rightarrow \{C, F\}$.

Consequently, for any pair $\nu(\mathbf{S} \cup \{X\}) = \nu(\mathbf{S})$ identified by Luther, we know that $\exists Z \in \mathbf{X}$ s.t. somewhere else in the (exact) do-SV computation, $\mathbf{S}$ will reappear and cannot be skipped: either $Z \notin \mathbf{S}, \nu(\mathbf{S} \cup \{Z\}) \neq \nu(\mathbf{S})$ or $Z \in \mathbf{S}, \nu(\mathbf{S}) \neq \nu(\mathbf{S} \setminus \{Z\})$ (non-parametrically). By the same argument, the same can be said about $\nu(\mathbf{S} \cup \{X\})$. Therefore, Luther cannot omit these new pairs and will eventually evaluate all $\nu(\mathbf{S})$, even if some were omitted in previous pairs.

Now consider FRA. If we encountered a pair $\nu(\mathbf{S} \cup \{X\}) = \nu(\mathbf{S})$, both must necessarily result in the same irreducible subset, thereby can be skipped without evaluating them. However, even if we later encounter them in a non-skippable pair, their irreducible subsets can be evaluated once and cached for subsequent reappearances. In other words, with Luther's strategy, all $2^K$ coalitions will need to be evaluated eventually, regardless if they were omitted previously in another pair. In contrast, by determining the irreducible subsets and storing their values in a cache, less evaluations will be necessary. Naturally, in the case of SHAP's approximate method, we may not re-encounter every coalition in an unskippable pair, but even in that case, our method need not compute the skippable pair either. Therefore, FRA can only improve upon Luther's speed-up.

### 4.3 do-Shapley explanations

So far, we have been talking about do-SHAP values *w.r.t.* a variable $Y \in \mathcal{V}$ in a certain SCM $\mathcal{M}$, but there are two use cases to consider in practice: either we want to explain a ML model that models $Y$ given some inputs $\mathbf{X'} \subseteq \mathcal{V} \setminus \{Y\}$ or we want to explain the original variable $Y$ directly.

If we want to explain a ML model $f(\mathbf{X'}) := \mathbb{E}[Y \mid \mathbf{X'}]$, $Y$ is replaced by $Y' := f(\mathbf{X'})$ (with no $E_Y$, since $f$ is deterministic); we then work on the projected SCM $\mathcal{M}[An(Y')]$ with $Pa_{Y'} = \mathbf{X'}$. Note that this subgraph may contain variables other than those in $\mathbf{X'}$, since any $X \in \mathbf{X} \setminus \mathbf{X'}$ would be an ancestor to an $X' \in \mathbf{X'}$, and therefore an ancestor of $Y$. With this SCM, we can apply EA procedures to estimate do-SHAP. We exemplify this case in the experiment in section 5.1.

If, instead, we wanted to explain $Y$ directly, we simply employ do-SHAP on a proxy SCM, but note that for a particular $(\mathbf{x}, y) \sim \mathcal{P}(\mathbf{X}, Y)$, $\sum_{X \in \mathbf{X}} \phi_{\nu_\mathbf{x}}(X) = \mathbb{E}[Y \mid \widehat{\mathbf{x}}] - \mathbb{E}[Y] \neq y - \mathbb{E}[Y]$ (unless $Y$ is a constant distribution). There is a *gap* between the contribution of $\mathbf{X}$ ($\Delta_{\nu_\mathbf{x}}(\mathbf{X})$) and the actual value of $Y$, because our particular $\nu$, an interventional query, is essentially a population estimate, and as such aggregates for the whole distribution. In order to explain a particular outcome, we need some kind of *counterfactual* value function $\nu$; this is a promising avenue of research, but is left for future work, since it is beyond the scope of this paper. As an alternative approach, the following theorem proves that, under additional assumptions, we can explain this gap through $E_Y$'s do-SV contribution.

**Theorem 4.9.** *do-Shapley Value for the Noise.*
*Assume that $\mathcal{U}_{\{Y, \cdot\}} = \varnothing$ and that $f_Y \in \mathcal{F}$ follows an additive noise model, i.e., $Y = f(Pa_Y) + E_Y$, for a certain $f$. Consider the do-Shapley game with players $\mathbf{X} \cup \{E_Y\}$; then, $\phi_{E_Y} = y - \mathbb{E}[Y \mid pa_Y]$, while $\phi_X$ for $X \in \mathbf{X}$ can be computed w.r.t. $\mathbf{X}$ only. Furthermore, $\sum_{X \in \mathbf{X}} \phi_X + \phi_{E_X} = y - \mathbb{E}[Y]$.*

Consequently, assuming an unconfounded $Y$ with additive noise, we can explain inaccessible DGPs with attribution to the noise and no computational overhead. In practice, we define a ML model $f'(pa_Y) \approx \mathbb{E}[Y \mid pa_Y]$ and explain it instead, computing the do-SV for $E_Y$ as $\phi_{E_Y} := y - f'(pa_Y)$.

## 5 Experiments

This section contains the empirical validation of our approach. We begin with a synthetic DGP, from which we can derive ground truth do-SVs, to measure estimation error on several EA methods. Secondly, we demonstrate the speedup resulting from the FRA-cache with an ablation test. Finally, we showcase do-SHAP explanations on two real world datasets, left to appendix F due to space restrictions. Please refer to the Supplementary Material for the code of these experiments.

### 5.1 Estimation performance

The goal of this experiment is to evaluate if a proxy SCM can correctly estimate (identifiable) do-SVs without access to the true underlying DGP. Let us design first a synthetic $\mathcal{M}_0$ with variables $\mathcal{V}$, following the graph in fig. 3. We train a ML model $f(pa_X) \approx \mathbb{E}[Y \mid pa_Y]$, and create an SCM $\mathcal{M}$ using the same structural equations as $\mathcal{M}_0$ for $\mathbf{X} := \mathcal{V} \setminus \{Y\}$ but with $Y$ replaced by $Y' := f(Pa_Y)$. This represents our SCM of study. We can consider two cases, both identifiable, by assuming $U_{\{X,B\}}$ is observed (Markovian) or latent (semi-Markovian).

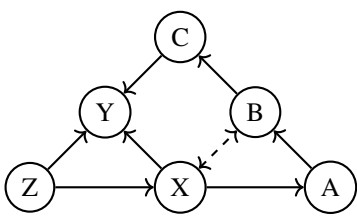

Figure 3: Semi-Markovian graph. The Markovian graph results from considering $U_{\{X,B\}}$ as measured.

We replicate the experiment 30 times with different seeds. Let $\mathcal{D}$ be a dataset generated from $\mathcal{M}$ with $N = 1000$ i.i.d. samples. With access to $\mathcal{M}$, we can estimate each query $\nu(\mathbf{S})$ with $M$ i.i.d. samples from the intervened SCM, passing them through $f$ and averaging the outputs; we will use the corresponding do-SVs $\Phi := (\phi_X^{(i)})_{i \in [N], X \in \mathbf{X}}$ as *ground truth*. We also train several proxy SCMs (with $Y'$ replacing $Y$) to learn $\mathcal{P}(\mathcal{V})$ and then estimate the do-Shapley values $\tilde{\Phi} = (\tilde{\phi}_X^{(i)})_{i \in [N], X \in \mathbf{X}}$, finally computing their **SHAP** estimation **loss** $\mathcal{L}_2(\Phi, \tilde{\Phi}) := \frac{1}{N|\mathbf{X}|} \sum_{i=1}^{N} \sum_{k=1}^{|\mathbf{X}|} (\phi_{X_k}^{(i)} - \tilde{\phi}_{X_k}^{(i)})^2$. We will also compare against a marginal-SHAP estimator (which should result in different values). We compute the average test log-likelihood (**loglk**) for each model as a way to measure distribution adjustment. Finally, for all $X \in \mathbf{X}$, we compute their Feature Importance (**FI**), defined as $FI_X := \frac{1}{N} \sum_{i \in [N]} |\phi_X^{(i)}| / \sum_{X' \in \mathbf{X}} |\phi_{X'}^{(i)}|$.

We will test do-SHAP with several SCM architectures[7]; for further implementation details and a justification of our choices please refer to section E.1.1. These methods are: 1) a **linear** SCM with Normal distributions for each variable, used as a baseline; 2) the Distributional Causal Node [20] (**DCN**), with every node modeled after a pre-defined distribution; and 3) Deep Causal Graph [13] (**DCG**) powered with Normalizing Flows. Additionally, in order to test the graph-as-a-whole approach (see section 2), we opt for Causal Normalizing Flows (**CNF**) [28].

See fig. 4 for the Markovian case. As expected, better distributional adjustment to $\mathcal{P}(\mathcal{V})$ (loglk) correlates with better estimates of do-SVs. Linear-SCM cannot properly model $\mathcal{P}(\mathcal{V})$, and so its do-SHAP performance suffers; DCN comes close to the best models, probably because of the synthetic nature of the data; DCGs and CNFs exhibit similar performance except for more variance on DCGs, possibly due to CNFs modeling all variables at once. Finally, marginal-SHAP significantly differs from the do-SHAP ground truth, showing that, evidently, do-SHAP and marginal-SHAP measure different attributions. FI comparisons *w.r.t.* ground truth values are also aligned with the previous results. We leave the semi-Markovian experiment for section E.1.2, with equivalent conclusions, even in presence of a latent confounder; DCGs display the best estimation performance. In the following, we will employ DCGs even if CNFs seem to be more stable variance-wise, because DCGs admit latent confounders and were orders of magnitude faster in our experiments.

---

[7]None of these methods are external baselines, as do-SHAP has not yet been tested with EA approaches and remains underexplored overall (see [37] for an example), largely due to the ad hoc nature of EB methods. Additionally, EB strategies are excluded from our experiments because they require manual adaptation for each coalition, whereas EA is generalizable and automatable, rendering a direct comparison inappropriate.

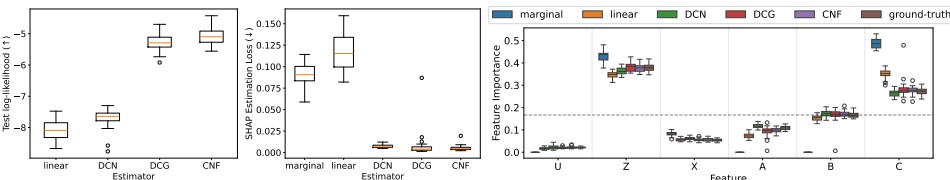

Figure 4: Markovian case. Box-plots computed over 30 realizations of the dataset. (a) Distribution adjustment score, log-likelihood (bigger is better). (b) SHAP estimation loss, $\mathcal{L}$ (lower is better). (c) Feature Importance (the closer to *ground-truth*, the better). Dashed horizontal line represents uniform importance ($\frac{1}{K}$). See section E.1 for a bigger figure.

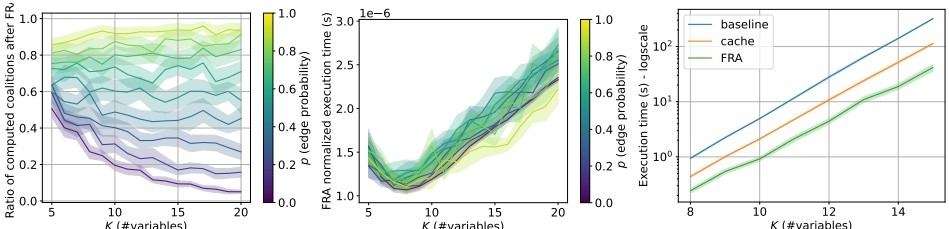

Figure 5: FRA experiments. (a) Ratio of computed coalitions after FRA. (b) FRA execution time per coalition. (c) do-SHAP execution time (logarithmic scale) without cache (*baseline*), with cache (*cache*) and with an FRA cache (*FRA*). Error bars at 2-sigma over 30 replications.

## 5.2 Frontier-Reducibility Algorithm

We now test the computational impact of FRA. Let us consider $\mathcal{G}_{K,p}$, the class of graphs $\mathcal{G}$ with $K+1$ nodes, defined in topological order, $\mathbf{X} := (V_0, \ldots, V_{K-1}), Y := V_K$, where $p \in (0, 1)$ represents the probability of any possible edge $V_i \to V_j$, $0 \le i < j \le K$ appearing in $\mathcal{G}$, and such that $\mathcal{G}$ fulfills two conditions: 1) $An(Y) = \mathbf{X} \cup \{Y\}$ and 2) $Pa_Y \subsetneq \mathbf{X}$. [8] We sample uniformly from $\mathcal{G}_{K,p}$ using rejection sampling to ensure both conditions. Figure 5 shows the results of our experiments, with error bars for the mean of each metric at 2-sigma over 30 random graphs per configuration.

Let $K \in \{5, \ldots, 20\}$ and $p \in \{0.1, \ldots, 0.9\}$. Figure 5 (a) shows the average ratio of coalitions (out of $|\mathbb{P}(\mathbf{X})| = 2^K$) that need to be passed through $\nu$ after reduction by FRA. Note that for higher values of $K$, each $p$-curve approaches $p$, i.e., in the case of $p = 0.1$, a reduction in $\nu$-computations of 90%. Figure 5 (b) shows the average execution time of FRA per coalition. Despite the exponentially-larger number of directed paths in the graph, the computation of FRA appears to grow linearly with $K$, due to the fact that it scales with the size of the coalition $\mathbf{S}$ to be evaluated and the depth of the graph, both at most $K$. For $K \le 20$, the error bars do not exceed $3\mu s$ per FRA call.

Finally, we evaluate FRA with an ablation test in fig. 5 (c). We design synthetic DGPs for random $\mathcal{G} \in \mathcal{G}_{K,p}$ with $\forall X \in \mathcal{V}, f_X(pa_X, \varepsilon_X) := \text{mean}(pa_X) + \varepsilon_X, \varepsilon_X \sim \mathcal{N}(0, 1)$. We choose a linear SCM for its fast execution; real-world SCMs, with far more complex architectures, will require even longer to execute, so FRA will have an even stronger impact. We evaluate do-SHAP with a linear DCG and using the approximate method with $N$ permutations such that at least half the total number of coalitions are expected to have been processed after $N$ permutations;[9] this value for $N$ is computed using eq. (6) in appendix B. We restrict this experiment to $K \ge 8$ so that $N \ge 30$ and set $p = 0.25$. We can now compare the mean execution time when evaluating every coalition $\mathbf{S}$ (**baseline**), when employing a **cache** to avoid repetitions in $\nu$-computations, and when employing an FRA-cache (**fra**) to reduce and cache coalitions. As a result, we can see a consistent pattern: FRA is an order of magnitude faster than the baseline and twice faster than the cache.

---

[8]Note that when $Pa_Y = \mathbf{X}$, FRA has no effect (see theorem 4.6); since this case is of no interest for this experiment, such cases are discarded. We discuss this limitation to the speed-up power of FRA in section 6.

[9]This particular choice for $N$ results in an exponential time-growth *w.r.t.* $K$, but this is not necessarily the case in the usual setting, where we choose $N$ based on estimation variance or computational limitations.

Note that it is not prohibitive to run our method for higher values of $K$; we repeat the experiment for $K = 100$ without replications to explain a sample with 1000 permutations. The DCG training time amounts to 16s and the do-SHAP executions for *no-cache*, *cache*, and *FRA-cache* lasted for 26m01s, 25m20s and 21m14s, respectively. The average error of estimation is $2.71e-4$. There is still an exponential trend in computation-time (we make no claim of reducing the exponential nature of SHAP) but: 1) it is possible to compute do-SVs on high $K$ with an automatized, single-model approach, in contrast with EB approaches that would require manual specification of procedures based on estimands, rendering them infeasible in practice; and 2) we achieve a considerable improvement time-wise (4m *w.r.t. cache*) at a negligible cost in running FRA (8s in total).

Please refer to section E.2 for the full experimental setup and further tests on FRA. There we show that FRA's execution time is negligible *w.r.t.* the computation of $\nu(\mathbf{S})$, even on linear SCMs. This difference can only increase with more complex SCM architectures; therefore, for virtually no cost, FRA skips computing $\nu(\mathbf{S})$ up to a significant factor, resulting in a marked speedup for do-SHAP.

## 6 Limitations

We devote this section to discussing the limitations of this work.

Firstly, we assume to know the causal graph $\mathcal{G}$; while Causal Discovery algorithms and/or domain experts can help to define it, our techniques are sensitive to graph misspecification. Regrettably, there are no doubly-robust guarantees for EA techniques yet, which is a definite disadvantage *w.r.t.* EB alternatives. Nevertheless, the ad-hoc nature of EB methods makes them impractical for do-SHAP, with or without doubly-robust guarantees, while our approach can adapt to complex graphs. In any case, this issue falls out of scope for this paper, which is focused on facilitating and accelerating do-SV estimation. An additional limitation for some EA methods is error propagation: how misalignment in modeling $P(\mathcal{V})$ can propagate to do-SV estimations. However, some works in the literature address this issue by modeling all variables in a single-pass $\mathcal{F}$, avoiding compounding errors [26, 28].

Secondly, our approach requires modeling the data distribution by training an appropriate SCM, which can take some (offline) time. However, once the SCM is trained, any new sample can be explained (online) from that same model. SCMs can typically be trained in the order of minutes (half an hour for the most complex graphs/distributions in this work, figs. 10 and 12 in the appendix), and after that offline training, SHAP execution (online) becomes the most pressing factor time-wise. The FRA algorithm helps in significantly accelerating this step, as demonstrated in section 5.2, but this speed-up only results when not all variables are parents of $Y$; otherwise, no coalition would be reducible. Fortunately, real-world DGPs rarely have all (proper) $Y$-ancestors as parents, and in ML systems, defining all $\mathbf{X}$ as model inputs ($Pa_{Y'}$) is hardly advisable, since they may contain non-ancestors of $Y$ (leading to spurious correlations or anti-causal effects [38]) or inputs $\mathbf{A} \subseteq \mathbf{X} \setminus Pa_Y$ that are blocked by $Pa_Y$, ($Y \perp\!\!\!\perp \mathbf{A} \mid Pa_Y$), in which case their inclusion could lead to overfitting and adversarial vulnerability. Consequently, feature selection strategies should aim at discarding these cases, which paves the way for FRA speed-ups.

## 7 Conclusion

In this work, we have introduced a practical method to estimate do-SVs on arbitrarily complex graphs by using the estimand-agnostic approach, with which we can estimate any identifiable query using general procedures agnostic to the query's estimand. This flexibility is essential to make these techniques accessible to practitioners, who may not necessarily be experts in Causal Inference. We have tested our approach on multiple SCM architectures, illustrating the relationship between distribution modeling and do-SHAP estimation performance. We have proposed the novel Frontier-Reducibility Algorithm, which speeds up do-SHAP significantly at virtually no cost. Finally, we have tested the capabilities of our method and applied it on two real-world datasets (see appendix F), showcasing do-SHAP's explanatory power, either for ML models or inaccessible DGPs.

Further work could propose new SCM architectures to better model the data distribution, overcoming some of the identified limitations, along with more efficient estimators, ideally with doubly-robust guarantees. A general graphical criterion for do-SV identifiability is also a worthwhile new direction. Finally, do-SVs are based on interventional queries, but these are inherently population measures; counterfactual value functions $\nu$ could result in a promising new kind of causal, local explanations.

## Acknowledgments and Disclosure of Funding

The research leading to these results has received funding from the AI4Science fellowship within the "Generacion D" initiative, Red.es, Ministerio para la Transformación Digital y de la Función Pública, for talent attraction (C005/24-ED CV1), funded by the European Union NextGenerationEU funds, through PRTR. It also was funded by the Horizon Europe Programme under the AI4DEBUNK Project (https://www.ai4debunk.eu), grant agreement num. 101135757. Additionally, this work has been partially supported by JDC2022-050313-I funded by MCIN/AEI/10.13039/501100011033al by the European Union NextGenerationEU/PRTR. We also gratefully acknowledge Novartis for sponsoring Tomàs Garriga's industrial PhD, and the Government of Catalonia's Industrial PhDs Plan for funding part of this research.

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

# A    Shapley value axioms

The SV $\phi = \{\phi_X\}_{X \in \mathbf{X}}$ is the unique attribution measure fulfilling a number of desirable properties:

- **Efficiency**: $\sum_{X \in \mathbf{X}} \phi_X = \nu(\mathbf{X}) - \nu(\varnothing) = \Delta(\mathbf{X})$; the sum of SVs adds up to the total contribution of $\mathbf{X}$.

- **Missingness**: if $\forall \mathbf{S} \subseteq \mathbf{X} \setminus \{X\}$, $\nu(\mathbf{S} \cup \{X\}) = \nu(\mathbf{S})$, then $\phi_X = 0$; players with no contribution to any coalition have Shapley value 0.

- **Symmetry**: if $\forall \mathbf{S} \subseteq \mathbf{X} \setminus \{X, Y\}$, $\nu(\mathbf{S} \cup \{X\}) = \nu(\mathbf{S} \cup \{Y\})$, then $\phi_X = \phi_Y$; players with identical contribution to any coalition have identical Shapley values.

- **Linearity**: if $\forall \mathbf{S} \subseteq \mathbf{X}, \nu(\mathbf{S}) := \nu_1(\mathbf{S}) + \nu_2(\mathbf{S})$, then $\forall X \in \mathbf{X}, \phi_\nu(X) = \phi_{\nu_1}(X) + \phi_{\nu_2}(X)$, and if $\forall \mathbf{S} \subseteq \mathbf{X}, \nu(\mathbf{S}) := a \cdot \nu'(\mathbf{S})$, then $\forall X \in \mathbf{X}, \phi_\nu(X) = a \cdot \phi_{\nu'}(X)$; $\phi$ is linear *w.r.t.* the coalition game $\nu$.

# B    Cache impact on the approximation algorithm

Consider the approximation method (see section 3.4), where we sample permutations of $K$ elements uniformly with replacement, $\pi \sim \mathcal{U}(\Pi([K]))$, so as to approximate the Shapley value with a Monte Carlo estimator. In this section, we want to evaluate how much we can accelerate the computation of new permutations as we fill a cache with the values of previously computed coalitions. When we use a cache, once we compute a coalition for the first time, we save its result in it (assuming no cache limit) and further computations of this coalition will incur in negligible computation time (simply a cache access), therefore speeding up the computation of new permutations. We want to measure exactly how much we can speed up the process.

Let us define some notation. Given $\pi \in \Pi([K])$, let us denote by $\mathcal{C}(\pi)$ the set of $K + 1$ coalitions $\mathbf{S} \in \mathbb{P}([K])$ defined by taking the first $s$ elements of $\pi$, $s = 0..K$ (e.g., for $\pi = (3, 1, 2)$, $\mathcal{C}(\pi) = \{\varnothing, (3), (3, 1), (3, 1, 2)\}$). Then, for an arbitrary $\mathbf{S} \in \mathbb{P}([K])$ and a permutation $\pi \sim \mathcal{U}(\Pi([K]))$:

$$P(\mathbf{S} \in \mathcal{C}(\pi)) = \frac{|\mathbf{S}|!(K - |\mathbf{S}|)!}{K!} = \binom{K}{|\mathbf{S}|}^{-1} \tag{4}$$

since $\mathbf{S}$ must appear at the beginning of $\pi$ in an arbitrary order, so there is $|\mathbf{S}|!$ possibilities, with the remaining $(K - |\mathbf{S}|)$ elements in an arbitrary order, so $(K - |\mathbf{S}|)!$, out of the total $K!$ possible permutations. Since we are taking $N$ i. i. d. permutations $(\pi^{(n)})_{n \in [N]}$, it follows that

$$P(\forall n \in [N], \mathbf{S} \notin \mathcal{C}(\pi^{(n)})) = \left(1 - \binom{K}{|\mathbf{S}|}^{-1}\right)^N, \tag{5}$$

which is the probability of an arbitrary coalition $\mathbf{S}$ not belonging to any of the $N$ previously sampled permutations, and therefore, it still needs to be computed when it appears in a future permutation. In particular, note that we do not need to know the elements of $\mathbf{S}$, only its cardinality $|\mathbf{S}|$, which we will denote by $s := |\mathbf{S}|$. Given the set of $K + 1$ coalitions $\mathcal{C}(\pi^{(N)})$ in permutation $\pi^{(N)}$, we can now compute the expected ratio of its coalitions not found in any of the previous permutations (therefore not cached); in other words, the expected ratio of computations we need to perform at the $N$-th permutation, $N > 1$, is:

$$\frac{1}{K+1} \sum_{\mathbf{S} \in \mathcal{C}(\pi^{(N)})} P(\forall n \in [N-1], \mathbf{S} \notin \mathcal{C}(\pi^{(n)})) = \frac{1}{K+1} \sum_{s=0}^{K} \left(1 - \binom{K}{s}^{-1}\right)^{N-1}. \tag{6}$$

For $N = 1$, the ratio is trivially 1. Morover, for $s = 0$ ($\mathbf{S} = \varnothing$) and $s = K$ ($\mathbf{S} = [K]$), the term $\left(1 - \binom{K}{s}^{-1}\right)$ becomes 0 (it is impossible not to have seen them in a previous permutation, since they are in every permutation), so we omit these cases in the following sums.

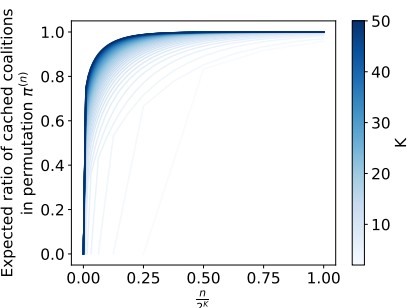
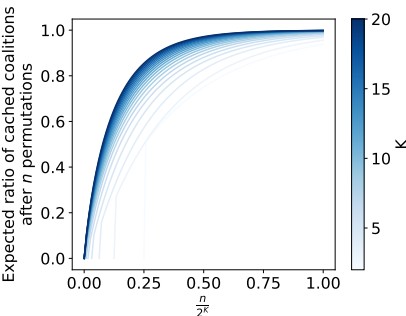

Figure 6: Cache evolution plots. a) Ratio of coalitions in $\pi^{(n)}$ already cached. b) Ratio of total coalitions already cached after $n$ permutations. Both x-axis represent the number of permutations $n$ divided by $2^K$, so as to compare between different values of $K$.

Finally, the expected ratio of cached coalitions (out of the total number of coalitions $2^K$) after $N \geq 1$ permutations is:

$$
\begin{aligned}
\frac{1}{2^K} \sum_{n=1}^{N} \sum_{\mathbf{S} \in \mathcal{C}(\pi^{(n)})} P(\forall n' \in [n-1], \mathbf{S} \notin \mathcal{C}(\pi^{(n')})) &= \frac{K+1}{2^K} + \frac{1}{2^K} \sum_{n=2}^{N} \sum_{s=1}^{K-1} \left(1 - \binom{K}{s}^{-1}\right)^{n-1} \\
&= \frac{K+1}{2^K} + \frac{1}{2^K} \sum_{s=1}^{K-1} \binom{K}{s}\left(1 - \binom{K}{s}^{-1}\right)\left(1 - \left(1 - \binom{K}{s}^{-1}\right)^{N-1}\right) \\
&= \frac{K+1}{2^K} + \frac{1}{2^K} \sum_{s=1}^{K-1} \left(\binom{K}{s} - 1\right) - \frac{1}{2^K} \sum_{s=1}^{K-1} \binom{K}{s}\left(1 - \binom{K}{s}^{-1}\right)^{N} \\
&= 1 - \frac{1}{2^K} \sum_{s=0}^{K} \binom{K}{s}\left(1 - \binom{K}{s}^{-1}\right)^{N},
\end{aligned}
\tag{7}
$$

where we first split the sum over $n$ for $n = 1$ and $n > 1$, and then swap the sums and apply, for $x := 1 - \binom{K}{s}^{-1}$, the equality $\sum_{n=1}^{N} x^n = x\frac{1-x^N}{1-x}$ for $x \in (0,1)$ (which is the case when $s \neq 0, K$), and noting that $\frac{1}{1-x} = \binom{K}{s}$. We then split the sum in two terms, with the first half adding up to 1 with $\frac{K+1}{2^K}$. The rest of the transformation is trivial.

We now plot eqs. (6) and (7) in fig. 6 (a) and (b), respectively, for several values of $K$ (represented by color opacity). The x-axis in both cases is $\frac{n}{2^K}$, so as to show how each curve progresses as $n \to 2^K$, where we will have encountered $(K+1)2^K$ coalitions. We can see that the likelihood of encountering previously-computed coalitions is very high early in the process, which means that the computations required per permutation speed up significantly in the early stages. However, if we wanted to cover the totality of possible coalitions with the permutation method, this would require many more coalitions given the high likelihood of collision; this is relevant particularly for section 5.2, where we set $N$ to be high enough so that at least half the total number of coalitions are expected to have been processed after $N$ permutations.

These plots are merely illustrative; we encourage researchers to make use of the derived equations to adjust for the appropriate number of permutations in terms of computation time budget.

## C  Causal Inference concepts

We include here some additional notation and concepts for Causal Inference, necessary for the proofs in appendix D.

**Notation**

Given r.v.s $X \neq Y$ and a disjoint set of r.v.s $\mathbf{Z}$ (possibly empty), we denote that $X$ is independent of $Y$ conditioned on $\mathbf{Z}$ in a distribution $\mathcal{P}$ by $(X \perp\!\!\!\perp Y \mid \mathbf{Z})_{\mathcal{P}}$. Given disjoint sets of r.v.s $\mathbf{X}, \mathbf{Y}, \mathbf{Z}$, we say that $\mathbf{X}$ is independent of $\mathbf{Y}$ given $\mathbf{Z}$ in a distribution $\mathcal{P}$, denoted by $(\mathbf{X} \perp\!\!\!\perp \mathbf{Y} \mid \mathbf{Z})_{\mathcal{P}}$, if and only if $\forall X \in \mathbf{X}, \forall Y \in \mathbf{Y}, (X \perp\!\!\!\perp Y \mid \mathbf{Z})_{\mathcal{P}}$. $\mathcal{P}$ can be omitted unless it leads to ambiguity.

Given $\mathbf{X}, \mathbf{Y} \subseteq \mathbf{V}$, let $\mathcal{G}_{\overline{\mathbf{X}}\underline{\mathbf{Y}}}$ denote the graph $\mathcal{G}$ modified such that all edges pointing towards nodes in $\mathbf{X}$ are removed (overline) and all edges starting from nodes in $\mathbf{Y}$ are removed (underline). We may incur in abuse of notation (e.g., $\mathcal{G}_{\overline{\mathbf{X}Y}} := \mathcal{G}_{\overline{\mathbf{X} \cup \{Y\}}}$) unless it leads to ambiguity.

## C.1  d-separability and do-calculus

In the following, we will define the concept of $d$-separability, its connection to independence, and the three rules of $do$**-calculus**. Please refer to [15] for more details.

**Definition C.1.** $d$**-separability**.
Given a DAG $\mathcal{G} = (\mathbf{V}, \mathbf{E})$, a path $p$ is $d$-*separated* (blocked) by a set $\mathbf{Z} \subseteq \mathbf{V}$ (possibly empty) if and only if either is true:

1. $p$ contains a *chain* $A \to B \to C$ or a *fork* $A \leftarrow B \to C$ such that $B$ is in $\mathbf{Z}$.

2. $p$ contains a *collider* $A \to B \leftarrow C$ such that no descendant of $B$ (including $B$) is in $\mathbf{Z}$.

Given disjoint sets $\mathbf{X}, \mathbf{Y}, \mathbf{Z} \subseteq \mathbf{V}$, we say that $\mathbf{Z}$ *d-separates $\mathbf{X}$ from $\mathbf{Y}$* in $\mathcal{G}$ if $\mathbf{Z}$ $d$-separates every path $p$ from a node $X \in \mathbf{X}$ to a node $Y \in \mathbf{Y}$. We denote this by $(\mathbf{X} \perp\!\!\!\perp \mathbf{Y} \mid \mathbf{Z})_{\mathcal{G}}$.

**Definition C.2.** **Markov Compatibility**.
We say that a distribution $\mathcal{P}(\mathcal{V})$ on a set of variables $\mathcal{V} = (V_1, \cdots, V_K)$ is (Markov) *compatible* with a DAG $\mathcal{G}$ with $\mathcal{V}$ as vertices in $\mathcal{G}$ if $P(\mathcal{V}) = \prod_{k \in [K]} \mathcal{P}(V_k \mid Pa_{\mathcal{G}}(V_k))$.

**Theorem C.3.** *Independence and $d$-separability*.
*Given an SCM $\mathcal{M} = (\mathcal{V}, \mathcal{W}, \mathcal{P}, \mathcal{F})$ compatible with a DAG $\mathcal{G}_{\mathcal{M}}$ and disjoint sets $\mathbf{X}, \mathbf{Y}, \mathbf{Z} \subseteq \mathcal{V}$, if $(\mathbf{X} \perp\!\!\!\perp \mathbf{Y} \mid \mathbf{Z})_{\mathcal{G}_{\mathcal{M}}}$ then $(\mathbf{X} \perp\!\!\!\perp \mathbf{Y} \mid \mathbf{Z})_{\mathcal{P}}$. Conversely, if $(\mathbf{X} \not\perp\!\!\!\perp \mathbf{Y} \mid \mathbf{Z})_{\mathcal{G}_{\mathcal{M}}}$, there exists at least one distribution $\mathcal{P}'$ compatible with $\mathcal{G}_{\mathcal{M}}$ (in fact, almost all) such that $(\mathbf{X} \not\perp\!\!\!\perp \mathbf{Y} \mid \mathbf{Z})_{\mathcal{P}'}$.*

*Remark* C.4. The second statement comes from the fact that precise parameter choices $\theta$ of distributions $\mathcal{P}_{\Theta}$ might result in independence in an otherwise unblocked path in $\mathcal{G}$. Fortunately, such specific tuning of $\Theta$ rarely occurs in practice.

*Remark* C.5. If we need to determine independence relationships $(\mathbf{X} \perp\!\!\!\perp \mathbf{Y} \mid \mathbf{Z})_{\mathcal{P}}$ ($\mathbf{Z}$ possibly empty), we simply verify that all paths connecting $\mathbf{X}$ and $\mathbf{Y}$ are blocked by $\mathbf{Z}$, using $d$-separability.

Next, we introduce the three rules of $do$-calculus, with which we can transform causal queries step by step, until we reach the desired estimand.

**Theorem C.6.** *Rules of $do$-calculus*.
*Given an SCM $\mathcal{M} = (\mathcal{V}, \mathcal{W}, \mathcal{P}, \mathcal{F})$ compatible with a DAG $\mathcal{G}_{\mathcal{M}}$, for any disjoint sets $\mathbf{X}, \mathbf{Y}, \mathbf{Z}, \mathbf{W}, \subseteq \mathcal{V}$ ($\mathbf{X}$ and $\mathbf{W}$ possibly empty):*

1. ***Insertion/deletion of observations (R1):***
   $P_{\boldsymbol{x}}(\mathbf{Y} \mid \mathbf{Z}, \mathbf{W}) = P_{\boldsymbol{x}}(\mathbf{Y} \mid \mathbf{W})$    *if* $(\mathbf{Y} \perp\!\!\!\perp \mathbf{Z} \mid \mathbf{X}, \mathbf{W})_{\mathcal{G}_{\overline{\mathbf{X}}}}$.

2. ***Exchange of interventions/observations (R2):***
   $P_{\boldsymbol{x},\boldsymbol{z}}(\mathbf{Y} \mid \mathbf{W}) = P_{\boldsymbol{x}}(\mathbf{Y} \mid \boldsymbol{z}, \mathbf{W})$    *if* $(\mathbf{Y} \perp\!\!\!\perp \mathbf{Z} \mid \mathbf{X}, \mathbf{W})_{\mathcal{G}_{\overline{\mathbf{X}}\underline{\mathbf{Z}}}}$.

3. ***Insertion/deletion of interventions (R3):***
   $P_{\boldsymbol{x},\boldsymbol{z}}(\mathbf{Y} \mid \mathbf{W}) = P_{\boldsymbol{x}}(\mathbf{Y} \mid \mathbf{W})$    *if* $(\mathbf{Y} \perp\!\!\!\perp \mathbf{Z} \mid \mathbf{X}, \mathbf{W})_{\mathcal{G}_{\overline{\mathbf{X}\mathbf{Z}(\mathbf{W})}}}$,
   *where $\mathbf{Z}(\mathbf{W}) := \mathbf{Z} \setminus An_{\mathcal{G}_{\overline{\mathbf{X}}}}(\mathbf{W})$, the set of nodes in $\mathbf{Z}$ that are not ancestors of $\mathbf{W}$ (including $\mathbf{W}$) in the graph $\mathcal{G}_{\overline{\mathbf{X}}}$.*

## C.2  Projected Structural Causal Models

**Definition C.7.** **Divergent Path**.
A *divergent path* between $X$ and $Y$ consists of two directed paths, from $W$ to $X$ and from $W'$ to $Y$, such that $W = W'$ or $W \leftrightarrow W'$.

**Definition C.8. Projected SCM**.
Given an SCM $\mathcal{M} = (\mathcal{V}, \mathcal{W}, \mathcal{P}, \mathcal{F})$ compatible with a DAG $\mathcal{G}_\mathcal{M}$ and a subset $\mathcal{V}' \subseteq \mathcal{V}$, we define the *projected causal DAG* $\mathcal{G}[\mathcal{V}']$ defined on vertices $\mathcal{V}'$ and $\mathcal{W}' := \mathcal{E}' \cup \mathcal{W}'$, with $\mathcal{E}' := \{E_X \in \mathcal{E} \mid X \in \mathcal{V}'\}$ and $\mathcal{U}'$ as defined next, such that:

- $\forall V_k, V_l \in \mathcal{V}'$, there is a directed edge $V_k \rightarrow V_l$ if there exists a directed path from $V_k$ to $V_l$ in $\mathcal{G}_\mathcal{M}$ where every internal node in the path is not in $\mathcal{V}'$.

- $\forall V_k, V_l \in \mathcal{V}'$, there is a bidirected edge $V_k \leftrightarrow V_l$ (connected by a latent confounder $U_{\{k,l\}} \in \mathcal{U}'$) if there exists a *divergent* path in $\mathcal{G}$ between them such that every internal node is not in $\mathcal{V}'$.

We define the *projected SCM* $\mathcal{M}[\mathcal{V}']$ by restricting its graph to $\mathcal{G}_\mathcal{M}[\mathcal{V}']$, with distribution $\mathcal{P}_{\mathcal{M}[\mathcal{V}']}(\mathcal{V}') = \mathcal{P}_\mathcal{M}(\mathcal{V}')$.

*Remark* C.9. The projected SCM respects all conditional independence relationships and the rules of *do*-calculus in the original graph. [39].

# D    Proofs

In this section, we will prove the results in the main paper and discuss the Frontier-Reducibility Algorithm.

## D.1    Non-ancestors

**Lemma D.1.** *Given a DAG $\mathcal{G} = (V, E)$ and disjoint subsets of vertices $X, Y \subseteq V$ (possibly empty), if there is a path $p$ in $\mathcal{G}_{\overline{X}\underline{Y}}$, then $p$ is a path in $\mathcal{G}$.*

*Proof.* $\mathcal{G}_{\overline{X}\underline{Y}}$'s edges are a subset of $\mathcal{G}$'s edges, since $\mathcal{G}_{\overline{X}\underline{Y}}$ only removes edges either ending in $\mathbf{X}$ or starting from $\mathbf{Y}$. Adding those edges back in $\mathcal{G}$ cannot remove any edge from the path; hence, $p$ is a path in $\mathcal{G}$. $\qquad\square$

**Proposition D.2. *Non-Ancestors do not Contribute*.**
*Let $\mathcal{M}$ be an SCM $\mathcal{M} = (\mathcal{V}, \mathcal{W}, \mathcal{P}, \mathcal{F})$, $Y$ the target r.v., $X$ a subset $X \subseteq \mathcal{V} \setminus \{Y\}$ and $\mathbf{x}$ a realization $\mathbf{x} \sim \mathcal{P}(\mathbf{X})$. For any $X \in \mathbf{X}$, if $X$ is not an ancestor of $Y$, then $\phi_{\nu_\mathbf{x}}(X) = 0$.*

*Proof.* We will prove that $\forall X \notin An_\mathcal{G}(Y), \forall \mathbf{S} \subseteq \mathbf{X} \setminus \{X\}, \nu(\mathbf{S} \cup \{X\}) = \mathbb{E}\left[Y \mid \widehat{\mathbf{s}}, \widehat{x}\right] = \mathbb{E}\left[Y \mid \widehat{\mathbf{s}}\right] = \nu(\mathbf{S})$. If that is the case, then

$$\phi_X = \sum_{\mathbf{S} \subseteq \mathbf{X} \setminus \{X\}} \frac{1}{K} \binom{K-1}{|\mathbf{S}|}^{-1} (\nu(\mathbf{S} \cup \{X\}) - \nu(\mathbf{S})) = 0.$$

Note that $\mathbb{E}\left[Y \mid \widehat{\mathbf{s}}, \widehat{x}\right] = \mathbb{E}\left[Y \mid \widehat{\mathbf{s}}\right]$ if $P_{\mathbf{s},x}(Y) = P_\mathbf{s}(Y)$, which is implied by R3 if $(Y \perp\!\!\!\perp X \mid \mathbf{S})_{\mathcal{G}_{\overline{\mathbf{S}X}}}$.

Let us prove this independence by contradiction: assume there is a path $p$ connecting $X$ and $Y$ unblocked conditioned on $\mathbf{S}$ in $\mathcal{G}_{\overline{\mathbf{S}X}}$. The path cannot start with $X \leftarrow \cdots$ since all edges pointing towards $X$ are removed in $\mathcal{G}_{\overline{\mathbf{S}X}}$, so $p = X \rightarrow \cdots \overset{?}{-} Y$. Since the path is unblocked, if there were any left arrows ($\leftarrow$) in the path, the resulting collider $\cdots \rightarrow B \leftarrow \cdots$ must necessarily fulfill $De_{\mathcal{G}_{\overline{\mathbf{S}X}}}(B) \in \mathbf{S}$ to unblock the path. There are two cases: 1) if $B \in \mathbf{S}$, then there is an edge $B \leftarrow \cdots$ for a node $B \in \mathbf{S}$, which cannot be true in $\mathcal{G}_{\overline{\mathbf{S}X}}$; 2) if $B \in An_{\mathcal{G}_{\overline{\mathbf{S}X}}}(\mathbf{S}) \setminus \mathbf{S}$, then there is a directed path from $B$ to a node in $\mathbf{S}$, which again cannot happen in $\mathcal{G}_{\overline{\mathbf{S}X}}$ because we have removed all edges pointing towards $\mathbf{S}$. Therefore, the path must necessarily not contain any left arrows, which means that $p$ is a directed path from $X$ to $Y$ in $\mathcal{G}_{\overline{\mathbf{S}X}}$, which must also be a directed path in $\mathcal{G}$ due to theorem D.1; therefore $X \in An_\mathcal{G}(Y)$, contradicting the initial assumption. No unblocked path can exist, which proves $(Y \perp\!\!\!\perp X \mid \mathbf{S})_{\mathcal{G}_{\overline{\mathbf{S}X}}}$ and the theorem in turn. $\qquad\square$

## D.2 Frontier-Reducibility Algorithm

We begin by defining the concept of *frontier* and proving several properties related to it, necessary for the definition of the Frontier-Reducibility Algorithm (FRA), introduced next. We finish with an alternative formulation of the FRA algorithm with integers for faster execution time and lesser memory usage.

### D.2.1 Frontiers and properties

In the following, consider an SCM $\mathcal{M} = (\mathcal{V}, \mathcal{W}, \mathcal{P}, \mathcal{F})$ with associated DAG $\mathcal{G} = (V, E)$, where $\mathcal{V} = (V_0, \ldots, V_K)$ is sorted in an arbitrary topological order of the graph. Let $\mathbf{X} := \{V_0, \ldots, V_{K-1}\}$, $Y := V_K$, and assume that $\mathbf{X} \subseteq An(Y)$. Note that there may be latent confounders ($\mathcal{U} \neq \varnothing$).

**Definition D.3.** Given any node $X \in \mathbf{X}$, a subset $\mathbf{S} \subseteq \mathbf{X}$ is a frontier between $X$ and $Y$ if $X \notin \mathbf{S}$ and all directed paths $p = (X, \ldots, Y)$ from $X$ to $Y$ are blocked by $\mathbf{S}$, i.e., $\exists Z \in \mathbf{S}$ s.t. $Z \in p$. We denote the set of frontiers between $X$ and $Y$ in $\mathcal{G}$ as $Fr_{\mathcal{G}}(X, Y)$.

**Proposition D.4.** *Given nodes $X \in \mathbf{X}$ and $Y$, and a subset $\mathbf{S} \in Fr_{\mathcal{G}}(X, Y)$, frontier from $X$ to $Y$, then $\nu(\mathbf{S} \cup \{X\}) = \nu(\mathbf{S})$.*

*Proof.* We will apply R3 by proving that $(Y \perp\!\!\!\perp X \mid \mathbf{S})_{\mathcal{G}_{\overline{\mathbf{S}X}}}$, in which case

$$\nu(\mathbf{S} \cup \{X\}) = \mathbb{E}\left[Y \mid \widehat{\mathbf{s}}, \widehat{x}\right] = \mathbb{E}\left[Y \mid \widehat{\mathbf{s}}\right] = \nu(\mathbf{S}).$$

Note that in $\mathcal{G}_{\overline{\mathbf{S}X}}$, all paths from $X$ to $Y$ are front-door paths. Consider any such $p = X \to \cdots \overset{?}{-} Y$. If the path is fully directed, since $\mathbf{S}$ is a frontier, $\exists Z \in \mathbf{S}$ s.t. $Z$ is in the path, thereby blocking it. If it is not directed, there exists a collider $\cdots \to Z \leftarrow \cdots$, which also blocks the path: $Z \notin An(\mathbf{S})$, since $\mathbf{S}$ has no ancestors other than itself in $\mathcal{G}_{\overline{\mathbf{S}X}}$ and $Z \notin \mathbf{S}$ because $\cdots \to Z$ is in $p$. Therefore, any path $p$ between $X$ and $Y$ must be blocked by $\mathbf{S}$ in $\mathcal{G}_{\overline{\mathbf{S}X}}$, which proves R3. □

*Remark* D.5. For any parent $X \in Pa_Y$, no subset $\mathbf{S} \subseteq \mathbf{X} \setminus \{X\}$ is a frontier between $X$ and $Y$.

**Proposition D.6.** *Given nodes $X \in \mathbf{X}$ and $Y$, and a frontier $\mathbf{S} \in Fr_{\mathcal{G}}(X, Y)$,*

1. *$\forall \mathbf{S}' \subseteq \mathbf{X} \setminus \{X\}, \mathbf{S}' \supseteq \mathbf{S}$, then $\mathbf{S}' \in Fr_{\mathcal{G}}(X, Y)$.*

2. *$\mathbf{S} \cap De(X) \in Fr_{\mathcal{G}}(X, Y)$.*

*Proof.*

1. Since $\mathbf{S} \in Fr_{\mathcal{G}}(X, Y)$, any directed path $p$ between $X$ and $Y$ is blocked by $\mathbf{S}$; being $\mathbf{S}'$ a superset of $\mathbf{S}$, it must also block all such paths.

2. Any non-descendant of $X$ cannot appear in a directed path from $X$ to $Y$, which means that it is superfluous in the frontier set. As such, $\mathbf{S} \cap De(X) \in Fr_{\mathcal{G}}(X, Y)$.

□

**Corollary D.7.** *Given $X \in \mathbf{X}$ and $\mathbf{S} \subseteq \mathbf{X} \setminus \{X\}$, let $\mathbf{S}_{>_{\mathcal{G}}X} := \{Z \in \mathbf{S} \mid Z >_{\mathcal{G}} X\}$.*

$$\mathbf{S} \in Fr_{\mathcal{G}}(X, Y) \Leftrightarrow \mathbf{S}_{>_{\mathcal{G}}X} \in Fr_{\mathcal{G}}(X, Y) \Leftrightarrow \mathbf{S} \cap De_{\mathcal{G}}(X) \in Fr_{\mathcal{G}}(X, Y). \tag{8}$$

*Proof.* If $\mathbf{S} \in Fr_{\mathcal{G}}(X, Y)$, $\mathbf{S} \cap De(X) \in Fr_{\mathcal{G}}(X, Y)$, and $\mathbf{S}_{>_{\mathcal{G}}X} \supseteq \mathbf{S} \cap De(X)$, since any $Z \in \mathbf{S} \cap De(X)$ fulfills $Z >_{\mathcal{G}} X$, which proves that $\mathbf{S}_{>_{\mathcal{G}}X} \in Fr_{\mathcal{G}}(X, Y)$. On the other hand, if $\mathbf{S}_{>_{\mathcal{G}}X} \in Fr_{\mathcal{G}}(X, Y)$, $\mathbf{S} \in Fr_{\mathcal{G}}(X, Y)$, since $\mathbf{S} \supseteq \mathbf{S}_{>_{\mathcal{G}}X}$. The remaining iff is trivial given theorem D.6. □

**Definition D.8.** A set $\mathbf{S} \subseteq \mathbf{X}$ is Frontier-Reducible (FR) in $\mathcal{G}$ if $\exists X \in \mathbf{S}$ s.t. $\mathbf{S} \setminus \{X\} \in Fr_{\mathcal{G}}(X, Y)$.

In particular, if $\mathbf{S}$ is FR by $X \in \mathbf{S}$, $\nu(\mathbf{S}) = \nu(\mathbf{S} \setminus \{X\})$.

**Theorem D.9.** *Consider any FR $\mathbf{S} \subseteq \mathbf{X}$, and let us define $\mathbf{Z} := \{X \in \mathbf{S} \mid \mathbf{S} \setminus \{X\} \in Fr_{\mathcal{G}}(X, Y)\} = \{X \in \mathbf{S} \mid \mathbf{S}_{>_{\mathcal{G}}X} \in Fr_{\mathcal{G}}(X, Y)\}$. Then $\nu(\mathbf{S}) = \nu(\mathbf{S} \setminus \mathbf{Z})$ and $\mathbf{S} \setminus \mathbf{Z}$ is not FR.*

*Proof.* Consider $\mathbf{Z} = \{Z_{i_1}, \ldots, Z_{i_n}\}$ in the order $<_{\mathcal{G}}$. Firstly, $\nu(\mathbf{S}) = \nu(\mathbf{S} \setminus \{Z_{i_1}\})$ by construction. Now, note that $\mathbf{S}_{>_{\mathcal{G}} Z_{i_j}} \in Fr_{\mathcal{G}}(Z_{i_j}, Y)$ by construction of $\mathbf{Z}$ and theorem D.7, and that $\mathbf{S} \setminus \mathbf{Z}_{\leq_{\mathcal{G}} Z_{i_j}} = \mathbf{S} \setminus \{Z_{i_1}, \ldots, Z_{i_j}\} \supseteq \mathbf{S}_{>_{\mathcal{G}} Z_{i_j}}$, so by Prop. D.6, $\mathbf{S} \setminus \mathbf{Z}_{\leq_{\mathcal{G}} Z_{i_j}} \in Fr_{\mathcal{G}}(Z_{i_j}, Y)$. Then, let us assume that $\forall 1 \leq j < n,\ \nu(\mathbf{S}) = \nu(\mathbf{S} \setminus \mathbf{Z}_{\leq Z_{i_j}})$. Given that $\mathbf{Z}_{\leq Z_{i_j}} \setminus \{Z_{i_{j+1}}\} \in Fr_{\mathcal{G}}(Z_{i_{j+1}}, Y)$, then $\nu(\mathbf{S} \setminus \mathbf{Z}_{\leq Z_{i_{j+1}}}) = \nu(\mathbf{S} \setminus \mathbf{Z}_{\leq Z_{i_j}}) = \nu(\mathbf{S})$, which proves $\nu(\mathbf{S}) = \nu(\mathbf{S} \setminus \mathbf{Z})$ by induction.

Additionally, $\mathbf{S} \setminus \mathbf{Z}$ is not FR since, if $\exists X \in \mathbf{S} \setminus \mathbf{Z}$ s.t. $\mathbf{S} \setminus \mathbf{Z} \setminus \{X\} \in Fr_{\mathcal{G}}(X, Y)$, then $\mathbf{S} \setminus \{X\} \supseteq \mathbf{S} \setminus \mathbf{Z} \setminus \{X\}$ is also a frontier between $X$ and $Y$, which implies that $X \in \mathbf{Z}$. $\qquad\square$

For a clearer characterization of the irreducible set, consider the following proposition.

**Proposition D.10.** *Given a FR $S \subseteq X$ and its corresponding irreducible subset $S' := S \setminus Z$, with $\mathbf{Z} := \{X \in S \mid S_{>_{\mathcal{G}} X} \in Fr_{\mathcal{G}}(X, Y)\}$, then $S' = S \cap An_{\mathcal{G}_{\overline{S}}}(Y)$.*

*Proof.* We will show that $\mathbf{S} \setminus \mathbf{Z} = \mathbf{S} \cap An_{\mathcal{G}_{\overline{\mathbf{s}}}}(Y)$, or equivalently, that $\forall X \in \mathbf{S},\ \mathbf{S}_{>_{\mathcal{G}} X} \notin Fr_{\mathcal{G}}(X, Y)$ iff $X \in An_{\mathcal{G}_{\overline{\mathbf{s}}}}(Y)$. Consider $X \in \mathbf{S}$. If $\mathbf{S}_{>_{\mathcal{G}} X} \notin Fr_{\mathcal{G}}(X, Y)$, then $\mathbf{S} \setminus \{X\} \notin Fr_{\mathcal{G}}(X, Y)$ by theorem D.7, so there is a directed path from $X$ to $Y$ not blocked by $\mathbf{S} \setminus \{X\}$. Consequently, $X$ is an ancestor of $Y$ in the graph where we remove any incoming edges to $\mathbf{S}$; in other words, $X \in An_{\mathcal{G}_{\overline{\mathbf{s}}}}(Y)$. Conversely, if $X \in An_{\mathcal{G}_{\overline{\mathbf{s}}}}(Y)$, there is a directed path from $X$ to $Y$ not blocked by $\mathbf{S} \setminus \{X\}$, therefore $\mathbf{S} \setminus \{X\} \notin Fr_{\mathcal{G}}(X, Y)$ and $\mathbf{S}_{>_{\mathcal{G}} X} \notin Fr_{\mathcal{G}}(X, Y)$, again by theorem D.7. $\qquad\square$

Finally, let us prove the property that explains why the identification of irreducible subsets results in less $\nu$-evaluations than Luther *et al.* 's [36] approach:

**Proposition D.11.** *Under assumption 4.3, $\forall S \subseteq \mathcal{X}, \exists Z \in \mathcal{X}$ s.t. at least one of the following is true:*

1. *$Z \notin S$ and $S \notin Fr_{\mathcal{G}}(Z, Y)$.*

2. *$Z \in S$ and $S \setminus \{Z\} \notin Fr_{\mathcal{G}}(Z, Y)$.*

*Proof.* We will reason by cases:

- If $\mathbf{S} = \mathbf{X}$, given that $\mathbf{X} = An_{\mathcal{G}}(Y) \setminus \{Y\}$, there must be a parent $Z$ of $Y$ in $\mathbf{S}$. A parent $Z \in \mathbf{S}$ of $Y$ can never have a frontier with $Y$, hence $\mathbf{S} \setminus \{Z\} \notin Fr_{\mathcal{G}}(Z, Y)$.

- If $\mathbf{S} = \varnothing$, $\forall Z \in \mathbf{X}, \varnothing \notin Fr_{\mathcal{G}}(Z, Y)$ (otherwise, $Z$ would not be an ancestor of $Y$, contradicting the assumption).

- Let $\mathbf{S} \neq \varnothing, \mathbf{S} \neq \mathcal{X}$:

  - If $\mathbf{S}$ is irreducible, $\forall Z \in \mathbf{S}, \mathbf{S} \setminus \{Z\} \notin Fr(Z, Y)$; otherwise, $\mathbf{S}$ would not be irreducible.
  - If $\mathbf{S}$ is not irreducible, let $\mathbf{S}' \subsetneq \mathbf{S}$ be its irreducible set (as determined by theorem D.9 and proposition D.10). We know that $\mathbf{S}' \neq \varnothing$; otherwise, $\forall Z \in \mathbf{S}, \varnothing \in Fr(Z, Y)$, meaning that $Z$ is not an ancestor of $Y$, contradicting the assumption. Now, $\forall Z \in \mathbf{S}', \mathbf{S} \setminus \{Z\} \notin Fr_{\mathcal{G}}(Z, Y)$; otherwise, by theorem D.9, $Z$ would not belong to the irreducible set of $\mathbf{S}$, which is $\mathbf{S}'$.

Under all possible cases, the result is proven. $\qquad\square$

### D.2.2  Algorithm soundness

Theorem D.9 identifies which elements can be removed from the computation of $\nu(\mathbf{S})$ for any set $\mathbf{S}$. As a result, if we compute and cache $\nu(\mathbf{S} \setminus \mathbf{Z})$, any other set with the same Frontier-Irreducible set can skip the $\nu$ computation and return the cached value instead. Additionally, we do not need to test identifiability for FR sets, only for the corresponding Frontier-Irreducible sets. We now need to define an efficient method to compute $\mathbf{S} \setminus \mathbf{Z}$, the Frontier-Reducibility Algorithm (FRA), described in algorithm 2; let us first demonstrate its soundness.

Given $\mathbf{S} = (X_{i_1}, \ldots, X_{i_n})$ in $<_{\mathcal{G}}$ order, at step $k = n..1$, $X := X_{i_k}$ and $\mathbf{P} := \{X_{i_n}, \ldots, X_{i_{k+1}}\} = \mathbf{S}_{>_{\mathcal{G}} X_{i_k}}$. At this stage, we test if $\mathbf{P} \in Fr_{\mathcal{G}}(X, Y)$, or equivalently, if $(\mathbf{P} \cap De_{\mathcal{G}}(X)) \setminus \mathbf{Z} \in Fr_{\mathcal{G}}(X, Y)$,

**Algorithm 2** Frontier-Reducibility Algorithm (FRA) – set version (with comments).

---

**Require:** $\mathbf{S} \subseteq \mathbf{X}$, coalition.
**Require:** Fr, a map: tuple[int] $\rightarrow$ bool.
**Require:** $\mathcal{G}$, causal graph.

1: **procedure** FRA($\mathbf{S}$, Fr; $\mathcal{G}$)
2:      SORT($\mathbf{S}, <_{\mathcal{G}}$)                                  $\triangleright$ Sort $\mathbf{S}$ topologically
3:      $\mathbf{P} \leftarrow \varnothing$                                     $\triangleright$ Posterior nodes, $\mathbf{S}_{>_{\mathcal{G}} X_{i_k}}$
4:      $\mathbf{Z} \leftarrow \varnothing$                                        $\triangleright$ Superfluous variables
5:      $k \leftarrow |\mathbf{S}|$                                      $\triangleright$ Iterator index, moving backwards
6:      **while** $k > 0$ **do**
7:          $X \leftarrow \mathbf{S}[k]$                                    $\triangleright X = X_{i_k}$
8:          **if** $X \notin Pa_{\mathcal{G}}(Y)$ **then**                $\triangleright$ Parents have no frontiers
9:              $\mathbf{P'} \leftarrow \mathbf{P} \cap De_{\mathcal{G}}(X)$             $\triangleright$ Auxiliary variable for the loop
10:            $\mathbf{T} \leftarrow (\mathbf{P'} \setminus \mathbf{Z}) \cup \{X\}$          $\triangleright$ Cache key for Fr: (node, frontier)
11:            **if** $\mathbf{T} \notin$ Fr **then**        $\triangleright$ Non-cached key, determine if $\mathbf{P'} \in Fr_{\mathcal{G}}(X, Y)$
12:               $\mathbf{C} \leftarrow \{X\}$                  $\triangleright$ Iterate from X to its descendants
13:               **while** $\mathbf{C} \neq \varnothing$ and $Y \notin \mathbf{C}$ **do**      $\triangleright$ While more nodes before $Y$
14:                  $\mathbf{P'} \leftarrow \mathbf{P'} \cup \mathbf{C}$        $\triangleright$ Add currently explored nodes to $\mathbf{P'}$
15:                  $\mathbf{C} \leftarrow \bigcup_{C \in \mathbf{C}} Ch_{\mathcal{G}}(C) \setminus \mathbf{P'}$    $\triangleright$ Filter frontier or previously-explored nodes
16:              **end while**
17:              Fr$[\mathbf{T}] \leftarrow C = \varnothing$        $\triangleright \mathbf{P} \in Fr_{\mathcal{G}}(X, Y) \Leftrightarrow$ we have not reached $Y$
18:            **end if**
19:            **if** Fr$[\mathbf{T}]$ **then**
20:              $\mathbf{Z} \leftarrow \mathbf{Z} \cup \{X\}$                $\triangleright$ Add to superflous nodes
21:            **end if**
22:          **end if**
23:          $\mathbf{P} \leftarrow \mathbf{P} \cup \{X\}$                      $\triangleright$ Update $\mathbf{S}_{>_{\mathcal{G}} X}$
24:          $k \leftarrow k - 1$                            $\triangleright$ Continue the global iteration
25:      **end while**
26:      **return** $\mathbf{S} \setminus \mathbf{Z}$                           $\triangleright$ Remove superfluous nodes
27: **end procedure**

---

in which case we will include it in $\mathbf{Z}$. At the end of the process, $\mathbf{Z} = \{X \in \mathbf{S} \mid \mathbf{S}_{>_{\mathcal{G}} X} \in Fr_{\mathcal{G}}(X, Y)\}$, which, by theorem D.9, means that $\nu(\mathbf{S}) = \nu(\mathbf{S} \setminus \mathbf{Z})$, and $\mathbf{S} \setminus \mathbf{Z}$ is not FR.

We include some optimizations to this algorithm. Firstly, we precompute $Pa_{\mathcal{G}}(Y)$, $(De_{\mathcal{G}}(X))_{X \in \mathbf{X}}$ and $(Ch_{\mathcal{G}}(X))_{X \in \mathbf{X}}$ so that we do not need to traverse the graph every time they are needed. Secondly, we employ the Fr cache, containing whether a certain coalition (a set of integers) is Frontier-Reducible by its first (sorted) element, which is populated as FRA processes more sets $\mathbf{S}$. On the other hand, when storing the results for $\mathbf{P} \in Fr_{\mathcal{G}}(X, Y)$ in the $Fr$ cache, we store instead $\mathbf{T} := ((\mathbf{P} \cap De_{\mathcal{G}}(X)) \setminus \mathbf{Z}) \cup \{X\}$, which is equivalent; this is so that we can better employ the Fr cache, collapsing different $P \cup \{X\}$ sets into reduced keys. However, when testing the frontier in lines 12–17, it will be faster with the larger number of nodes in $\mathbf{P'} := \mathbf{P} \cap De_G(X)$ (including previously removed nodes in $\mathbf{Z}$), since they could cut paths earlier. As a result, for any given set $\mathbf{S} \subseteq \mathbf{X}$, algorithm 2 requires $|\mathbf{S}|$ global iterations (one per element of $\mathbf{S}$), some of them skipped because $X \in Pa_{\mathcal{G}}(Y)$, some already cached in Fr. Finally, this cache can be reused between explanations of do-SHAP for the same graph; only the $\nu$ cache must be reset every time. This speeds up further explanations with virtually zero cost from the FRA.

The next step is how to determine if the set $\mathbf{P'} \subseteq \mathbf{X} \setminus \{X\}$ is a frontier between $X$ and $Y$. Naively, we could check if all directed paths between $X$ and $Y$ are blocked by (intersect with) $\mathbf{P'}$; we could precompute all paths and store them for faster access, but the number of paths grows exponentially (in the worst case scenario, i.e., a complete graph, there are $2^K - 1$ directed paths), which would in turn require an exponential number of iterations per frontier check. Instead, we devise a more efficient method, described in lines 12–17.

We now prove the validity of this procedure. Let us define $\mathbf{C}_0 := \{X\}$. $\mathbf{C}_0$ will never be empty nor contain $Y$, so we always enter the loop. At each while-step $l > 0$, $\mathbf{P'}_l := \bigcup_{l' < l} \mathbf{C}_{l'} \cup \mathbf{P'}$ and $\mathbf{C}_l := \bigcup_{C \in \mathbf{C}_{l-1}} Ch_{\mathcal{G}}(C) \setminus \mathbf{P'}_l$. All directed paths from $X$ to $Y$ are sequences of parent-child pairs,

just as any node $C \in \mathbf{C}_l$ is a child of a certain node $C' \in \mathbf{C}_{l-1}$. Additionally, since every node in $\mathbf{X}$ is an ancestor of $Y$, by exploring these parent-child sequences we will necessarily result in a directed path from $X$ to $Y$. Therefore, every directed path is covered by a sequence of nodes $C_l \in \mathbf{C}_l$ unless they are discarded by $\mathbf{P'}_l$, in which case either $\mathbf{P'}$ blocked the node in the path, or it was a node already visited in a different path before, which would continue with a subpath $C \to \cdots \to Y$ that is currently being explored or has already been discarded.

Note that since $\mathbf{P'}_l$ removes any already-visited nodes from $\mathbf{C}_l$, and we always move one level deeper in the graph, $\mathbf{C}_l$'s nodes are all necessarily at depth $l$ from $X$. Given that the graph $\mathcal{G}$ is finite and acyclic, $\mathbf{C}$ will eventually be empty or contain $Y$ (since it is the last node in any path), which guarantees that the loop ends. Let $\mathbf{C}_n$ denote the last step. Note that if $\mathbf{C}_n \neq \varnothing$, then $Y \in \mathbf{C}_n$, which means that there was a sequence of nodes, each a child of the previous one, that were never filtered by $\mathbf{P'}$; in other words, there exists a directed path from $X$ to $Y$ that is not blocked by $\mathbf{P'}$. Therefore $\mathbf{P'} \notin Fr_{\mathcal{G}}(X, Y)$. On the other hand, if $\mathbf{C}_n = \varnothing$, then every sequence of nodes (every path) was eventually blocked by $\mathbf{P'}$. Therefore, $\mathbf{P'} \in Fr_{\mathcal{G}}(X, Y)$.

In terms of execution time, since every step results in nodes one depth-level deeper, the number of iterations of this procedure cannot be higher than the maximum depth of the graph, which, in the worst case scenario (e.g., a chain graph) is $K - d$, $d$ being the depth of $X$, making it much more efficient than the naive strategy.

### D.2.3 Integer formulation

We can further optimize FRA by transforming set operations into integer and binary operations, resulting in algorithm 3. Let us demonstrate that both algorithms are equivalent. Given $\mathbf{X} = (V_0, \ldots, V_{K-1})$, $K := |\mathbf{X}|$, there is a bijection $\phi : \mathbb{P}(\mathbf{X}) \to \{0, \ldots, 2^K - 1\}$ such that $\phi(\mathbf{S}) = \sum_{V_k \in \mathbf{S}} 2^k$. Note that $\phi(\mathbf{S})$ is a $K$-length binary array with 1s in each position $k$ (starting from the end) such that $V_k \in \mathbf{S}$. Consequently, let us define, for any $s \in \{1, \cdots, 2^K - 1\}$, $\psi(s) := \lfloor \log_2 s \rfloor$ [10]; then $\psi(s) = \max \{k \mid V_k \in \phi^{-1}(s)\}$; if we subtract $2^{\psi(s)}$ from $s$, we can apply $\psi$ again to retrieve the second-largest element, and so on until $s = 0$ ($\mathbf{S} = \varnothing$). The sequence of elements $\psi(s)$ returns the original set $\mathbf{S} = \phi^{-1}(s)$.

Thanks to this bijection, we can "$\phi$-encode" any node or coalition as a unique integer, and we can perform all our operations directly on integers with arithmetic and binary operations, which are less expensive, timing- and memory-wise, than with operations over sequences of integers. Note that, $\forall \mathbf{S}, \mathbf{S'} \subseteq \mathbf{X}$:

1. $\phi(\mathbf{S} \cap \mathbf{S'}) = \phi(\mathbf{S}) \ \& \ \phi(\mathbf{S'})$, with $\&$ the bitwise AND operator.

2. $\phi(\mathbf{S} \cup \mathbf{S'}) = \phi(\mathbf{S}) \mid \phi(\mathbf{S'})$, with $\mid$ the bitwise OR operator.

3. $\phi(\mathbf{S} \setminus \mathbf{S'}) = \phi(\mathbf{S}) \ \& \ \neg \phi(\mathbf{S'})$, with $\neg$ the bitwise NOT operator.

4. $\mathbf{S} \cap \mathbf{S'} = \varnothing \Rightarrow \phi(\mathbf{S} \cup \mathbf{S'}) = \phi(\mathbf{S}) + \phi(\mathbf{S'})$.

5. $\mathbf{S} \supseteq \mathbf{S'} \Rightarrow \phi(\mathbf{S} \setminus \mathbf{S'}) = \phi(\mathbf{S}) - \phi(\mathbf{S'})$.

Let us compare between the set and integer versions of the algorithm. Firstly, we precompute and $\phi$-encode `PaY`, `De` and `Ch`, since they will be used repeatedly throughout the algorithm. Note that we do not need to sort $\mathbf{S}$ beforehand, instead passing it in its $s := \phi(\mathbf{S})$ representation.[11] We can obtain the elements $x$ in descending order, already encoded, by computing $x := 2^{\lfloor \log_2 s \rfloor}$ (line 5) and subtracting it from $s$ (line 28). We can check if an encoded $x$ is a parent of $Y$ with the $\&$ operator (line 6). We can restrict $\mathbf{P}$, encoded by an integer $p$, to $\mathbf{P'} := \mathbf{P} \cap De_{\mathcal{G}}(X)$, encoded by an integer $p'$, by using the $\&$ operator on the precomputed encoded set $\texttt{De}[x] := \phi(De_{\mathcal{G}}(X))$ (line 7). We can set the cache-key $\mathbf{T} := (\mathbf{P'} \setminus \mathbf{Z}) \cup \{X\}$ by its code $(p' \ \& \ \neg z) + x$ (line 8). Finally, in order to determine if $\mathbf{P'}$ is a frontier between $X$ and $Y$, we iterate over the elements in $\mathbf{C}_k$ by employing the same strategy as before (lines 10–21).

---

[10] In practice, for larger values of $K$ (e.g., $K = 100$), we must use bit operations to detect the largest 1-bit in $s$, instead of using the logarithm; these are faster and not subject to numerical error. However, we denote this operation as $\lfloor \log_2 s \rfloor$ throughout the text and the algorithm for clarity and simplicity.

[11] It is more efficient to pass $s$ directly to the procedure, since we can pre-encode all indices $\{0, \ldots, K - 1\}$ before generating permutations of them; then, we just need to pass the sum of the chosen coalition.

---
**Algorithm 3** Frontier-Reducibility Algorithm (FRA) – integer version.

---
**Require:** $s := \phi(\mathbf{S}),\ \mathbf{S} \subseteq \mathbf{X}$.
**Require:** Fr, a map int $\to$ bool.
**Require:** PaY $:= \phi(Pa_{\mathcal{G}}(Y))$.
**Require:** De$[2^k] := \phi(De_{\mathcal{G}}(V_k)), \forall V_k \in \mathbf{X}$.
**Require:** Ch$[2^k] := \phi(Ch_{\mathcal{G}}(V_k)), \forall V_k \in \mathbf{X}$.

```
 1: procedure FRA(s, Fr; PaY, De, Ch)
 2:     p ← 0                                          ▷ Posterior nodes, S_{>_G X_{i_k}}
 3:     z ← 0                                          ▷ Superfluous variables
 4:     while s > 0 do                                 ▷ Iterating through S topologically-backwards
 5:         x ← 2^{⌊log₂ s⌋}                           ▷ X = X_{i_k}
 6:         if x & PaY = 0 then                        ▷ Parents have no frontiers
 7:             p' ← p & De[x]                         ▷ Auxiliary variable for the loop
 8:             t ← (p' & ¬z) + x                      ▷ Cache key for Fr: (node, frontier)
 9:             if t ∉ Fr then                         ▷ Non-cached key, determine if P' ∈ Fr_G(X, Y)
10:                 c ← x                              ▷ Iterate from X to its descendants
11:                 while c ≠ 0 and y & c = 0 do       ▷ While more nodes before Y
12:                     p' ← p' | c                    ▷ Add currently explored nodes to P'
13:                     c' ← c                         ▷ C ← ⋃_{C∈C} Ch_G(C) by iterating
14:                     while c' > 0 do
15:                         x' ← 2^{⌊log₂ c'⌋}         ▷ Get an element X' ∈ C'
16:                         c ← c | Ch[x']             ▷ Add Ch_G(X') to C
17:                         c' ← c' − x'               ▷ Remove X' from C' to continue the iteration
18:                     end while
19:                     c ← c & ¬p'                    ▷ Filter frontier or previously-explored nodes
20:                 end while
21:                 Fr[t] ← c = 0                      ▷ P' ∈ Fr_G(X, Y) ⇔ we have not reached Y
22:             end if
23:             if Fr[t] then
24:                 z ← z + x                          ▷ Add to superflous nodes
25:             end if
26:         end if
27:         p ← p + x                                  ▷ Update S_{>_G X}
28:         s ← s − x                                  ▷ Remove X from S to continue the global iteration
29:     end while
30:     return p − z                                   ▷ Remove superfluous nodes
31: end procedure
```

---

All of these changes result in an equivalent algorithm, more time-efficient (as demonstrated in section E.2) and memory-efficient (since we operate and cache integers rather than tuples of integers).

### D.3 do-Shapley value for the noise

**Theorem D.12.** *do-Shapley Value for the Noise*.
*Given a target r.v. $Y \in \mathcal{V}$, consider the projected SCM $\mathcal{M}[An(Y)]$, with $\mathbf{X} := An(Y) \setminus \{Y\}$ and realizations $(\mathbf{x}, y) \sim \mathcal{P}(\mathbf{X}, Y)$. Let $(\phi_X := \phi_{\nu_x}(X))_{X \in \mathbf{X}}$ be the do-Shapley values associated with $K$ players $\mathbf{X}$.*

*Let us assume that there is no latent confounder connected to $Y$, and that $f_Y \in \mathcal{F}$ follows an additive noise model, i.e., $Y = f_Y(Pa_Y, E_Y) = f(Pa_Y) + E_Y$ for an unknown function $f$. Let $\phi'$ be the do-Shapley values w.r.t. players $\mathbf{X}' := \mathbf{X} \cup \{E_Y\}$; then, for any $X \in \mathbf{X}, \phi'_X = \phi_X$ and $\phi'_{E_Y} = y - \mathbb{E}[Y \mid pa_Y]$. Furthermore, $\sum_{X \in \mathbf{X}} \phi'_X + \phi'_{E_Y} = y - \mathbb{E}[Y]$.*

*Proof.* Let us define some notation for convenience:

- $\forall \mathbf{S} \subseteq \mathbf{X}$, let $\mathbf{S}^c := \mathbf{X} \setminus \mathbf{S}$.

- Note that $Pa_Y \subseteq \mathbf{S} \cup \mathbf{S}^c$; let us denote the selected values $pa_Y$ as the output of a function $Pa_Y(\mathbf{s}, \mathbf{s}^c)$ for ease of exposition.

- Let $\nu'(\mathbf{S}) := \mathbb{E}_{\mathbf{S}^c|\widehat{\mathbf{s}}}[f(Pa_Y(\mathbf{s}, \mathbf{S}^c))]$ for convenience of notation.

We want to compute do-Shapley values $\phi'$ for the $(K+1)$-game (including $E_Y$) with realizations $(\varepsilon_Y, \mathbf{x}, y) \sim \mathcal{P}(E_Y, \mathbf{X}, Y)$ (with $\varepsilon_Y$ latent, unknown) based on the values $\phi$ for the $K$-game (only including $\mathbf{X}$) with the same realizations $(\mathbf{x}, y) \sim \mathcal{P}(\mathbf{X}, Y)$. Let us first determine the value of the following two quantities for any $\mathbf{S} \subseteq \mathbf{X}$ ($E_Y \notin \mathbf{S}$):

$$\nu(\mathbf{S} \cup \{E_Y\}) = \mathbb{E}\left[Y \mid \widehat{\mathbf{s}}, \widehat{\varepsilon_Y}\right] = \mathbb{E}_{\mathbf{S}^c|\widehat{\mathbf{s}}, \widehat{\varepsilon_Y}}\left[Y \mid \widehat{\mathbf{s}}, \mathbf{S}^c, \widehat{\varepsilon_Y}\right]$$
$$= \mathbb{E}_{\mathbf{S}^c|\widehat{\mathbf{s}}}\left[f(Pa_Y(\mathbf{s}, \mathbf{S}^c))\right] + \varepsilon_Y = \nu'(\mathbf{S}) + \varepsilon_Y. \tag{9}$$

We can perform the first step by marginalizing over $\mathbf{S}^c$ in $\mathcal{P}_{\mathbf{s}, \varepsilon_Y}$. Then, $(\mathbf{S}^c \perp\!\!\!\perp E_Y \mid \mathbf{S})_{\mathcal{G}_{\overline{\mathbf{S}, E_Y}}}$ because any path $p$ connecting $E_Y$ must necessarily have a collider in $Y$, since $De(Y) = Y$, therefore blocking the path. By R3, $\mathcal{P}_{\widehat{\mathbf{s}}, \widehat{\varepsilon_Y}}(\mathbf{S}^c) = \mathcal{P}_{\widehat{\mathbf{s}}}(\mathbf{S}^c)$ so we can remove it from the expectation over $\mathbf{S}^c$. On the other hand, we know that $Y = f(Pa_Y) + E_Y$ and by the linearity of expectations, we can remove $\varepsilon_Y$ from the expectation. Finally, for later clarity, we can denote the first term by $\nu'(\mathbf{S})$.

Next, let us solve the analogous term for $\mathbf{S}$:

$$\nu(\mathbf{S}) = \mathbb{E}\left[Y \mid \widehat{\mathbf{s}}\right] = \mathbb{E}_{\mathbf{S}^c, E_Y|\widehat{\mathbf{s}}}\left[Y \mid \widehat{\mathbf{s}}, \mathbf{S}^c, E_Y\right]$$
$$= \mathbb{E}_{\mathbf{S}^c|\widehat{\mathbf{s}}}\left[f(Pa_Y(\mathbf{s}, \mathbf{S}^c))\right] + \mathbb{E}\left[E_Y\right] = \nu'(\mathbf{S}) + \mathbb{E}\left[E_Y\right]. \tag{10}$$

We proceed similarly, beginning with a marginalization over $\mathbf{S}^c$ and $E_Y$ this time. In order to separate $E_Y$ from the first term, note that $(\mathbf{S}^c \perp\!\!\!\perp E_Y)_{\mathcal{G}_{\overline{\mathbf{s}}}}$ and $(\mathbf{S} \perp\!\!\!\perp E_Y)_{\mathcal{G}_{\overline{\mathbf{s}}}}$ for the same reason as before: any path connecting $E_Y$ must necessarily pass through $Y$, which acts as an unconditioned collider, thereby blocking it. Hence, $P(\mathbf{S}^c, E_Y \mid \widehat{\mathbf{s}}) = P(\mathbf{S}^c \mid \widehat{\mathbf{s}})P(E_Y \mid \widehat{\mathbf{s}}) = P(\mathbf{S}^c \mid \widehat{\mathbf{s}})P(E_Y)$, with the former step by the rules of independence and the latter by R3. We can then split the expectation, this time with $\mathbb{E}\left[E_Y\right]$ as the second term and using the same notation $\nu'(\mathbf{S})$ again.

With these two computations, we can see that:

$$\nu(\mathbf{S} \cup \{E_Y\}) - \nu(\mathbf{S}) = \varepsilon_Y - \mathbb{E}\left[E_Y\right], \tag{11}$$

and substituting it into the SHAP formula (with $K+1$ players):

$$\phi'_{E_Y} = \sum_{\mathbf{S} \subseteq \mathbf{X}} \frac{1}{K+1} \binom{K}{|S|}^{-1} (\varepsilon_Y - \mathbb{E}\left[E_Y\right])$$
$$= \sum_{s=0}^{K} \binom{K}{s} \frac{1}{K+1} \binom{K}{s}^{-1} (\varepsilon_Y - \mathbb{E}\left[E_Y\right])$$
$$= \varepsilon_Y - \mathbb{E}\left[E_Y\right]. \tag{12}$$

We transform the first to second step by realizing that we do not need to know what the coalitions $\mathbf{S}$ are, only their cardinality, so we can transform $\sum_{\mathbf{S} \subseteq \mathbf{X}}$ into $\sum_{s=0}^{K}$ by multiplying by $\binom{K}{s}$, the number of combinations of $s$ elements. This term and its inverse cancel out, and $K+1$ constant terms summed together cancels with $\frac{1}{K+1}$, resulting in $\varepsilon_Y - \mathbb{E}\left[E_Y\right]$.

Now, note that $\varepsilon_Y = y - f(pa_Y)$ and:

$$\mathbb{E}\left[Y \mid pa_Y\right] = \mathbb{E}\left[f(pa_Y) + E_Y\right] = f(pa_Y) + \mathbb{E}\left[E_Y\right], \tag{13}$$

so $f(pa_Y) = \mathbb{E}\left[Y \mid pa_Y\right] - \mathbb{E}\left[E_Y\right]$. Then,

$$\phi'_{E_Y} = \varepsilon_Y - \mathbb{E}\left[E_Y\right] = y - f(pa_Y) - \mathbb{E}\left[E_Y\right] = y - \mathbb{E}\left[Y \mid pa_Y\right]. \tag{14}$$

This proves the value for $\phi'_{E_Y}$. Let us now compute $\phi'_X$ for any $X \in \mathbf{X}$. Let $\mathbf{S} \subseteq (\mathbf{X} \cup \{E_Y\}) \setminus \{X\}$. If $E_Y \in \mathbf{S}$, we can apply eq. (9) and $\nu(\mathbf{S}) = \nu'(\mathbf{S} \setminus \{E_Y\}) + \varepsilon_Y$. Otherwise, eq. (10) gives us

$\nu(\mathbf{S}) = \nu'(\mathbf{S}) + \mathbb{E}[E_Y]$. Now, $\phi'_X$'s computation can use both results:

$$\phi'_X = \sum_{\mathbf{S} \subseteq \mathbf{X} \setminus \{X\}} \frac{1}{K+1} \binom{K}{|\mathbf{S}|+1}^{-1} (\nu(\mathbf{S} \cup \{E_Y, X\}) - \nu(\mathbf{S} \cup \{E_Y\}))$$

$$+ \sum_{\mathbf{S} \subseteq \mathbf{X} \setminus \{X\}} \frac{1}{K+1} \binom{K}{|\mathbf{S}|}^{-1} (\nu(\mathbf{S} \cup \{X\}) - \nu(\mathbf{S}))$$

$$= \sum_{\mathbf{S} \subseteq \mathbf{X} \setminus \{X\}} \frac{1}{K+1} \left( \binom{K}{|\mathbf{S}|+1}^{-1} (\nu(\mathbf{S} \cup \{X\}) - \nu(\mathbf{S})) + \binom{K}{|\mathbf{S}|}^{-1} (\nu(\mathbf{S} \cup \{X\}) - \nu(\mathbf{S})) \right)$$

$$= \sum_{\mathbf{S} \subseteq \mathbf{X} \setminus \{X\}} \frac{1}{K+1} \left( \binom{K}{|\mathbf{S}|+1}^{-1} + \binom{K}{|\mathbf{S}|}^{-1} \right) (\nu(\mathbf{S} \cup \{X\}) - \nu(\mathbf{S}))$$

$$= \sum_{\mathbf{S} \subseteq \mathbf{X} \setminus \{X\}} \frac{1}{K} \binom{K-1}{|\mathbf{S}|}^{-1} (\nu(\mathbf{S} \cup \{X\}) - \nu(\mathbf{S})) = \phi_X \qquad (15)$$

We first split the SHAP formula in two: those sets that include $E_Y$ and those that do not; note that the combination terms are altered to reflect the size of the base set applied to $\nu$ ($|\mathbf{S}| + 1$ in the first case since the base set is $\mathbf{S} \cup \{E_Y\}$). For the first to second step, we can bring together the two sums, and transform the first difference,

$$\nu(\mathbf{S} \cup \{E_Y, X\}) - \nu(\mathbf{S} \cup \{E_Y\}) = \nu'(\mathbf{S} \cup \{X\}) - \nu'(\mathbf{S}) = \nu(\mathbf{S} \cup \{X\}) - \nu(\mathbf{S}), \qquad (16)$$

by applying eq. (9) and eq. (10) and cancelling the $\varepsilon_Y$ and $\mathbb{E}[E_Y]$ terms, respectively. We now sum the two inverse combination terms; let $s := |\mathbf{S}|$, with $s \leq K - 1$ since $\mathbf{S} \subseteq \mathbf{X} \setminus \{X\}$:

$$\binom{K}{s+1}^{-1} + \binom{K}{s}^{-1} = \frac{(s+1)!(K-s-1)! + s!(K-s)!}{K!}$$

$$= \frac{(s+1) \cdot s!(K-s-1)! + (K-s) \cdot s!(K-s-1)!}{K \cdot (K-1)!}$$

$$= \frac{(s+1+K-s)s!(K-s-1)!}{K \cdot (K-1)!}$$

$$= \frac{K+1}{K} \cdot \frac{s!(K-s-1)!}{(K-1)!} = \frac{K+1}{K} \binom{K-1}{s}^{-1} \qquad (17)$$

Substituting this result back into eq. (15), we can cancel out $K + 1$. We arrive at the last formula, which is exactly the value for do-SHAP in the $K$-player game without $E_Y$.

Finally, in order to prove the last step, let us first prove that $\mathbb{E}[Y \mid \widehat{\mathbf{x}}] = \mathbb{E}[Y \mid pa_Y]$:

$$\mathbb{E}[Y \mid \widehat{\mathbf{x}}] = \mathbb{E}\left[Y \mid \widehat{pa_Y}, \widehat{pa_Y^c}\right] = \mathbb{E}[Y \mid \widehat{pa_Y}] = \mathbb{E}[Y \mid pa_Y]. \qquad (18)$$

We first split $\mathbf{X} = Pa_Y \cup Pa_Y^c$. Next, we apply R3 with $(Y \perp\!\!\!\perp Pa_Y^c \mid Pa_Y)_{\mathcal{G}_{\overline{\mathbf{x}}}}$ since $\mathcal{G}_{\overline{\mathbf{x}}}$ can only contain edges leading from $Pa_Y$ to $Y$ and from confounders $\mathcal{U}_{\{Y, \cdot\}}$ to $Y$; therefore, no path can exist from $Pa_Y^c$ to $Y$. Then, we apply R2 with $(Y \perp\!\!\!\perp Pa_Y)_{\mathcal{G}_{\underline{Pa_Y}}}$: since $Y$ has no descendants and we remove any incoming edges to $Y$ in $\mathcal{G}_{\underline{Pa_Y}}$ except for the ones starting from confounders in $\mathcal{U}_{\{Y, \cdot\}} = \varnothing$ (by assumption), $Y$ must be independent of $Pa_Y$ in $\mathcal{G}_{\underline{Pa_Y}}$, proving the expression.

Now we can prove the remaining fact:

$$\sum_{X \in \mathbf{X}} \phi'_X + \phi'_{E_Y} = (\mathbb{E}[Y \mid \widehat{\mathbf{x}}] - \mathbb{E}[Y]) + (y - \mathbb{E}[Y \mid pa_Y]) = y - \mathbb{E}[Y] \qquad (19)$$

$$\square$$

# E    Experiments

These experiments are executed on personal computers (particularly, a Macbook with an M3 Pro chip) and do not require an infrastructure of workers for their execution. No experiment lasted longer than 6 hours to execute in total throughout their multiple replications.

### E.1 Synthetic dataset

We include here further details about the Synthetic experiment in section 5.1, bigger figures for the Markovian case (for better visibility) and a discussion on the semi-Markovian case. Please refer to the supplementary code for the actual implementation of these experiments.

#### E.1.1 Implementation details

We chose several SCM architectures for the experiment; here we justify these choices. Regarding the classic approach of modeling an SCM with each of its functions $f_k \in \mathcal{F}$ separately, some of the approaches mentioned in section 2 focus on how to model complex multivariate r.v.s (e.g., images), but the remaining univariate r.v.s are modelled by specifying which probability distribution family they belong to. Since our DGP consists exclusively of univariate continuous r.v.s, these proposals are equivalent. Instead, we will employ Deep Causal Graph (DCG) [13], a general framework for all sorts of implementations of SCMs. In particular, with DCGs, we can train three different kinds of SCM: 1) a **linear** SCM with Normal distributions for each variable, used as a baseline; 2) the Distributional Causal Node [20] architecture, where every node is modeled after a probability distribution family with a feed-forward network for the computation of its parameters (**DCN**); and 3) **DCG**s powered with its most flexible implementation for continuous nodes, based on Normalizing Flows. Finally, in order to test the alternative approach of modeling SCMs not node-wise, but the graph as a whole, we could use Variational Causal Autoencoder (VACA) [26] or Causal Normalizing Flows (**CNF**) [28]. However, as stated by the authors of CNF based on their experiments, "VACA shows poor performance, and is considerably slower due to the complexity of GNNs". For this reason, we opt for CNFs as a representative of this alternative approach for SCM modeling.

Regarding the definition of the synthetic DGP, we employ a set of non-linear functions along with exogenous samples from diverse continuous r.v.s for the generation of new samples. Let $\chi^2(k)$ be the Chi-squared distribution with $k$ degrees of freedom, $\mathcal{B}(\alpha, \beta)$ the Beta distribution, $\mathcal{N}(\mu, \sigma)$ the Normal distribution, and $\texttt{Exponential}(\lambda)$ the Exponential distribution. We sample from each latent variable first and then apply the functions in $\mathcal{F}$ in topological order:

$$
\begin{cases}
u \sim \chi^2(k = 10); \\
\varepsilon_Z \sim \mathcal{B}(\alpha = 2, \beta = 5); \\
\varepsilon_X \sim \mathcal{N}(\mu = 0, \sigma = 0.1); \\
\varepsilon_{A,1} \sim \texttt{Exponential}(\lambda = 1); \\
\varepsilon_{A,2} \sim \mathcal{N}(\mu = 0, \sigma = 0.1); \\
\varepsilon_B \sim \mathcal{N}(\mu = 0, \sigma = 1); \\
\varepsilon_C \sim \mathcal{N}(\mu = 0, \sigma = 0.5); \\
\varepsilon_Y \sim \mathcal{N}(\mu = 0, \sigma = 0.5);
\end{cases}
\qquad
\begin{cases}
z \leftarrow \varepsilon_Z; \\
x \leftarrow |z(u - 5) + \varepsilon_X|; \\
a \leftarrow |\sqrt{x} + \varepsilon_{A,1} + \varepsilon_{A,2}|; \\
b \leftarrow 5\sin(a) - \frac{u}{10} + \varepsilon_B; \\
c \leftarrow \log(1 + b^2) + \varepsilon_C; \\
y \leftarrow \log \frac{z}{1-z} + \left(\frac{x}{10}\right)^2 + c + \varepsilon_Y;
\end{cases}
\tag{20}
$$

Note that we have diverse continuous variables: some non-negative, some restricted to $(0, 1)$, some with unrestricted support. For the **DCN** implementation, where we need to assume the r.v.'s probability distribution family, we will employ Exponential, Beta and Normal distributions respectively. Each set of parameters $\Theta_X$ can be computed with a shared feed-forward network using a Graphical Conditioner [13]. For the **DCG** implementation with Normalizing Flows, we will use a Normal Distribution as its base noise ($E_X$) and the following flow structure (from $X$ to $E_X$):

- Affine layer, $f(x) = \sigma x + \mu$, with $\mu, \sigma$ learnable parameters.
- 3 blocks of:
    - Rational-Quadratic Spline Flow [40], defined on the interval $[-5, 5]$, with $K = 8$ bins.
    - Affine layer.

Additionally, depending on the support of the r.v. to be modelled, we prepend another layer to transform the original domain into $\mathbb{R}$ before the application of the first Affine layer. In particular, for flows defined in $(0, 1)$, we use a Logit transformation $f(x) := \log \frac{x}{1-x}$, and for non-negative flows, an inverse Softplus layer $f(x) := \log(e^x - 1)$ (using the identity after a certain threshold for numerical stability). All parameters not set as hyperparameters are computed by an external trainable Conditioner that takes the node's parents as input and outputs their value.

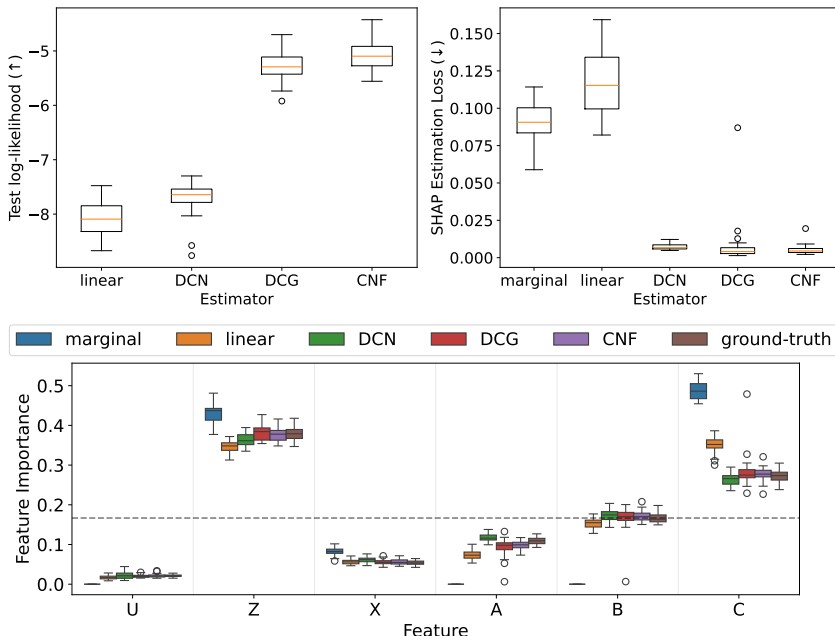

Figure 7: Markovian case. Box-plots computed over 30 realizations of the dataset. (a) Distribution adjustment score, log-likelihood (bigger is better). (b) SHAP estimation loss, $\mathcal{L}$ (lower is better). (c) Feature Importance (the closer to *ground-truth*, the better). Dashed horizontal line represents uniform importance $\left(\frac{1}{K}\right)$.

Regarding the Conditioner network's architecture, it is a standard feed-forward network with ELU activations [41], 2 hidden layers of dimension 32, and a standardizer layer at the beginning defined with the training dataset. This is used for the **DCN** and **DCG** SCMs. Regarding the **CNF** architecture, it uses a Softclip-constrained NSF-based architecture similar to ours, but with 4 stacked Spline layers (the diameter of the graph) and a MADE Conditioner to learn to model all variables at once. In this case, the conditioner uses 3 hidden layers with dimension 32 and ELU as its activation.

In terms of training, we use the AdamW optimizer [42] with Early Stopping (after 100 epochs with no improvement), using learning rate $10^{-3}$, weight decay $10^{-2}$ and batch size 100. Regarding SHAP estimation, since we only have 5 variables, we use the exact permutation method, taking 1,000 samples from each SCM for the Monte Carlo estimators of $\nu(\mathbf{S})$.

Finally, see fig. 7 for a bigger representation of the plots in the Markovian case.

### E.1.2 Semi-Markovian case

We also test our approach on the semi-Markovian case, where the latent confounder $U_{\{X,B\}}$ is not observed. As stated in section 5.1, CNF cannot be applied without the causal sufficiency assumption, so we will proceed with the remaining SCMs.

Figure 8 shows the same three plots as in the Markovian case, with similar results. The linear SCM cannot properly estimate $\mathcal{P}(\mathcal{V})$, which results in worse SHAP loss. DCNs achieve similar results to DCGs, but DCGs exhibit the best distribution adjustment and estimation performance. Finally, marginal SHAP results in different estimations than do-SHAP, something that is patently clear in the FI plot, where $Z$ and $C$'s contribution are overestimated while $A$ and $B$ are underestimated. Note that we obtain the same explanations as in the Markovian case even though we cannot measure the latent confounder nor know its distribution.

### E.2 FRA experiments

We will now describe additional tests for FRA.

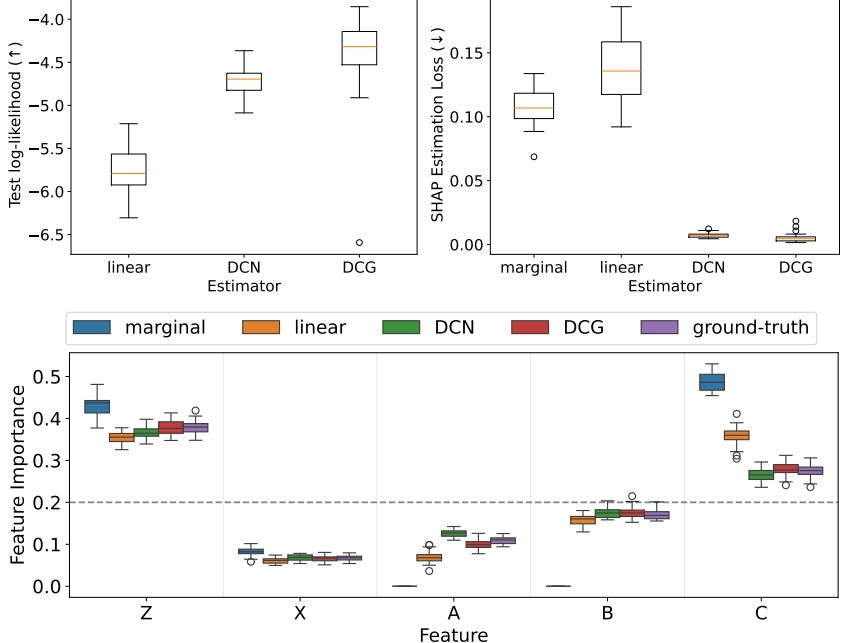

Figure 8: Semi-Markovian case. Box-plots computed over 30 realizations of the dataset. (a) Distribution adjustment score, log-likelihood (bigger is better). (b) SHAP estimation loss, $\mathcal{L}$ (lower is better). (c) Feature Importance (the closer to ground-truth, the better). Dashed horizontal line represents uniform importance $\left(\frac{1}{K}\right)$.

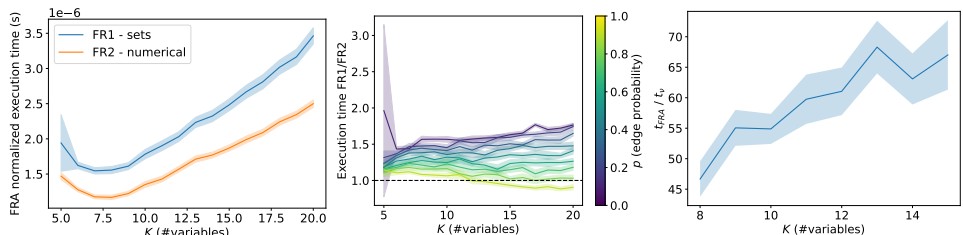

Figure 9: FRA experiments. (a) shows the errors bars (at 2-sigma over 270 replications) for the average normalized execution times of FR1 (sets) and FR2 (numerical) per number of nodes ($K$). (b) shows the error bars (at 2-sigma over 30 replications) of the ratio between FR1 time and FR2 time, split by edge probability ($p$), over the number of nodes $K$. If above 1, FR2 is faster. (c) shows the error bars (at 2-sigma over 30 replications) for the mean of quotients between the total time executing FRA (+permutations) and for estimating all sampled $\nu(\mathbf{S})$.

Figure 9 shows comparisons between both versions of the FRA algorithm: FR1 (using sets, algorithm 2) and FR2 (using integers, algorithm 3). Note that FR2 is consistently and significantly faster that FR1, and less memory-intensive, since the Fr cache needs only store numerical encodings of sets, instead of sets of integers. However, depending on graph topology, particularly with many edges ($p = 0.9$), FR1 can be faster than FR2, as shown by fig. 9 (b). In any case, FRA is negligible time-wise *w.r.t.* the time needed for the estimation of $\nu(\mathbf{S})$, as we will see next.

In the second experiment, as described in section 5.2, we employed a synthetic DGP consisting of a linear function. We generate 1,000 i. i. d. samples from this DGP to create training, validation and test sets with ratios 8:1:1. We train a linear DCG with Normal DCN nodes, which is fitting for this dataset and deliberately lightweight, so as to show the improvements from the application of FRA even in the case where the model is particularly fast; FRA can only improve time-gains with more expressive and expensive models. We use a learning rate of $10^{-2}$, weight decay of $10^{-2}$, a batch

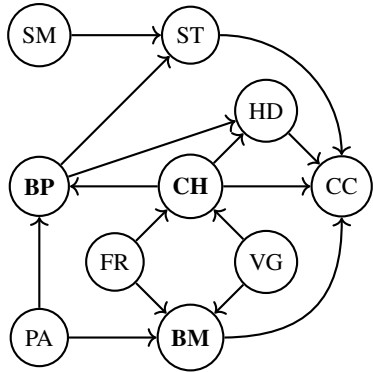

Figure 10: Diabetes Causal Graph, with variables Physical Activity (PA), Fruit (FR), Veggies (VG), Smoker (SM), BMI (**BM**), High Cholesterol (**CH**), High Blood Pressure (**BP**), Heart Disease or Attack (HD), Stroke (ST), Chol. Check (CC). Boldface letters denote $Pa_Y$, with $Y$, Diabetes, the target variable, not represented for clarity. We can skip modeling any non-ancestors of $Y$ (theorem 4.2): SM, ST, HD and CC.

size of 100, early stopping with patience of 10 epochs and AdamW [42] as the optimizer. For each $\nu$ estimation, we generate 100 samples from the SCM.

For this experiment, we kept track of do-SHAP execution time as well as the time for the computation of $\nu(\mathbf{S})$ specifically; this allows us, by subtraction, to compute the time needed for FRA and generating the permutations $\pi$ from where the sets $\mathbf{S}$ emerge. If we compare these two quantities, dividing the $\nu$ execution time by the FRA+permutation time, as shown in fig. 9 (c), we see that FRA is orders of magnitude faster than $\nu$; this means that, at a negligible cost, we can speed up do-SHAP by a significant ratio, specifically the number of FR coalitions.

# F Applications

We now showcase do-SHAP explanations on two real-world datasets as illustrative examples.

### F.1 Diabetes dataset

Here we discuss the Diabetes Health Indicators Dataset [43], containing healthcare statistics and lifestyle survey information along with their diagnosis (or not) of diabetes, for the year 2015. Note that the dataset is biased, with 14% of individuals having diabetes. We start with a preprocessed version of the original questionnaire dataset, from which 21 features and the target variable are extracted. We select 10 out of 21 features to provide a more easily understandable problem for the reader.

We construct a causal graph (see fig. 10) relating these variables and train an SCM to model them, with which we finally compute their do-Shapley values. We design the graph using common sense. Note that any causal analysis depends on its graph being sound, but it can be replicated at any time once a better graph is found; for this reason, please take this as an illustrative example, since its conclusions regarding healthcare are not necessarily rigorous. We train a DCG with the same setup as before, except that every variable other than BMI is binary, so they are modelled with Bernoulli DCNs.

Our objective here is not to explain a ML model, but the data itself; particularly, how each variable affects the likelihood of diabetes. However, the effect of the noise is not as clear in a classifier as in a regressor, since $\phi_{E_X} = y - \mathbb{E}[Y \mid pa_Y] = y - P(Y \mid pa_Y)$; for an application of theorem 4.9, please refer to section F.2.

We compute do-SHAP for the first 1,000 test set entries, and measure FI. See fig. 11 a); HighBP, HighChol and BMI appear to be the most important variables, with Physical Activity, and fruit and vegetable intake having a less pronounced role. It is also important to consider the dependency between feature values and do-SHAP values; see the beeswarm plot in fig. 11 c), which shows

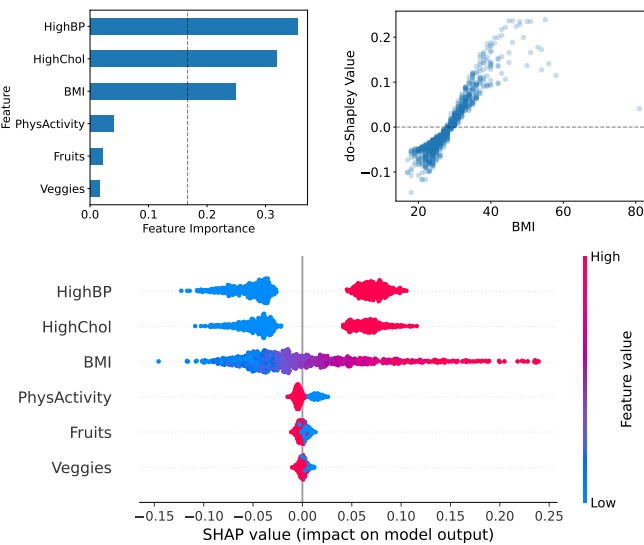

Figure 11: Diabetes Dataset. a) do-SHAP Feature Importance. Dashed vertical line represents uniform importance ($\frac{1}{K}$). b) Scatterplot between BMI value and do-SVs. c) do-SHAP beeswarm plot, relating do-SVs and feature values.

clear-cut effects in do-SV sign and magnitude for HighBP and HighChol, with a more nuanced relationship between BMI (continuous) and do-SVs, which we plot in fig. 11 b); BMI 30 (typically categorized as obese) seems to be the cutting point after which higher BMI values increase the chances of diabetes up to 20%, while lower values decrease that likelihood up to 10%.

## F.2  Bike Rental dataset

We now study the Bike Rental Dataset [44], describing the number of rentals in a bike sharing service in Washington D.C., between 2011 to 2012 (both included), measured on an hourly basis, along with weather data and whether that day was a working day. Again, our objective in this case is to explain the data itself; particularly, how each variable affects the number of rented bikes.

We design a causal graph, depicted in fig. 12 (a). As stated before, please do not take the conclusions from this experiment as is, but merely as an illustrative example. We train a DCG with the same architecture and training parameters as the synthetic experiment, except that *hour* can be modeled with a uniform distribution on the integer interval $[0, 23]$. We train our DCG and compute the do-Shapley values for the test set entries. However, since we are operating on an inaccessible DGP in this case, we also want to measure the effect of the noise, employing theorem 4.9.

We measure FI, represented in fig. 12 (b). Hour seems to be the major cause of the target variable, as is to be expected, followed by noise (given by the inherent variance of the target conditioned on its parents) and temperature, which also conditions on the likelihood of users renting bikes. Other variables also do have an impact, with only wind speed and weather (categorical, with three levels) having a less pronounced effect.

The relationship between feature values and do-SVs is more informative; we use scatter plots in order to study how each value affects the outcome; see fig. 13. *Hour* presents positive attribution during daytime from 8AM to 8PM, with night-time having negative attribution; *temperature*'s effect is mainly negative, with certain temperatures being less inviting for cycling (below 15ºC and above 35ºC); in the same way, humidity only affects past 80%, as well as wind speed, past 15 km/h.

## G  Comparison between do-SVs and non-causal approaches

In this appendix, we expand on the discussion about why do-SHAP results in more reliable explanations than its non-casual alternatives (marginal-SHAP and conditional-SHAP). We follow on the example presented in section 1, here replicated in fig. 14 for reader's convenience.

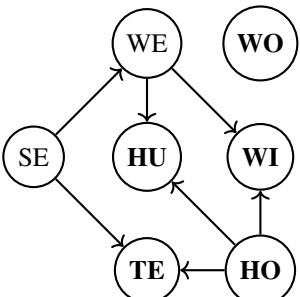 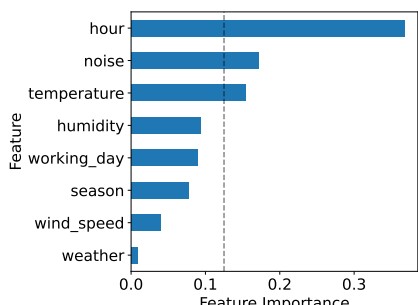

Figure 12: Bike Rental Dataset. a) Causal Graph with nodes Season (SE), Weather (WE), Humidity (HU), Temperature (TE), Working day (WO), Wind speed (WI), Hour (HO). Boldface letters denote $Pa_Y$, with target $Y$, Rentals, not represented for clarity. b) Feature Importance (FI) for each variable. Dashed vertical line represents uniform importance $(\frac{1}{K})$.

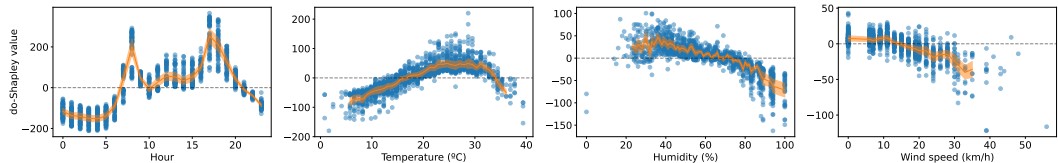

Figure 13: Bike Rental, continuous features' values against their SHAP values. Errors bars at 2-sigma.

Firstly, let us reason about how each method will behave in this particular Data Generating Process (DGP). In marginal-SHAP, consider for example $\nu(\{E\})$, where we would marginalize $A$ and $S$ regardless of the assigned value to $E$, thereby ignoring the impact that education level may have on the seniority level of the employee (their standing in the company), and producing out-of-distribution configurations $(a, e, s)$. Alternatively, with conditional SHAP, we would operate with $P(A, S \mid E = e)$, thereby including this dependency between $E$ and $S$, but also taking whichever value of $A$ would have generated this specific value $e$, which introduces in turn an anti-causal effect $(E \to A)$.

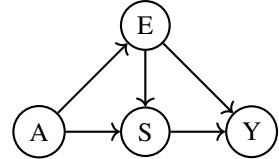

Figure 14: *Salary* causal graph: Age (A), Education (E), Seniority (S) and Salary (Y).

Since both approaches ignore the causal structure, they incorporate non-causal effects that fail to reflect the real DGP, and would therefore lead to unreliable explanations. In contrast, do-SHAP does take into account this causal effect, by using the intervention $do(E = e)$, therefore affecting S ($E \to S$) while not affecting A ($E \leftarrow A$); not only that, but $A$'s effect is de-confounded by blocking the back-door path $E \leftarrow A \to S \to Y$. See table 1 for a depiction of which coalitions **S** result in the same value as do-SHAP's $\nu$, for marginal-SHAP and conditional-SHAP.

|  | $\varnothing$ | $\{A\}$ | $\{E\}$ | $\{S\}$ | $\{A, E\}$ | $\{A, S\}$ | $\{E, S\}$ | $\{A, E, S\}$ |
|---|---|---|---|---|---|---|---|---|
| **marginal-SHAP** | $=$ | $\neq$ | $\neq$ | $\neq$ | $\neq$ | $\neq$ | $\neq$ | $=$ |
| **conditional-SHAP** | $=$ | $=$ | $\neq$ | $\neq$ | $=$ | $\neq$ | $\neq$ | $=$ |

Table 1: When does do-SHAP's $\nu$ equal marginal-SHAP's or conditional-SHAP's $\nu$?

Secondly, we will illustrate this reasoning with an example SCM corresponding to the same graph; see eq. (21). Let us consider Bernoulli r.v.s $A, E, S, Y$ with parameters $p_A, p_E, p_S, p_Y$, respectively. These parameters are computed following the aforementioned causal graph, with the following structural equations, which generate samples $(a, e, s, y)$ through Bernoulli sampling.

$$\begin{cases} p_A = 0.25 \\ p_E = 0.5a + 0.25 \\ p_S = 0.25a + 0.5e + 0.1 \\ p_Y = 0.5e + 0.3s + 0.1 \end{cases} \tag{21}$$

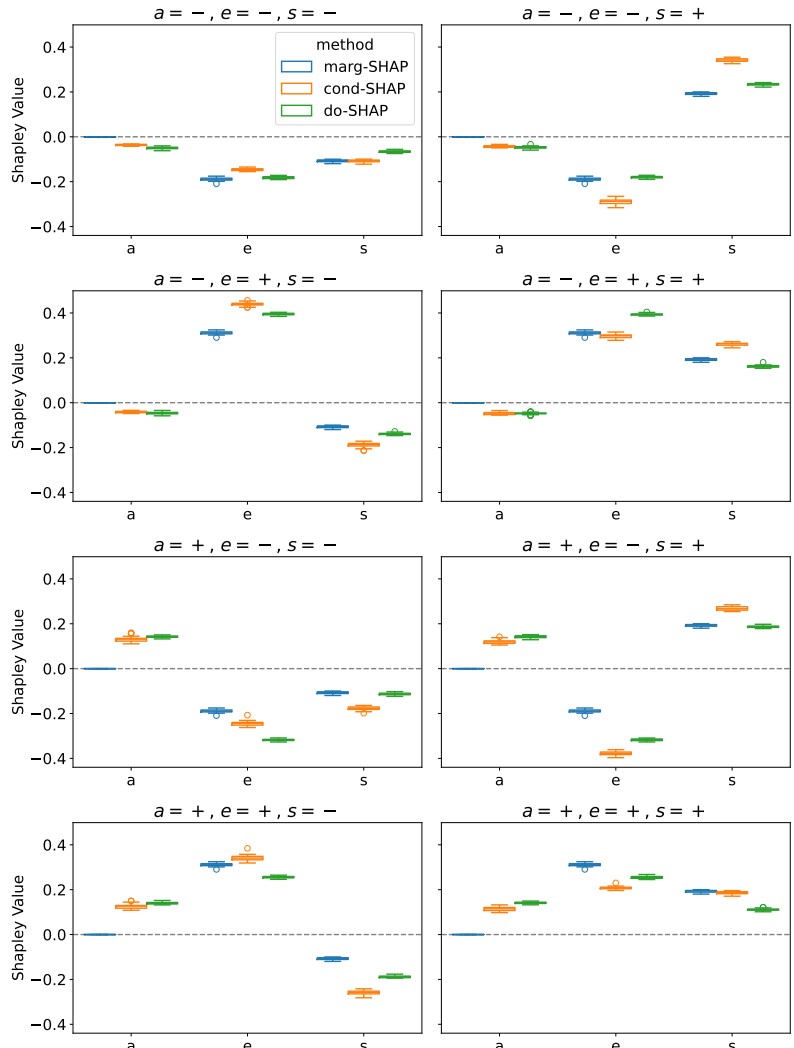

Figure 15: Salary example, comparison between marginal-SHAP, conditional-SHAP and do-SHAP for each input variable $(A, E, S)$ on every factual combination (title).

We evaluate marginal-SHAP, conditional-SHAP, and do-SHAP on this DGP. We run the following procedure 30 times to obtain error bars. For each replication, we generate $1,000$ background samples (used for marginalization in the first two methods) and $8$ test samples (one for every configuration $(a, e, s)$) to explain. We estimate marginal-SVs on these test set samples $\mathbf{x}$ by approximating $\nu_{marg}(\mathbf{S}) := \mathbb{E}_{\mathbf{x}' \sim \mathcal{P}(\mathbf{X})} \left[ P(Y \mid \mathbf{x_S}, \mathbf{x}'_{\mathbf{S}^c}) \right]$ with the i.i.d. background samples. Similarly, we estimate conditional-SVs by approximating $\nu_{cond}(\mathbf{S}) = \mathbb{E}_{\mathbf{x}' \sim \mathcal{P}(\mathbf{X}|\mathbf{x_S})} \left[ P(Y \mid \mathbf{x_S}, \mathbf{x}'_{\mathbf{S}^c}) \right]$, averaging over those $\mathbf{x}'$ background samples that fulfill $\mathbf{x}'_{\mathbf{S}} = \mathbf{x_S}$. Finally, we compute do-SVs by estimating $\nu(\mathbf{S}) = \mathbb{E}_{\mathbf{x}' \sim \mathcal{P}(\mathbf{X}|do(\mathbf{S}=\mathbf{x_S}))} \left[ P(Y \mid \mathbf{x_S}, \mathbf{x}'_{\mathbf{S}^c}) \right]$, sampling and intervening on the DGP directly with interventions $do(\mathbf{S} = \mathbf{x_S})$. Note that for this particular experiment, we do have access to the true structural equation $Y = f_Y(a, e, s)$ and we employ it in our estimations instead of training a model to approximate it. We split by each test sample (every factual combination $(a, e, s)$) and plot the corresponding estimations with boxplots in fig. 15.

We use the results of this experiment to exemplify how marginal-SHAP and conditional-SHAP fail to address the behavior of the underlying DGP, hence providing explanations whose insights are not reliably applicable to the real world. Meanwhile, do-SHAP does take into account the corresponding data structure, and overcomes these weaknesses.

On the one hand, marginal-SHAP sets $\phi_A = 0$ for all possible configurations, since $A$ does not appear in $Y$'s structural equation; as such, $A$ cannot have an effect on $Y$ within marginal-SHAP, since interventions on $A$ have no impact on $E$ or $S$ (even if they do in the real DGP). As for $\phi_E$, it disregards the effect of $E$ on $S$ while also not de-confounding the back-door effect $E \leftarrow A \rightarrow S \rightarrow Y$, resulting in significant differences with do-SHAP, particularly on $(a = -, e = +)$ and $(a = +, e = -)$, given that the correlation between the two is ignored with marginal-SHAP's intervention. Finally, $\phi_S$ is closer to an intervention with do-SHAP since $S$ does not propagate its intervention on any other variable before $Y$; however, it is unable to control for the back-door effects ($S \leftarrow E \rightarrow Y$ or $S \leftarrow A \rightarrow E \rightarrow Y$), which explains the difference with do-SHAP.

On the other hand, conditional-SHAP results in more similar attributions to do-SHAP, but still significantly different. This similarity can be explained by the fact that half the coalitions result in the same $\nu$-output as do-SHAP in this particular graph, as made explicit in table 1. However, even if the conditional may affect descendants of a variable as a do-intervention does, it can also introduce anti-causal effects and cannot block back-door paths, which explains the differences with do-SHAP.

In summary, both marginal-SHAP and conditional-SHAP misrepresent the underlying DGP, given that they ignore its underlying causal structure. This is the reason why do-SHAP results in more reliable explanations than the alternatives.

## H  do-SHAP estimation example

This section is devoted to explaining how to apply EA estima-
tion in practice for do-SHAP. This is meant as an illustrative
written explanation; for a code example, please refer to the
experiments code, submitted with the supplementary material.

We will focus on the graph introduced in section 5.1, here
replicated in fig. 16 for reader's convenience. We will employ
the semi-Markovian version of this example (meaning, there is a
latent confounder $U_{\{X,B\}}$ between $X$ and $B$). We will assume
the (measured) data distribution is composed of unconstrained
continuous r.v.s, with an unknown prior distribution for $U_{\{X,B\}}$,
but with a known causal graph $\mathcal{G}$.

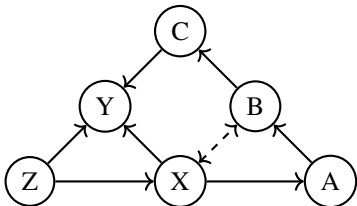

Figure 16: Synthetic semi-
Markovian graph.

In terms of notation, let $Pa'_X = Pa_X \sqcup \mathcal{U}_{\{X,\cdot\}}$ be the concate-
nation of the parents of a certain node $X \in \mathbf{X}$ and any latent confounders pointing to it. For example,
in our current graph, $Pa'_B = (A, U_{\{X,B\}})$.

### H.1  do-SHAP identifiability

Before starting, we need to confirm that do-SVs are indeed identifiable in this particular graph; otherwise, two SCMs trained on the same distribution may output different do-SV estimations, rendering them useless.

If there was a graphical criterion that fit the structure of our graph $\mathcal{G}$, it could be used at this step; for instance, if there are no latent confounders in a graph, do-SHAP is trivially identifiable. Since this is not the case, we need to test it using the ID algorithm [29]. See [31] or [32] for implementations in R and Python, respectively.

To guarantee do-SHAP identifiability, we need to test it for every coalition $\mathbf{S}$. In other words, ensure that $\nu_{\mathbf{x}}(\mathbf{S}) := \mathbb{E}\left[Y \mid do(\mathbf{S} = \mathbf{s})\right]$ is identifiable $\forall \mathbf{S} \subseteq \mathbf{X}$. Therefore, we run the ID algorithm on each of the $2^5 = 32$ queries $\nu(\mathbf{S})$; if all are identifiable, so are the do-SVs. Note that this test is independent to the data distribution, as it only requires the corresponding graph structure $\mathcal{G}$.

Given that this is a small graph, with only 5 $\mathbf{X}$ variables, amounting to 32 coalitions, it is feasible to test for identifiability before starting the process. In the general case, with potentially bigger graphs and $2^{|\mathbf{X}|}$ coalitions to test, this becomes infeasible or very expensive, and it is therefore recommended to test for identifiability *during* do-SV estimation; this lets us skip the identifiability check for any coalitions not appearing during the sampling process, as well as leveraging the FRA-cache to skip further identifiability checks on non-irreducible coalitions. In the following, we will indicate where in the do-SHAP process this check should be performed.

## H.2 SCM implementation

Given that $\mathcal{G}$ contains a latent confounder, we choose DCGs [13] as the SCM architecture, and for generality (to adjust to more complex data distributions) employ Normalizing Causal Flows (NCFs) as the node architecture. Refer to section E.1.1 for a possible implementation of NCFs (without domain adjustment, given that our variables are unconstrained real-valued r.v.s). These NCFs provide a general function $X = f_X(E_X, \Theta_X(Pa'_X))$ (with $E_X$ the corresponding exogenous noise signal for node $X$), which is used to sample new values $x \sim \mathcal{P}(X \mid Pa'_X)$ as well as to compute the log-likelihood of these values given its parents: $\log P(x \mid pa'_X)$. The choice of prior distribution for the exogenous signal $E_X$ and for the latent confounder $U_{\{X,B\}}$ is irrelevant, as long as the desired queries to estimate are identifiable (which has been previously tested) and the particular choice of prior guarantees enough modeling capacity to represent the data distribution $P(\mathbf{X})$. In this particular case, we choose a $\mathcal{N}(0, 1)$ for both priors.

Regarding the function $\Theta_X(Pa'_X)$, it returns the appropriate function parameters $\theta_X$ for each node's function $f_X$. These parameters define the shape of the distribution $\mathcal{P}(X \mid Pa'_X)$ and as such depend on the values $pa'_X$ of $Pa'_X$. After that, $f_X$ only depends on the parameters $\theta_X$ and the exogenous noise $\varepsilon_X \sim \mathcal{P}(E_X)$. This function $\Theta_X(.)$ could be modeled with a simple Multilayer Perceptron (MLP), one for each node, but this would inevitably result in overfitting for larger graphs. Instead, we employ the Graphical Conditioner presented in [13], a single MLP network that takes every node $\mathbf{X}$ and latent confounders $\mathcal{U}$ as the input and returns all parameters $\{\theta_X \mid X \in \mathbf{X}\}$ as the output. By using a particular training and inference strategy, the Graphical Conditioner allows to compute each of these functions $\Theta_X$ independently through a single network, thereby reducing training time and overfitting risk.

## H.3 SCM training

DCGs are trained with Maximum Likelihood Estimation, so we will define, for a particular sample $\mathbf{x} \sim \mathcal{P}(\mathbf{X})$, the Negative Log-Likelihood (NLL) loss to minimize as the training objective. Given the graph structure $\mathcal{G}$ of the data distribution to model, we can derive two formulas, one for the Markovian case (no latent confounders),

$$\mathcal{L} := -\log P(\mathbf{x}) = -\sum_{X \in \mathbf{X}} \log P(x \mid pa_X), \tag{22}$$

the other for the semi-Markovian case (with latent confounders),

$$\mathcal{L} := -\log P(\mathbf{x}) = -\log \mathbb{E}_{\mathcal{U}} \left[ \exp \sum_{X \in \mathbf{X}} \log P(x \mid pa'_X) \right]. \tag{23}$$

In our example, we need to employ eq. (23), given that $\mathcal{U} = \{U_{\{X,B\}}\} \neq \varnothing$. This latter expectation can be estimated by generating $M$ i.i.d. samples from $P(\mathcal{U})$ and averaging the results of the expectation's contents. The terms $\log P(x \mid pa'_X)$ can be estimated by the predefined node architectures (NCF in our example) using a simple function. Finally, the Monte Carlo average can be computed using the log-sum-exp trick for numerical stability.

By running an optimization algorithm (e.g., AdamW [42]) on the average of these NLL losses for random mini-batches of samples, we can learn the data distribution $P(\mathbf{X})$ with our SCM, from which we can now estimate (identifiable) do-SVs.

## H.4 do-SHAP estimation

Let us describe how to estimate do-SVs with our approach.

Firstly, we need to run the SHAP formula. In this case, with only 5 variables, we could employ the exact formula directly, using eq. (1). Instead, we exemplify the more general approach, compatible with larger graphs, with the permutations formula in eq. (2), which we approximate as described in section 3.4, here re-established for reader's convenience:

$$\phi_{\nu_{\mathbf{x}}}(X) = \mathbb{E}_{\pi \sim \mathcal{U}(\Pi(\mathbf{X}))} \left[ \nu_{\mathbf{x}}(\mathbf{X}_{\leq_\pi X}) - \nu_{\mathbf{x}}(\mathbf{X}_{<_\pi X}) \right], \tag{24}$$

where the expectation over permutations $\pi$ is estimated by generating $M$ i.i.d. permutations uniformly, and the internal $\nu_{\mathbf{x}}(.)$ terms are estimated as described in the following. However, before we run the

$\nu_{\mathbf{x}}$-estimation procedure, we employ an FRA-cache to compute the Frontier-Irreducible subsets $\mathbf{S}'$ of $\mathbf{X}_{\leq_\pi X}$ and $\mathbf{X}_{<_\pi X}$, hence reducing the number of $\nu_{\mathbf{x}}$-computations required.

Let us now discuss FRA. Consider the permutation $\pi = (A, B, Z, X, C)$. We need to attempt to reduce sets $\mathbf{X}_{\leq_\pi X} = \{A, B, Z, X\}$ and $\mathbf{X}_{<_\pi X} = \{A, B, Z\}$. By running the FRA algorithm (see algorithm 2 for a simpler definition with sets), we can see that both sets can be reduced by removing $A$, since $B$ acts as a frontier between $A$ and $Y$. Hence, we compute both values $\nu_{\mathbf{x}}(\{B, Z, X\})$ and $\nu_{\mathbf{x}}(\{B, Z\})$ and store their results in a cache for later look-up. If, on another permutation $\pi'$, we encounter coalitions with irreducible sets that have been computed before, we can retrieve the results from the cache directly, thereby reducing the number of required computations.

Now, if we have not tested identifiability at the beginning of the process, we must confirm identifiability of all encountered queries $\nu(\mathbf{S})$ before proceeding with their estimation, but only for the corresponding irreducible sets $\mathbf{S'}$. Whether one of such queries is identifiable or not is stored in a separate cache to avoid running the ID algorithm again once another coalition results in the same irreducible set. Finally, if any such query is deemed unidentifiable, the process must halt with an error, meaning that do-SVs cannot be estimated in this particular graph.

Then, let us describe how to estimate an arbitrary coalition value, $\nu_{\mathbf{x}}(\mathbf{S})$, which is accomplished with a general sampling procedure. In order to estimate this query, we need to generate $M$ i. i. d. samples $\mathbf{x}^{(i)} \sim \mathcal{P}(\mathbf{X} \mid do(\mathbf{S} = \mathbf{s}))$. We start by generating values $\varepsilon^{(i)} \sim \mathcal{P}(\mathcal{E})$ and $u^{(i)} \sim \mathcal{P}(\mathcal{U})$ from their respective prior distributions. Afterwards, we go node by node $X \in \mathbf{X}$ following any topological order of the graph, using the corresponding sampling functions $x^{(i)} = f_X(\varepsilon_X^{(i)}, \Theta_X(pa'_X{}^{(i)}))$ unless the variable $X$ is in the intervened coalition $\mathbf{S}$, in which case $x^{(i)}$ becomes the corresponding value from the to-be-explained sample $\mathbf{x}$. After we have sampled from every variable in the graph, we have a joint sample $\mathbf{x}^{(i)} \sim \mathcal{P}(\mathbf{X} \mid do(\mathbf{S} = \mathbf{s}))$. We pass each of these samples through our model $f_Y(.)$ to compute the corresponding $y^{(i)}$ values, which will finally be averaged for the final estimation of $\nu_{\mathbf{x}}(\mathbf{S})$.

As a note, while smaller coalitions are generally more expensive to evaluate than larger ones, since every node not in $\mathbf{S}$ requires to use its sampling mechanism to generate its values, this extra cost is negligible in comparison to evaluating extra coalitions that reduce to the same irreducible subset.

Finally, we can add further optimizations to this procedure. Consider the tuple $\nu_{\mathbf{x}}(\pi) = (\nu_{\mathbf{x}}(\pi_{:k}))_{k=0..K}$, with $\pi_{:k}$ the set of variables up to index $k$ on $\pi$ (inclusive) (note that $\pi_{:0} = \varnothing$). When computing SVs, instead of using the formula directly, we sample permutations $\pi$, compute the corresponding tuples $\nu_{\mathbf{x}}(\pi)$ and, from there, their diff-tuple $\Delta\nu_{\mathbf{x}}(\pi) = (\nu_{\mathbf{x}}(\pi_{:k}) - \nu_{\mathbf{x}}(\pi_{:k-1}))_{k=1..K}$. Note that each of these $\Delta\nu_{\mathbf{x}}(\pi)_k$ terms is the difference in the SHAP formula for variable $X := \pi_k$, so we can update our do-SV estimations $(\phi_X)_{X \in \mathbf{X}}$ simultaneously, which accelerates this computation by reducing the number of FRA-cache accesses and employing tensor operations. Along with this, other estimation approaches, such as antithetic sampling for these permutations, can be used to obtain better estimator efficiency, further reducing the number of required permutations.

### H.5 Final considerations

Putting all of this together amounts to a seemingly complex method to estimate feature attribution: from finding the assumed Causal Graph $\mathcal{G}$, defining the appropriate SCM architecture, training such a model, estimating the $\nu$ values, running FRA to avoid computations, to finally arriving at the do-SV estimations. However, given already-implemented SCM architectures with appropriate training and estimation procedures ready for use, EA do-SHAP is made practical. For this reason, we advocate for an open-source library bringing together different SCM architectures for easier switching between approaches, facilitating the general applicability of EA methods, and as a result, of do-SHAP explanations.

## I  Impact statement

This work introduces a tool for estimating do-SVs on any black-box system. As an explainability tool, it offers positive societal impact by enabling deeper understanding of complex AI systems, benefiting both scientific progress and business decision-making. Crucially, it addresses ethical concerns about AI trustworthiness by supporting the *right to explanation* in human-facing decision

systems, facilitating debugging, and enhancing transparency and accountability. Additionally, by enabling audits of opaque systems, the tool promotes responsible AI regulation, protecting human rights against the risks of powerful but opaque AI systems.

One potential negative application of these techniques is in terms of willingly or unwillingly obfuscating harmful behavior in black-box models. Given the complexity of these techniques and the subsequent analysis required to derive conclusions from its outputs, explainability techniques could be used to provide a superficial layer of supervision and result in misleading conclusions about the behavior of the system. Great care with respect to the assumptions and outputs of these tools must be taken in their application.

