# OpenReview forum: "Practical do-Shapley Explanations with Estimand-Agnostic Causal Inference"
_NeurIPS.cc/2025/Conference — NeurIPS 2025 spotlight_

### Official Review · Reviewer_kUZY · 2025-06-20

**Clarity:** 2
**Significance:** 3
**Originality:** 2
**Rating:** 4
**Confidence:** 4

**Summary:**

Do-Shapley values are expensive to estimate since they require evaluating exponentially many causal queries. The authors propose to make their estimation more feasible by employing two techniques in the context of do-Shapley values:
1) They propose to employ "estimand-agnostic (EA)" causal inference which -- instead of fitting a different model for every query (EB) -- fits a Structural Causal Model (SCM) once (offline) which encodes all possible interventional distributions, and which can then be used to answer the causal queries.
2) They propose to skip causal queries which are known to evaluate to zero since the intervention is causally independent of the target given the interventions that were already implemented before.
Furthermore, the authors touch on discussing whether do-Shapley values should be based on counterfactual (rung 3) instead of interventional (rung 2) causal queries, and how they can be defined depending on whether the prediction or the underlying target is explained.
In Experiments, the authors evaluate how the choice of SCM (linear vs nonlinear) affects the quality of the effect estimation in a small synthetic setting, demonstrate that do-SHAP is different from marginal SHAP, and show that contribution 2) allows speeding up the estimation of do-SHAP in small synthetic settings.

**Questions:**

You claim that the approach makes the estimation of do-Shapley feasible for "arbitrarily complex graphs". Where do you substantiate this claim?
You continue by saying that you achieve this "by using the estimand-agnostic approach". Where do you show that EA is responsible for the claimed performance improvements?

**Ethical Concerns:**

["NO or VERY MINOR ethics concerns only"]

**Final Justification:**

I raise my score by to acknowledge the bridging fields contribution of suggesting EA in the present context and since questions about the FRA algorithm were resolved during the author discussion period

**Limitations:**

Limitations and social impact are discussed appropriately.

**Paper Formatting Concerns:**

The fonts in the Figures are too small.

**Quality:**

2

**Strengths And Weaknesses:**

**Strengths**

*Significance:* I am convinced that in practice many causal queries can be skipped given knowledge of the causal graph, and that this can lead to a significant speedup in estimation. As such, the main idea of the paper has practical significance.

**Weaknesses**

*Clarity:* The paper touches on several distinct problems, and I found it challenging to understand which of them the authors actually intend to tackle:

- A Accurately estimating (many) causal queries from data. (EB vs EA)
- B Identifying queries that are redundant. (FWR)
- C Discussing whether do-SHAP are the "right" explanation method. (e.g. section 4.3)

In my view B is the main contribution, and I would recommend the authors to focus on clearly explaining that.

In particular, I was confused about the role of EB vs EA for the paper: The EA approach is prominently discussed in the introduction and conclusion, and the EA approach introduced in great length in the background. But in the main paper EA only receives very little attention (a bit at the beginning of 4.1) -- there is no methodological or empirical contribution.

In the main section of the paper, Section 4, there are three small subsections: 1 "Do-Shapley value", 2 "Efficient estimation of the do-Shapley values", 3 "do-Shapley explanations". 1 could be part of the background, and 3 should be left out or should be discussed together with 1. The really interesting part 2 is very dense and relevant details are missing (for example, what does it mean to $\Phi$-encode nodes?)

*Originality:* The paper employs existing ideas in a somewhat novel context. Luther et al (2023) already showed that Shapley approximation can be significantly accelerated using knowledge of the causal graph, the difference is that Luther et al make use of statistical independence to speed up conditional SHAP, while the reviewed paper uses causal independence to speed up do-SHAP.
I don't see relevant contributions in the context of EA vs EB.

*Quality:* The theory is sound. However, the proposed algorithm is quite complex given how simple the solution could be: Shapley values evaluate v(S u j) - v(S) for many sets S. Why not simply check causal independence of j and Y given S for every evaluation (very easy using a causal graph) and skip the evaluation if there is a causal independence in the spirit of Luther et al (2023)? What is the benefit of first identifying all possible irreducible frontiers first?

---

> ### Author Rebuttal · Authors · 2025-07-28
>
> Thank you for your review. We believe there are a number of misconceptions and would like to clarify them in the following.
>
> # Strengths and Weaknesses
>
> - We do tackle all three of the contributions expressed in sections 4.1 to 4.3. While FRA is the most novel contribution, it is not the only one; each is different in importance and nature, but they all contribute to making do-SVs feasible in practice in their own way. Additionally, we believe that the third one has been misunderstood. More details below.
> - “I was confused about the role of [estimand-based] EB vs [estimand-agnostic] EA”. The contribution w.r.t. the EB approach is in bridging the gap between EA general procedures and do-SV estimation (where the original authors “are not aware of any general causal effect estimators suitable for estimating the expression" [12]). With the base approach (EB), every single query $\nu(S)$ has its own estimand and estimation procedure one needs to follow; this is an ad-hoc process that renders do-SVs impractical whenever the number of variables is larger. Instead, with the EA approach, we can define a single model from which any (identifiable) query can be computed automatically, following a general estimation procedure that can be implemented as a simple function call. Identifying this connection is the contribution, which makes do-SVs practical and feasible in complex, larger graphs.
>
> The extensive discussion on the validity of the EA approach is to demonstrate that they are indeed capable of estimating do-SVs even though we do not use the corresponding estimands. This point was contested numerous times before by reviewers in the field, since EB is the dominant approach, which is why we devote a whole section (3.2) to explain why it works, as well as a set of experiments (5.1) to show how an EA method correctly estimates causal queries, and in particular, the true do-SV. (Additionally, we show how more complex architectures better model the data distribution, which in turn improves estimation performance). Section 4.1, where the EA approach is proposed for do-SVs, is much shorter since it does not require further explanation after the extensive previous section. Regardless, we did add Appendix H with a detailed example illustrating the application of our approach from start to finish.
> - About the three parts of section 4: 1 is part of the contribution for the aforementioned reasons. 3 explains how to approach Machine Learning explanations with do-SHAP, and presents Theorem 4.8, which is a novel contribution in itself, explaining the role of the exogenous noise on the do-SHAP game (under certain assumptions). Finally, we agree that section 4.2 (FRA) is quite dense, but this is as a result of space restrictions and the extensive discussion on the EA approach to properly demonstrate its validity, since without it, previous reviewers argued about this point and ignored our other contributions. We include Appendix D with a much more extensive explanation on this topic, together with all requisite proofs.
>
> About the “$\phi$-encode” terminology, it meant that a set $S$ is $\phi$-encoded by an integer $s$ if $s=\phi(S)$, with the injective encoding function $\phi$. This was meant for economy of notation, but we will clarify it in the camera-ready version. Other than that, was there anything else missing in our exposition?
> - Originality: We extend Luther’s approach in that it originally applied to conditional-SHAP (for do-SHAP, we had to derive new properties) and, more importantly, in that it does not only identify cases where $\nu(S \cup \{X}) = \nu(S)$ but instead returns the irreducible subset $S’$ such that $\nu(S) = \nu(S’)$. This is more powerful, as we will discuss in the following related point.
> - Quality: your proposed solution is less powerful than what is offered by the FRA algorithm. Consider an irreducible coalition S, and the maximal set $T \subseteq \mathcal{X}$ such that $T \cap S = \emptyset$ and $\forall X \in T, S \in Fr(X, Y)$. Then, $\forall T’ \subseteq T, \nu(S \cup T’) = \nu(S)$. Now, consider any $X \in \mathcal{X} \setminus S \setminus T$; there is no guarantee that $\nu(S \cup T’ \cup \{X\}) = \nu(S \cup T’)$ (because $T$ is maximal) so we must compute the difference according to Luther. However, the FRA algorithm tells us that $\nu(S \cup T’) = \nu(S)$ which can be cached for subsequent evaluations, thereby speeding up cases where Luther cannot offer any advantage. This applies to any irreducible coalition $S$ and superflous subsets $T’$. Hence the power of our proposal.
>
> Finally, note that we do not need to identify all possible irreducible coalitions beforehand. Instead, as we encounter new coalitions during do-SHAP execution, we can check which is the corresponding irreducible coalition and cache its value, so that further collisions can be omitted. We also made sure to make the FRA algorithm efficient, memory- (thanks to the integer formulation) and time-wise (as illustrated in our experiment 5.2; see figure 4b and 4c).
>
> # Questions
>
> When we claim that do-SHAP is made feasible on arbitrarily complex graphs due to the EA approach, we do not mean in terms of performance improvements, but on overcoming the ad-hoc nature of the EB approach, as argued above.
>
> Nevertheless, we do demonstrate that computational time is not prohibitive for higher values of K, on a graph with $K=100$ nodes. See the paragraph in Section 5.2 starting on line 323, where we compare computation time between baseline do-SHAP, a simple cache, and an FRA-cache.
>
> # Other comments
>
> Your score on clarity is 1 (poor), whereas other reviewers chose 3 (good) or 4 (excellent). Could you please clarify what specific points made it harder to read our work? Was there a section that was misunderstood or confusing?
>
> As for the fonts in the figures, we apologize for the inconvenience, but given space restrictions it was not possible to increase their size further. We do acknowledge this problem in the text and point towards the appendixes, where the smallest figures are replicated in a larger size for reader convenience.
>
> We hope to have clarified your concerns, and are open to further questions.
>
> We would like to kindly ask you to reconsider your review, in light of this rebuttal and the more favorable reviews by other reviewers. Thank you for your time and consideration.

---

> ### Comment · Reviewer_kUZY · 2025-08-04
> **Follow up questions**
>
> Thank you for the rebuttal. I have a  few quick follow-up questions.
>
> - In particular I don't understand the example about FRA being stronger than simply checking whether the surplus is zero before any evaluation. Could you clarify why FRA can identify surplus evaluations that evaluate to zero that cannot be identified with a quick check of the causal graph? (To check whether a variable causally influences another given some interventions we only have to check whether there is an open causal path from the variable to the target after removing incoming edges to the intervened upon variables.)
> - Are value functions with smaller sets easier to evaluate? (Assume that nothing is cached yet)

---

> > ### Author Response · Authors · 2025-08-05
> > **Response to follow-up questions**
> >
> > Thank you for bringing up these points; this line of questioning brought forth a new theorem that proves that FRA leads to less evaluations than Luther's approach. While the previous argument supported the claim, we believe this new theorem will answer your question unequivocally.
> >
> > Let us start with an illustrative example:
> >
> > Consider the chain graph $A \rightarrow B \rightarrow Y$. Every coalition is irreducible except for $\nu(A, B)=\nu(B)$. Now, for $\phi_A$, the pairs are $\nu(A)\neq\nu(\varnothing)$ and $\nu(A, B)=\nu(B)$, while for $\phi_B$, $\nu(B)\neq\nu(\varnothing)$ and $\nu(A, B)\neq\nu(A)$. In total, Luther's method would ask us to evaluate every single coalition (although $\nu(A,B)=\nu(B)$, they appear in the 4th and 3th unequal pair, respectively, so they will be evaluated regardless), while FRA would only ask to compute the irreducible sets $\varnothing,(A),(B)$, since $\nu(A, B)=\nu(B)$ (therefore, on the 4th pair, $\nu(A,B)$ will already be cached as $\nu(B)$).
> >
> > Indeed, at least in this simple graph, FRA results in less evaluations than Luther's method. Now, let us prove it in general:
> >
> > **Theorem**. Under Assumption 4.3 (in particular, all nodes in $\mathcal{X}$ are ancestors of $Y$), $\forall\textbf{S}\subset\mathcal{X}, \exists Z\in\mathcal{X}$ s.t. at least one of the following is true:
> > 1. $Z\not\in\textbf{S}$ and $\textbf{S}\not\in Fr(Z, Y)$.
> > 2. $Z\in\textbf{S}$ and $\textbf{S}\setminus\{Z\}\not\in Fr(Z,Y)$.
> >
> > **Remark**. If the theorem is true, for any pair $\nu(\textbf{S}\cup\{X\}) = \nu(\textbf{S})$ identified by Luther, we know that $\exists Z\in\mathcal{X}$ s.t. somewhere else in the do-SV computation, $\textbf{S}$ will appear and cannot be ommitted: either $Z\not\in\mathbf{S}, v(\mathbf{S}\cup\{Z\}) \neq v(\mathbf{S})$ or $Z\in\mathbf{S}, v(\mathbf{S}) \neq v(\mathbf{S}\setminus\{Z\})$ (non-parametrically). Therefore, Luther cannot omit these new pairs and will inevitably evaluate $\nu(\textbf{S})$, even if before it was omitted. By the same argument, the same can be said about $\nu(\textbf{S}\cup\{X})$.
> >
> > In other words, under Luther, all $2^K$ coalitions will need to be evaluated eventually, regardless if they were omitted previously in another pair. In contrast, FRA will progressively identify the irreducible subsets and only evaluate those, which will result in less evaluations necessarily (unless all nodes are parents of $Y$, which is the extreme case).
> >
> > **Proof**. We will reason by cases:
> >
> > 1. If $\textbf{S}=\mathcal{X}$, let us assume that $\forall Z\in\mathcal{X}$, $\mathbf{S}\setminus\{Z\} \in Fr(X,Y)$. Then, no node is a parent of $Y$ (if there were, it could never be blocked by any set), and therefore, no node in $\mathcal{X}$ is an ancestor of $Y$ in $G$, contradicting Assumption 4.3. Therefore, $\exists Z \in \mathcal{X}$ s.t. $Z\in\mathbf{S}$ and $\mathbf{S} \setminus \{Z\} \not\in Fr(Z, Y)$.
> >
> > 2. If $\textbf{S}=\varnothing$, $\forall Z\in\mathcal{X}, \varnothing \not\in Fr(Z, Y)$ (otherwise, $Z$ would not be an ancestor of $Y$, contradicting Assumption 4.3).
> >
> > 3. Let $\textbf{S} \neq \varnothing$, $\textbf{S} \neq \mathcal{X}$, and let $\textbf{S'} \subseteq \textbf{S}$ be its irreducible set (as determined by Theorem D.9 and Proposition D.10). We know that $\textbf{S'} \neq \varnothing$ (otherwise, $\forall Z \in \textbf{S}, \varnothing \in Fr(Z, Y)$, meaning that $Z$ is not an ancestor of $Y$, contradicting the assumption). Now, $\forall Z \in \textbf{S'}, \textbf{S'} \setminus \{Z\} \not\in Fr(Z, Y)$ (otherwise, $\textbf{S'}$ would not be irreducible).
> >
> > - If $\textbf{S}$ is irreducible, $\textbf{S} = \textbf{S'}$ and $\textbf{S} \setminus \{Z\} \not\in Fr(Z, Y)$, proving the result.
> >
> > - If $\textbf{S}$ is not irreducible ($\textbf{S} \subsetneq \textbf{S'}$), assume that $\textbf{S} \setminus \{Z\} \in Fr(Z, Y)$. By Theorem D.9, $Z$ belongs in the set of nodes that need to be removed from $\textbf{S}$ to create the irreducible set $\textbf{S'}$; therefore, $Z \not\in \textbf{S'}$, contradicting the assumption.
> >
> > Under all possible cases, the result is proven. $\square$
> >
> > The theorem proves that Luther can never lead to less evaluations than FRA, since the former eventually evaluates all coalitions, while the latter only evaluates the irreducible coalitions.
> >
> > This discussion will be added to the camera-ready version of the paper, as it can assuage similar doubts by other readers. I hope this clarifies your concerns and shows the value of our contribution as it was, now extended thanks to this clarification.
> >
> > Regarding your second question, while it is true that smaller coalitions are more expensive to evaluate than larger ones (since  every node not in $S$ requires to use its sampling mechanism to generate its values), this extra cost is negligible in comparison to evaluating extra coalitions that reduce to the same sub-coalition. This is an interesting point, however, and we will discuss it in the camera-ready version.
> >
> > Please let us know if you have any more questions.

---

### Official Review · Reviewer_Da1T · 2025-06-26

**Clarity:** 3
**Significance:** 3
**Originality:** 3
**Rating:** 5
**Confidence:** 4

**Summary:**

The manuscript proposes a practical method for computing do-Shapley values (do-SVs) — causal, interventional variants of Shapley value feature attributions — using an estimand-agnostic (EA) approach. The authors replace the original estimand-based (EB) procedure from prior work with an EA technique, which involves fitting a single structural causal model (SCM) to the data and using this proxy model to estimate any identifiable causal query. The paper introduces a novel Frontier-Reducibility Algorithm (FRA) that exploits graph structure to accelerate do-SV computation by identifying coalitions whose interventional effects can be safely skipped. Experiments on real and simulated data suggest that FRA can lead to major speedups in practice.

**Questions:**

Full disclosure: I have reviewed a previous version of this manuscript. I think it is much improved, and am generally disposed to recommend acceptance. I have just one major question/comment, and a few minor ones.

**Major comment**

*Clarifying tractability*: As noted above, there is some confusing language around the precise advantages conferred by the FRA approach. Experiments appear to confirm that this often speeds things up in practice, as we would expect. But the word "efficient" has a precise meaning in computer science – or maybe I'm just old fashioned :) – and it is clear from the discussion in Sect. 5 that the proposed method is not, in fact, a polytime algorithm. Now it may turn out to be polytime under some graph sparsity assumptions, and it would be very cool to spell this out in a bit more detail. But in lieu of such an analysis, I recommend striking all mention of "efficient" algorithms for arbitrary graphs, etc.


**Minor comments/questions**

-Since EA methods rely on a known, accurate causal graph, how sensitive are the estimated do-SVs to graph errors?

-Linearity is usually included at a Shapley value axiom (Appx. A).

**Ethical Concerns:**

["NO or VERY MINOR ethics concerns only"]

**Final Justification:**

I said in my initial review that I would revise my score upward if the authors could address my comments. I am satisfied that they have done so (despite some lingering confusion about the appropriate use of the word "efficient"). I have therefore upped my score from 4 to 5 and wish the authors the best of luck.

**Limitations:**

Yes.

**Quality:**

3

**Strengths And Weaknesses:**

**Strengths**

*Problem significance*: Extending Shapley values to causal settings is important for fair, trustworthy, and scientifically grounded explainability in ML.

*Originality*: As far as I'm aware, the proposed EA framework for computing do-SVs is novel, contributing to both the explainability and causal inference literatures.

*Technical contribution*: The FRA algorithm is a well-motivated and technically interesting technique to exploit graph structure for computational gains. Though somewhat similar to prior work (Luther et al., 2023), the similarities and differences are properly acknowledged in the manuscript.

*Empirical validation*: The experiments convincingly demonstrate that the method works on synthetic and real-world data.

*Clear exposition*: The paper is generally well-written, with clear descriptions of causal concepts, algorithm design, and experimental setup. I especially appreciated the thorough appendix.

**Weaknesses**

*Confusing and potentially misleading claims about tractability*: Throughout the paper, the authors repeatedly imply that their method renders do-Shapley value computation practical on arbitrarily complex graphs (e.g., abstract lines 4–7; intro, final paragraph; top of Sect 4; conclusion, first paragraph). However, the method does not in fact circumvent the fundamental #P-hardness of Shapley value computation, as acknowledged in Sect. 5.2. The FRA algorithm accelerates the process in some cases but does not resolve the exponential scaling in the number of features; more generally, the speedups it offers appear to be a function of graph sparsity, although the precise nature of this relationship is never spelled out in detail. This is a missed opportunity, as there has been a number of interesting works on the complexity of Shapley value computation in the last few years (Van den Broeck et al., 2022; Arenas et al., 2023; Marzouk et al., 2025). This tension between claimed practicality and residual intractability needs to be clarified.

*Limited discussion of identifiability limitations*: The method assumes all required causal queries are identifiable from the causal graph, but the paper downplays the practical difficulty of ensuring this in complex or partially observed systems.

*No quantitative comparison to baseline SHAP runtimes*: The paper lacks a direct empirical comparison of total runtime or computational cost between EB do-SHAP, conditional/marginal SHAP, and the proposed EA do-SHAP+FRA method. (Unless I'm misreading Fig. 4c?)

**References**

-https://www.jair.org/index.php/jair/article/view/13283/26815

-https://jmlr.org/papers/v24/21-0389.html

-https://arxiv.org/abs/2502.12295

---

> ### Author Rebuttal · Authors · 2025-07-28
>
> Thank you for your review and for your transparency in disclosing having reviewed a previous version of this work before. We will address your concerns section by section.
>
> # Strengths and Weaknesses
>
> - “*Confusing and potentially misleading claims about tractability”:* while we acknowledge the importance of the tractability of Shapley values (be it marginal, conditional, or causal-interventional) we do not address this topic in our work nor do we claim to do so. We were very careful to avoid talking about tractability anywhere in this work so as not to misrepresent our contribution; this was a common concern of previous reviewers and we rectified accordingly.
>
> When we talk about practicality, we do not refer to the #P-#NP question, but on the fact that estimand-agnostic (EA) approaches make it possible to automatize do-SHAP estimation once the corresponding SCM is trained (assuming do-SV identifiability). This is in contrast with do-SHAP’s authors proposal with estimand-based (EB) approaches, which required defining separate estimands for each coalition and following those estimands to arrive at the final do-SV estimation. This ad-hoc approach is overcome with the EA approach and its general estimation procedures, which is what makes our approach “practical”, in the sense of being adaptable to general graphs with more nodes than what we can assume with an ad-hoc EB approach.
>
> On the other hand, the FRA algorithm reduces the number of computations to be carried out throughout do-SV estimation, but this cannot resolve the exponential nature of exact do-SHAP, nor was it meant to. Instead, we propose its application to the approximation method with sampled permutations (section 3.4); here the FRA algorithm can really speed-up computation of do-SV estimation, as shown in the experiments.
>
> To summarize, the issue of tractability falls out of scope of our contribution, but we agree on its importance and would be a worthwhile future direction. We will include this discussion in the camera-ready version to further clarify this point.
> - *“Limited discussion of identifiability limitations”:* we do acknowledge this topic, and state it as a central limitation of our work (as well as for any other estimation approach without further assumptions): assumption 4.5, footnote 5, and Appendix H.1. Please note that there are algorithms (as pointed out in line 127) to determine whether a certain query is identifiable (no matter how complex the graph is, or if it contains latent variables), which can be used to determine the identifiability of any coalition query $\nu(S)$. The only remaining point is to develop a general criterion for do-SHAP identifiability based on graph structure, but so far we have not found one; we left it for future work (line 372).
> - “*No quantitative comparison to baseline SHAP runtimes”:* as discussed with reviewer #Mmbz (see the Strengths and Weaknesses section in our rebuttal), it is not possible to compare runtime between the default EB do-SHAP approach and our EA approach. EB’s drawback is in its ad-hoc nature that requires the manual definition of estimators based on estimands for every coalition, in contrast with the EA’s approach that defines a single SCM with automatic general procedures for any coalition query. As such, the improvement lies in the generality of our approach and the possibility to automatize the whole process (as implemented in our code) once the SCM is trained.
>
> As for the marginal/conditional approaches, we do not believe it makes sense to compare time-wise given that they estimate completely different quantities (see Figure 3 for a comparison between marginal-SVs and do-SVs, and the paragraph on line 27, extended on Appendix G with an actual Data Generating Process).
>
> # Questions
>
> - Major comment: we understand your point. We argued our position in the previous section on tractability and will include a section in the camera-ready version for further clarification. As for the efficiency aspect, we meant efficiency as in reducing the resources needed to run the FRA algorithm: memory-wise, thanks to the integer encoding, and time-wise, as illustrated in Figure 4b and c, where FRA takes negligible time in comparison with the $\nu(S)$ computation. We do not mean to imply that our do-SHAP estimator runs in polynomial time. After these points and the new planned section discussing this topic, do you believe it necessary to remove any mentions of efficiency?
> - Sensitivity to graph misspecification: this is an important point and we express it as a limitation of our work; see line 338 about the lack of doubly-robust guarantees. This lack is not specific to our approach though, but general for EA approaches, in contrast with EB alternatives; however, the impracticality of EB methods in this setting justifies the necessity for EA on do-SHAP, even without these guarantees. As for sensitivity studies, since our approach is general in the sense that any EA techniques could be applied (as long as they admit semi-Markovian graphs), we believe it would not make sense to run a sensitivity study on a specific class of EA model, as it would not generalize to other models. However, it is an important concern and we will point it out in the camera-ready version to signal novel directions for this work.
> - Linearity: that is true, thank you for pointing out this omission. We will include it in the appendix.
>
> We hope these arguments will ease your concerns. Let us know if there are any further questions.

---

> > ### Comment · Reviewer_Da1T · 2025-08-03
> > **Reply to authors**
> >
> > I thank the authors for their detailed reply to my comments. I insist that "efficient" is a dangerous word to bandy about in technical contexts when you just mean "faster than other stuff". It is easily misread as "polytime", especially when you make claims about the usability of the algorithm for "arbitrary graphs". The latter is especially misleading, since it sounds an _awful lot_ like you're saying the method's complexity scales as $\mathcal O(\text{poly}(n))$. Of course, on careful reading, you make no such claim. But to avoid any potential confusion, I would go through the manuscript very closely – especially the passages I flagged in my initial review – to avoid this misunderstanding.
> >
> > With that point out of the way, I am satisfied with the rest of the authors' replies and will adjust my score upward accordingly. Overall, I really enjoyed this manuscript and sincerely hope to see the camera ready version in the proceedings! :)

---

> > > ### Author Response · Authors · 2025-08-04
> > >
> > > Agreed, we will carefully integrate these changes to avoid any misunderstanding in our contributions.
> > >
> > > Thank you again for your time and consideration!

---

### Official Review · Reviewer_oZ6x · 2025-07-02

**Clarity:** 4
**Significance:** 3
**Originality:** 3
**Rating:** 4
**Confidence:** 3

**Summary:**

The Shapley value is a well-known concept from game theory. We are given a set of players X and a valuation function v which assigns each subset S ("coalition") of X a value v(S). The Shapley value of a variable P is the average value that adds P to every coalition S, that is, v(S \cup X) - v(S).

The Shapley-value is now used to explain the influence of certain observed samples x on a target variable Y. To do so, the valuation function is set to E[Y | do(S = s)] for every subset S of X. Here s is the restriction of x to S. This assumes that there is an underlying SCM whose structure as a graph is known s.t. the expected values are all identifiable (using do-calculus).

To turn this into a practical scheme, the authors present an estimand-agnostic approach. To compute the Shapley value, one needs to estimate an exponential number of terms. The key is to just learn the distribution of a single SCM, then one can check whether v(S) is identifiable using do-calculus and then estimate v(S) using the learned distribution and SCM-calculation rules.

The authors present a way to calculate the do-Shapley value faster by the following observation. If X is an ancestor of Y and the coalition we want to estimate contains a blocking set between X and Y, then adding X to the coalition does not change the value of the coalition. This allows one to do a faster than brute-force computation, the authors design a nice frontier reduction algorithm to perform this computation fast.

The author also provide an implementation and do extensive testing.

**Questions:**

- Are there any known formal hardness results for computing Shapley values in this setting? Shapley value per se is an exponential sum, but depending on the valuation function, it can be done faster.
- Can one get estimates in Appendix B when also the underlying SCM is taken into account?

**Ethical Concerns:**

["NO or VERY MINOR ethics concerns only"]

**Final Justification:**

As I said before, I like the theoretical parts of the paper. However, since I working on the theoretical side of causality, I am not very confident in evaluating the practical aspects, even after rebuttal. Therefore, I maintain my score.

**Limitations:**

Yes

**Quality:**

3

**Strengths And Weaknesses:**

I think this is a nice paper. I combines nice theoretical insights with a practical implementation. The proofs presented in Appendix D are very detailed and well-written.  I am mainly working on ML theory, therefore I found it hard to evaluate the partical part. Therefore, my confidence is only 3. My reservations only stem from the fact that I am not able to confidently evaluate the practical part.


Comments:

- line 91: The permutation only induces an ordering if S is ordered
- line 152: SCM G -> graph G (?)
- line 186: Assumption 4.1 needs to be reworded in my opinion
- line 205: Should Y also not be in the blocking set
- line 210: An irreducible frontier would be called minimal cut in graph theory
- line 219: I find it easier to write that you you encode the set by its characteristic vector stored in an integer. Also the word $\phi$-encodes is not defined.
- line 223: That being said, I would present the set-version (in the appendix) of the algorithm as Algorithm 1. While of course the presented version runs faster, the set-version is much easier to understand.

---

> ### Author Rebuttal · Authors · 2025-07-28
>
> Thank you for your appreciative review. We will address your questions section by section:
>
> # **Strengths And Weaknesses**
>
> Thank you for these Comments (C). We will include your suggested improvements in the camera-ready version.
>
> - C1. Do you mean $\pi$ only induces an ordering if the elements are *unique*? That is true, we will clarify this part.
> - C2. Noted: “underlying SCM’s graph G”.
> - C3. We acknowledge the wording is quite dense, due to the limited paper size. Thank you for pointing this out, we will extend it briefly to make it more understandable in the camera-ready version.
> - C4. It is not necessary, because $\mathbf{X}$ does not include Y (line 203), so the frontier $\mathbf{S} \subseteq \mathbf{X}$ cannot include Y.
> - C5. Thank you for pointing this out, we prefer to preserve the frontier terminology in this case to make it more easily understandable.
> - C6. The $\phi$-encode terminology (i.e., a set $S$ is $\phi$-encoded by an integer $s$ if $s=\phi(S)$ with the injective encoding function $\phi$) was meant for economy of notation in trying to shorten the paper. We will clarify it.
> - C7. That is a good point. We opted for the numerical version to show the final version of FRA in the main text, leaving the preceding version with sets for the appendix along with more detailed explanations about the two.
>
> # Questions
>
> - Q1. Not that we are aware of; the do-SHAP value has not seen much use in the community yet (possibly due to the estimand-based (EB) approach suggested by the authors, which limited its practicality) so there has not been much work on this aspect. Regardless, it is rarely the case where we compute the exact formula for SHAP, instead using the approximate method by sampling permutations. However, further work in this direction would surely be worthwhile for do-SHAP practicality.
> - Q2. We are not sure if we understand your question. The results in Appendix B are general for any kind of SHAP estimation with the cache strategy, to measure the impact of a (non-FRA) cache on the approximation algorithm for an arbitrary number of nodes $K$. This formula is applicable to SHAP as a whole, not specifically do-SHAP. If you meant what the impact of the FRA-cache would be knowing the underlying graph G, this depends on the topology of the graph and it is hard to give conclusive answers in general. However, we did include an experiment related to this point on Figure 4a, where we see the ratio of coalitions left to evaluate after running FRA. Does this answer your question?
>
> Thank you again for your insightful review. Let us know if there are any more questions.

---

> > ### Comment · Reviewer_oZ6x · 2025-08-03
> >
> > Thank your for your detailed review. Indeed this is what I meant with question 2. I will take your comments into account for my final evaluation.

---

> > > ### Author Response · Authors · 2025-08-04
> > >
> > > Thank you for your time and consideration. Let us know if there are any more clarifications needed.

---

### Official Review · Reviewer_Mmbz · 2025-07-03

**Clarity:** 3
**Significance:** 2
**Originality:** 2
**Rating:** 4
**Confidence:** 4

**Summary:**

The authors claim the following contributions:
1. The work constructs an neural causal model-style approach to estimate do-Shapley-values, assuming identifiability.
2. The work also limits the exponential number of terms computed requusing their Frontier-Reduction Algorithm (FRA) to generate irreducible subsets.

**Questions:**

1. Is there a performance or runtime improvement over (Jung et al., 2022)'s estimand based approach when not using FRA?
2. Is it correct to say that FRA does not yield runtime improvements when all variables X affect Y directly (e.g., when $|Pa_Y| = |\mathbf X|$ or $p = 1$)?
3. How significant is the improvement with respect to the proportion of variables that do affect Y directly (e.g., how does the improvement change as $p$ goes to 1, given it is currently set to 0.25)?
4. Does the FRA algorithm work with semi-Markovian causal diagrams?

**Ethical Concerns:**

["NO or VERY MINOR ethics concerns only"]

**Final Justification:**

The authors highlighted the size of the empirical performance gain as proportional to $p$, the proportion of variables which directly cause the outcome.

**Limitations:**

Yes.

**Paper Formatting Concerns:**

None.

**Quality:**

2

**Strengths And Weaknesses:**

Originality: The idea of training a proxy model to estimate do-Shapley values specifically is only marginally novel, given that it comes from the neural causal model framework. The concept of the frontier reducibility algorithm seems logical, stemming from d-separation results, and it primarily seems to be an algorithm for empirically speeding up do-SV estimation.

Significance & quality: The contribution seems primarily empirical and are claims are the made on the basis of experiments on synthetic data. There does not seem to be a performance or runtime comparison between estimand-based approaches to do-Shapley estimation and the described neural causal model-style approach, so it is challenging to see an improvement from baseline. In cases where not all variables have direct effects on Y, FRA is shown to boost estimation efficiency.

Clarity: The work is clearly written.

---

> ### Author Rebuttal · Authors · 2025-07-28
>
> Thank you for your review. We will answer your concerns section by section.
>
> # Strengths and Weaknesses
>
> While the neural causal model framework (i.e., the EA framework) is not a novel contribution, its application to the do-SHAP estimation problem is, in our opinion, significant, given that without it, we would need to define and follow separate estimands for each of the $\nu(S)$ queries, which renders do-SHAP impractical in settings with a moderate number of nodes (2^K queries/estimands in a graph with K nodes). Let us remark that it is in the impracticality of the EB approach for this setting that ours is “faster”, since the process can be automatized through general estimation procedures, as described in [13], only requiring human input once during the DCG definition/training phase.
>
> While this contribution may be seen as incremental, we believe it important to be brought forth to the community given that the authors of do-SHAP did not acknowledge the EB approach as a possibility for estimating do-SVs, which limited its applicability: “[we] are not aware of any general causal effect estimators suitable for estimating the expression" [12]. We do propose such an estimator, and here lies this part of the contribution.
>
> Finally, regarding the request for runtime comparisons between EB and EA approaches, these do not make sense; as stated, EB’s main drawback is in its ad-hoc nature, depending on the causal graph and particular query to estimate, whereas EA approaches are general for any graph and query (provided the given query is identifiable, for which there are also algorithms to automate this check) and therefore can be automatized. Hence, we could not possibly compare their runtime. In other words, it is not in runtime that this contribution lies, but in making it feasible/practical to estimate do-SVs on graphs with larger number of nodes (K).
>
> The other (and probably more significant) part of our contribution lies on the definition of the FRA algorithm, which significantly speeds up do-SV estimation. Finally, we also derive Theorem 4.8, which lets us measure the do-SV of the target’s exogenous noise at no computational cost under certain assumptions.
>
> # # Questions
>
> - Q1. The FRA contribution is independent to the EB/EA choice; FRA could be used with Jung’s EB approach and would inevitably result in an improvement because it can only reduce the number of coalitions to compute at a negligible cost (see Figure 4b). On the other hand, as stated before, the improvement of the EA approach over the EB approach is not in runtime, but in practicality for larger graphs when using the automatized EA approach.
> - Q2. Yes, this is addressed in Remark 4.6, Footnote 7 and Section 6 (line 352). There we argue that, even though FRA cannot bring any improvement over graphs where *all* proper nodes are parents of the target Y (i.e., no frontiers exist), this kind of graph is unrealistic and, in the case of ML models, unadvisable, since it could lead to spurious correlations, anti-causal effects, overfitting or adversarial vulnerability (see line 358). Hence why we believe this limitation is not of practical importance in general settings.
> - Q3. For the experiment in Figure 4b/c, we set $p=0.25$ instead of replicating it for several values of p due to the costly nature of the experiment. However, we did measure the impact of p indirectly through Figure 4a: there, for several values of p, we can see the proportion of coalitions that need to be evaluated after FRA, which tends towards p. In other words, we can expect a reduction ratio of p thanks to FRA (i.e., 50% total time w.r.t. baseline do-SHAP for p=0.5). I hope this answers your question.
> - Q4. Yes, FRA works irrespective of whether the underlying queries are identifiable or not, because any coalition $S$ that can be reduced to $S’$ entails that their respective queries are equal $\nu(S) = \nu(S’)$ on distributions P(V) Markov-compatible to the graph, be it semi-Markovian or not. See the proofs on Appendix D.2.
>
> I hope we have addressed your questions sufficiently; let us know if there are any more concerns. Thank you for your review.

---

> > ### Comment · Reviewer_Mmbz · 2025-08-08
> > **Thank you for your detailed response.**
> >
> > I believe the notion of an "EA" approach may still be too small to be significant, given that it is a direct application of previous work.
> >
> > On the other hand, FRA seems to be novel and useful for the purpose of improving computation of do-Shapley values.
> >
> > I have updated my score by one point.

---

### Note · Authors · 2025-08-13

Thank you for the extensive and insightful discussion. Your feedback helped identify minor issues, omissions, and dense sections, all addressed in the final version. In this final remark, let us highlight the strengths identified by reviewers and common concerns that have been answered.

## Strengths

- The work’s relevance in bringing causal considerations to explainability and bridging do-SHAP with Estimand-Agnostic (EA) causal estimation.
- Introduction of the Frontier-Reducibility Algorithm (FRA), reducing coalition evaluations in do-Shapley Value (do-SV) estimation with negligible overhead.
- Clear writing (except §4.2), thorough 21-page appendix, and detailed proofs.

## Concerns

- Some saw applying EA to do-SHAP as only marginally novel, and that explaining why EA works for do-SHAP (§3.2) took too much space. We argue EA’s generality overcomes the default (estimand-based, EB) ad-hoc nature, making do-SV practical for non-experts. Even do-SHAP’s authors were unaware of this link [12]. We justify why EA works, even on semi-Markovian graphs, as often questioned by previous reviewers. Finally, we show empirically (§5.1) that EA works with do-SVs, which is novel and important, given that compounding errors in each coalition evaluation towards do-SV's result might have rendered EA infeasible in this context.
- No EB vs. EA runtime comparison. Such comparison is meaningless since EB depends on specific graphs/queries, while EA is general and automatable. This part of our contribution lies in feasibility in complex graphs, not runtime gains.
- Concern over implied polytime performance. We never claimed tractability and will revise mentions of efficiency to avoid this impression. Our results show practicality: 100-node, 1,000-permutation do-SHAP in 21m14s, with FRA saving 4+ minutes and scaling linearly in overhead, only amounting to 8s in total.
- FRA seen as incremental over Luther [36], and §4.2 was too dense. Luther handled conditional-SHAP; we adapt to do-queries proving new properties, and go beyond zero-differential pairs by identifying irreducible subsets, enabling a more effective cache. We added a theorem proving fewer evaluations than Luther (see #kUZY). In the final version, §4.2 will be clearer, using the set-based algorithm from §D2.2 and an illustrative figure.

We hope this summary addresses remaining concerns and supports acceptance of this work, making Causal Inference more accessible to the explainability community.

---

### Decision · Program_Chairs · 2025-09-17

**Decision:**

Accept (spotlight)

**Comment:**

Summary: This paper addresses the problem of efficiently computing *do-Shapley values* (do-SVs), which are causal, interventional analogues of Shapley value feature attributions. The authors propose an *estimand-agnostic* (EA) approach: instead of computing each interventional expectation from scratch, they fit a single neural structural causal model (SCM) to the data, which can then be used to estimate any identifiable causal query. The key technical contribution is the *Frontier-Reduction Algorithm* (FRA), which leverages d-separation and graph structure to avoid redundant computations by identifying irreducible coalitions—subsets of variables whose addition to a coalition can affect the outcome. This reduces the number of terms that must be computed, potentially offering significant speedups over brute-force enumeration. The method is evaluated on synthetic and real data, demonstrating empirical gains in runtime and estimation efficiency, particularly when the proportion of variables with direct effects on the outcome is low.

Strength:

1. FRA is a well-motivated and technically interesting algorithm that exploits causal graph structure to accelerate do-SV computation. The theoretical insights are clearly presented, the proofs are detailed.

2. The paper is well-written, with clear explanations of causal concepts, algorithmic design, and experimental setup.

3. The method is implemented and tested on both synthetic and real data, with results showing practical speedups in relevant settings.

Weakness:

1. The paper sometimes overstates the practical tractability of the approach. While FRA can yield significant speedups, it does not resolve the fundamental exponential complexity of Shapley value computation in the general case. The relationship between graph structure (e.g., sparsity) and computational gains is not fully characterized.
2. There is a lack of direct, quantitative runtime comparisons between the proposed method, prior estimand-based approaches, and standard SHAP baselines. This makes it difficult to assess the practical impact of the method in absolute terms.
3. The method assumes all required interventional queries are identifiable from the causal graph, but the challenges and limitations of this assumption in real-world settings are not sufficiently discussed.
4. The robustness of the approach to errors in the assumed causal graph or SCM is not empirically or theoretically analyzed.
5. The empirical evaluation is still somewhat limited in scope and does not fully explore the method’s behavior in complex, high-dimensional, or partially observed systems.


This is a technically solid and well-presented paper with a meaningful contribution to causal explainability. The paper would benefit from a more nuanced discussion of its limitations and a more thorough empirical evaluation. The reviewers converged on acceptance decisions with 3 borderline accept and 1 accept.